# Adversarial Robust Generalization of Graph Neural Networks

Chang Cao[1]  Han Li[1,2]  Yulong Wang[1,2]  Rui Wu[3]  Hong Chen[1,2]

## Abstract

While Graph Neural Networks (GNNs) have shown outstanding performance in node classification tasks, they are vulnerable to adversarial attacks, which are imperceptible changes to input samples. Adversarial training, as a widely used tool to enhance the adversarial robustness of GNNs, has presented remarkable effectiveness in node classification tasks. However, the generalization properties for explaining their behaviors remain not well understood from the theoretical viewpoint. To fill this gap, we develop a high probability generalization bound of general GNNs in adversarial learning through covering number analysis. We estimate the covering number of the GNN model class based on the entire perturbed feature matrix by constructing a cover for the perturbation set. Our results are generally applicable to a series of GNNs. We demonstrate their applicability by investigating the generalization performance of several popular GNN models under adversarial attacks, which reveal the architecture-related factors influencing the generalization gap. Our experimental results on benchmark datasets provide evidence that supports the established theoretical findings.

## 1. Introduction

As powerful architectures for processing complex graph-structured data, GNNs have shown excellent performance in certain security scenarios (Yan et al., 2023), such as malware detection (Liu et al., 2023; Gu et al., 2024), intrusion detection (Zhou et al., 2021; Tran & Park, 2024), and blockchain (Cai et al., 2023; Seo et al., 2024). Despite their tremendous success, Zügner et al. (2018) first demonstrate the

susceptibility of GNNs to adversarial attacks in node classification tasks. These attacks are deliberately designed by the attackers to mislead the graph model's prediction, which significantly deteriorates the performance and applicability of GNNs in security-related domains (Zhang & Zitnik, 2020; Finkelshtein et al., 2022; Liu et al., 2023). Therefore, it is crucial to enhance the adversarial robustness of GNNs for node classification tasks.

One principal approach to learning robust models is adversarial training, which has proven to be a powerful defense against adversarial attacks (Bai et al., 2021). Its core mechanism is a min-max optimization problem by incorporating the adversarial samples that maximize the classification error into the training to minimize the classification error (Goodfellow et al., 2014; Madry et al., 2019). The empirical advancements of adversarial training in the literature (Wang et al., 2020; Lin et al., 2023) prompt numerous studies to develop relevant theoretical analysis. However, generalization analysis in adversarial learning presents more obstacles as compared to its non-adversarial counterpart, i.e., the non-smoothness of adversarial loss caused by the maximization operation in the min-max process (Bai et al., 2021). Previous works mitigate the negative impact of non-smoothness via restricted assumptions about losses or model functions (Awasthi et al., 2020; Gao & Wang, 2021; Xiao et al., 2022), or controlling the adversarial perturbation set (Mustafa et al., 2022). Since the aforementioned works are confined to non-graph data, this raises a question of whether adversarial generalization can be extended to graph learning.

Unlike non-graph data, each node in GNNs aggregates messages from its neighbor nodes through the message-passing mechanism. This causes the accessibility of predictors to unlabeled samples in the test set during training (Oono & Suzuki, 2020), which results in the invalidation of previous analytical methods based on inductive learning (Günnemann, 2022; Mustafa et al., 2022). The information interaction of nodes also leads to the correlation of perturbations between different nodes, making the adversarial perturbation set of graph data different from that of non-graph data.

To overcome these difficulties, we explore the adversarial generalization of GNNs in transductive settings by assessing the complexity of the adversarial loss function class through

[1]College of Informatics, Huazhong Agricultural University, Wuhan 430070, China [2]Engineering Research Center of Intelligent Technology for Agriculture, Ministry of Education, China [3]Horizon Robotics, Haidian District, BeiJing 100190, China. Correspondence to: Han Li <lihan125@mail.hzau.edu.cn>.

*Proceedings of the 42nd International Conference on Machine Learning*, Vancouver, Canada. PMLR 267, 2025. Copyright 2025 by the author(s).

*Table 1.* Summary of generalization analysis of GNNs ($m$-number of training data, $u$-number of test data).

| Reference | Under Attack | Learning Mode | Analysis Tool | Convergence Rate |
|---|---|---|---|---|
| Verma and Zhang (2019) | No | Inductive | Uniform stability | $\mathcal{O}(1/\sqrt{m})$ |
| Zhou and Wang (2021) | No | Inductive | Uniform stability | $\mathcal{O}(1/\sqrt{m})$ |
| Garg et al. (2020) | No | Inductive | Rademacher complexity | $\mathcal{O}(1/\sqrt{m})$ |
| Oono and Suzuki (2020) | No | Transductive | Rademacher complexity | $\mathcal{O}(\max\{1/\sqrt{m}, 1/\sqrt{u}\})$ |
| Esser et al. (2021) | No | Transductive | Rademacher complexity | $\mathcal{O}(\max\{1/\sqrt{m}, 1/\sqrt{u}\})$ |
| Tand and Liu (2023) | No | Transductive | Rademacher complexity | $\mathcal{O}((1/m + 1/u)\sqrt{m + u})$ |
| Sun et al. (2024) | No | Transductive | PAC-Bayes | $\mathcal{O}(1/\sqrt{m})$ |
| Ours | Yes | Transductive | Covering number | $\mathcal{O}(\max\{1/\sqrt{m}, 1/\sqrt{u}\})$ |

covering number. Our analysis exhibits broad applicability across a wide range of GNNs and losses. Our findings provide theoretical support for the empirical success of adversarial training and offer valuable insights for training robust GNNs that generalize well. The key contributions are as follows.

- We derive the first high-probability generalization bound of general GNNs in adversarial learning. Our analysis focuses on controlling the covering number of the whole feature matrix affected by the perturbation set, thereby achieving the complexity estimate of the adversarial loss. This addresses the key obstacles of extending the adversarial generalization to graph learning, caused by interactions between perturbations.

- We conduct a comprehensive analysis on several popular GNNs and derive the covering number bound of each perturbed GNN model. Our results indicate the relation between the adversarial perturbations and graph-based predictors, revealing the role of some GNN-related factors for reducing adversarial generalization, such as appropriate model architecture selection, weight norm normalization, etc..

- Based on our theoretical findings, we analyze and compare the key factors influencing generalization capability in adversarial learning, which is confirmed in our experimental evidence.

## 2. Related Work

**Adversarial training on GNNs.** To enhance the adversarial robustness of GNNs, Chen et al. (2020a) introduce Smoothing Adversarial Training (SAT), which improves GNNs' robustness by reducing the amplitude of adversarial gradients. Jaeckle and Kumar (2021) propose a novel method called AdvGNN, which efficiently generates adversarial examples by combining elements from both optimization-based attacks and generative methods. Yang, Zhang and Yang (2021) develop Graph Adversarial Self-supervised Learning (GASSL), which can automatically generate challenging views by adding perturbations to the input, thereby facilitating adversarial training on the encoder. Kone et al. (2022) introduce Free Large-scale Adversarial Augmentation on Graph (FLAG), which augments node features for better performance under attack by iteratively adding gradient-based adversarial perturbations during training. Deng et al. (2023) propose Batch Virtual Adversarial Training (BVAT), which promotes output smoothness of GNNs by applying virtual adversarial perturbations to the nodes. Due to the effectiveness of these empirical works in overcoming the vulnerability of GNNs against adversarial attacks, it is crucial to provide theoretical support for these advancements.

**Generalization analysis of GNNs.** Verma and Zhang (2019) first apply uniform stability on one-layer GCN to derive a generalization bound. Zhou and Wang (2021) extend their work to multi-layer GCNs and show a exponentially dependency of generalization on the number of layers. Through the lens of Rademacher complexity, Garg et al. (2020) establish the first data-dependent generalization bound of message-passing GNNs; Oono and Suzuki (2020) provide generalization and optimization guarantees via boosting theory, and focus on a specific type of multi-scale GNNs; Esser et al. (2021) establish the generalization bound of GNNs in the semi-supervised transductive setting and demonstrate the effectiveness of residual connections in improving generalization of GNNs; Tang and Liu (2023) derive high probability bounds of several popular GNNs, including both linear and non-linear models. Moreover, Sun et al. (2024) develop a PAC-Bayesian bound for GNNs, which incorporates the interplay in the message passing mechanism. Table 1 summarizes the related works on generalization analysis of GNNs. Though the above studies cannot be directly extended to adversarial settings due to the maximization over the adversarial loss, they provide valuable insights into the adversarial generalization analysis.

## 3. Preliminary

Given an undirected graph $G = (\boldsymbol{A}, \boldsymbol{X})$ with $n$ nodes, where $\boldsymbol{A} \in \mathbb{R}^{n \times n}$ is the adjacency matrix and $\boldsymbol{X} =$

$[\boldsymbol{x}_1, \ldots, \boldsymbol{x}_n] \in \mathbb{R}^{n \times d}$ contains $n$ node features with $d$ dimensions. We consider transductive learning in our analysis, where the unlabeled test samples are available during training. Let $S = \{\boldsymbol{z}_i = (\boldsymbol{x}_i, y_i)\}_{i=1}^n$ be the set of instances, where $\boldsymbol{x} \in \mathcal{X}$, $y \in \mathcal{Y}$ and $\boldsymbol{z} \in \mathcal{Z} = \mathcal{X} \times \mathcal{Y}$. A labeled training set $S_m$ with size $m$ is selected from $S$ randomly. The goal of transductive learning is to predict the labels of the samples in the test set $S_u$, where $n = m + u$. The empirical error over the training set $S_m$ is denoted by $R_m(f) = \frac{1}{m} \sum_{i=1}^m \ell(f_i(\boldsymbol{A}, \boldsymbol{X}, \boldsymbol{W}), y_i)$, and the population risk over the test set $S_u$ is denoted by $R_u(f) = \frac{1}{u} \sum_{i=m+1}^{m+u} \ell(f_i(\boldsymbol{A}, \boldsymbol{X}, \boldsymbol{W}), y_i)$, where $\boldsymbol{W}$ is the learning parameters, $f \in \mathcal{F} : \mathcal{X} \to \mathbb{R}^n$ represents the output function with $f_i(\cdot)$ denoting the prediction of each node feature $\boldsymbol{x}_i$, and $\ell \in \mathcal{L} : \mathcal{F} \times \mathcal{Y} \to \mathbb{R}$ is the loss function.

However, by perturbing the input samples and maximizing the loss, the adversary can mislead the classifier to make erroneous predictions (Huang et al., 2016). Given a noise space $\mathcal{B} = \{\boldsymbol{\delta} : \|\boldsymbol{\delta}\|_\infty \leq \theta\}$, for any $\boldsymbol{\delta} \in \mathcal{B}$, we assume that a set of adversarial nodes are generated from the neighbor $\mathcal{A}_{\boldsymbol{\delta}} = \{\widetilde{\boldsymbol{X}} = [\tilde{\boldsymbol{x}}_1, \ldots, \tilde{\boldsymbol{x}}_n], \tilde{\boldsymbol{x}}_i = \boldsymbol{x}_i + \boldsymbol{\delta}\}$. The adversary create the worst case perturbation $\boldsymbol{\delta}^* \in \mathcal{B}$ from

$$\widetilde{\boldsymbol{X}}^* = \arg \max_{\widetilde{\boldsymbol{X}} \in \mathcal{A}_{\boldsymbol{\delta}}} \ell(f_i(\boldsymbol{A}, \widetilde{\boldsymbol{X}}, \boldsymbol{W}), y_i),$$

where $\widetilde{\boldsymbol{X}}^* = [\tilde{\boldsymbol{x}}_1^*, \ldots, \tilde{\boldsymbol{x}}_n^*]$ and $\tilde{\boldsymbol{x}}_i^* = \boldsymbol{x}_i + \boldsymbol{\delta}^*$. To train a robust model, the goal of adversarial training is to minimize the following adversarial empirical risk

$$\widetilde{R}_m(f) = \frac{1}{m} \sum_{i=1}^m \max_{\widetilde{\boldsymbol{X}} \in \mathcal{A}_{\boldsymbol{\delta}}} \ell(f_i(\boldsymbol{A}, \widetilde{\boldsymbol{X}}, \boldsymbol{W}), y_i),$$

which measures the performance of the predictor $f$ under adversarial attacks defined above. To better understand the robust generalization of the model, we define the adversarial population risk

$$\widetilde{R}_u(f) = \frac{1}{u} \sum_{i=m+1}^{m+u} \max_{\widetilde{\boldsymbol{X}} \in \mathcal{A}_{\boldsymbol{\delta}}} \ell(f_i(\boldsymbol{A}, \widetilde{\boldsymbol{X}}, \boldsymbol{W}), y_i),$$

which measures the ability of $f$ to generalize to unseen adversarial samples. In this paper, we focus on the adversarial generalization gap, which is denoted by

$$Gen(f) = |\widetilde{R}_m(f) - \widetilde{R}_u(f)|.$$

Results from learning theory (Bartlett & Mendelson, 2002) indicate that the generalization gap can be bounded by assessing the complexity of the adversarial loss class. Following previous studies (Bartlett et al., 2017; Mustafa et al., 2022), we provide the definition below

$$\mathcal{L}_{adv} := \{\boldsymbol{z}_i \to \max_{\widetilde{\boldsymbol{X}} \in \mathcal{A}_{\boldsymbol{\delta}}} \ell(f_i(\boldsymbol{A}, \widetilde{\boldsymbol{X}}, \boldsymbol{W}), y_i) : f \in \mathcal{F}\}.$$

Nevertheless, measuring the complexity of $\mathcal{L}_{adv}$ directly is difficult due to the maximum operation on the adversarial loss function. By introducing statistical learning tools, we overcome this problem through the lens of covering number, which is defined as follows.

**Definition 3.1.** (Mustafa et al., 2022) Let $(\mathcal{A}, \mathcal{D})$ be a metric space. Given a positive real number $\epsilon$, we say that $\mathcal{C} \subset \mathcal{A}$ is an $(\epsilon, \mathcal{D})$-cover of $\mathcal{A}$ if

$$\sup_{a \in \mathcal{A}} \inf_{c \in \mathcal{C}} \mathcal{D}(a, c) \leq \epsilon.$$

The covering number of $\mathcal{A}$ is the minimum cardinality of any subset that covers $\mathcal{A}$ at scale $\epsilon$, denoted as $\mathcal{N}(\mathcal{A}, \epsilon)$. Given the dataset $S$, a function class $\mathcal{G}$ taking values in a real vector space, and an $\ell_p$ metric with $p = \infty$, we denote by $\mathcal{N}(\mathcal{G}, \epsilon, \|\cdot\|_\infty, S)$ the $(\epsilon, \|\cdot\|_\infty)$-covering number of the set $\mathcal{G} = \{g(z_1), \ldots, g(z_n) : g \in \mathcal{G}\}$.

## 4. Main Results

In this section, we first investigate the complexity of the adversarial loss function class via covering number and derive a general adversarial generalization bound of GNNs. Then we analyze the adversarial generalization properties of several classical GNNs.

### 4.1. Generalization Analysis Over Adversarial Risk

To establish the generalization guarantee, we first make some mild assumptions that are easy to satisfy.

**Assumption 4.1.** For the model function $f$ and any node $\boldsymbol{x}_i$, assume the following inequality holds for any $\boldsymbol{\delta} \in \mathcal{B}$

$$\|f_i(\boldsymbol{\delta}) - f_i(\boldsymbol{\delta}')\|_\infty \leq K_f \|\boldsymbol{\delta} - \boldsymbol{\delta}'\|_\infty.$$

**Assumption 4.2.** For the loss function $\ell$ and any node $\boldsymbol{x}_i$, assume the following inequality holds for any $f \in \mathcal{F}$

$$|\ell(f_i(\cdot), y_i) - \ell(f_i'(\cdot), y_i)| \leq C_\ell \|f_i(\cdot) - f_i'(\cdot)\|_\infty.$$

*Remark* 4.3. Assumption 4.1 is readily satisfied by a wide range of attacks (Awasthi et al., 2020; Fan et al., 2021), where the Lipschitz constant $K_f$ is determined by specific model architectures. Assumption 4.2 is a mild assumption that can be satisfied by some common losses, such as cross-entropy loss and hinge loss.

Then, we quantify the covering number of the adversarial loss class $\mathcal{L}_{adv}$ in an infinite space by presenting Lemma 4.4 as follows.

**Lemma 4.4.** *Suppose Assumptions 4.1 and 4.2 hold. For any $\boldsymbol{\delta} \in \mathcal{B}$, we define the loss class*

$$\mathcal{L}_{dis} := \{(\boldsymbol{z}_i, \hat{\boldsymbol{\delta}}) \to \ell(f_i(\boldsymbol{A}, \widehat{\boldsymbol{X}}, \boldsymbol{W}), y_i) : f \in \mathcal{F}\}$$

and the extended dataset $\widehat{S} = \{(\boldsymbol{x}_i, \hat{\boldsymbol{\delta}}, y_i) : i \in [n], \hat{\boldsymbol{\delta}} \in \mathcal{C}_\mathcal{B}\}$, where $\mathcal{C}_\mathcal{B}$ is an $(\frac{\epsilon}{2C_\ell K_f}, \|\cdot\|_\infty)$-cover of $\mathcal{B}$ and $\widehat{\boldsymbol{X}} = [\boldsymbol{x}_1 + \hat{\boldsymbol{\delta}}, \ldots, \boldsymbol{x}_n + \hat{\boldsymbol{\delta}}]$. The following inequality holds

$$\mathcal{N}(\mathcal{L}_{adv}, \epsilon, |\cdot|_\infty, S) \leq \mathcal{N}(\mathcal{L}_{dis}, \frac{\epsilon}{2}, |\cdot|_\infty, \widehat{S}).$$

*Remark* 4.5. Lemma 4.4 enables us to control the complexity of the adversarial loss class by the complexity of a constructed loss class, which is a finite discrete version of the former. Notably, we use $|\cdot|$ in the covering number of $\mathcal{L}_{adv}$, as the functions in $\mathcal{L}_{adv}$ involve a maximum operation over $\mathcal{B}$. This operation is eliminated in $\mathcal{L}_{dis}$ by controlling the complexity of the perturbation set, where each element in $\mathcal{L}_{dis}$ is a function.

By utilizing a constructed finite loss class, the difficulty in measuring adversarial generalization caused by maximization over the adversarial loss is solved. However, the interplay between the perturbed nodes introduces a further difficulty in the complexity estimate of $\mathcal{L}_{dis}$ on $\widehat{S}$, which is also the key challenge in measuring adversarial generalization of GNNs, compared with previous works (Farnia & Ozdaglar, 2021; Mustafa et al., 2022). To address this issue, we split the task of estimating the complexity of the perturbed GNNs function class $\widehat{\mathcal{F}}$ on $\widehat{S}$ by utilizing the constructed perturbation cover set $\mathcal{C}_\mathcal{B}$. We begin with fixing the perturbation $\hat{\boldsymbol{\delta}}_c \in \mathcal{C}_\mathcal{B}$, and define the corresponding GNNs function class as follows

$$\widehat{\mathcal{F}} := \{(\boldsymbol{x}_i, \hat{\boldsymbol{\delta}}_c) \to f_i(\boldsymbol{A}, \widehat{\boldsymbol{X}}_c, \boldsymbol{W}) : \hat{\boldsymbol{\delta}}_c \in \mathcal{C}_\mathcal{B}\},$$

where $\widehat{\boldsymbol{X}}_c = [\boldsymbol{x}_1 + \hat{\boldsymbol{\delta}}_c, \ldots, \boldsymbol{x}_n + \hat{\boldsymbol{\delta}}_c]$. Then, we step to the following lemma to bound the covering number of $\mathcal{L}_{dis}$ on $\widehat{S}$.

**Lemma 4.6.** *Given the function classes $\mathcal{L}_{dis}$ and $\widehat{\mathcal{F}}$ be defined above. Suppose Assumptions 4.1 and 4.2 hold. The following inequality holds*

$$\mathcal{N}(\mathcal{L}_{dis}, \frac{\epsilon}{2}, |\cdot|_\infty, \widehat{S}) \leq \left(\frac{6\theta C_\ell K_f}{\epsilon}\right)^d \mathcal{N}(\widehat{\mathcal{F}}, \frac{\epsilon}{2C_\ell}, \|\cdot\|_\infty, S).$$

*Remark* 4.7. Lemma 4.6 overcomes the key difficulty of estimating the covering number of the finite adversarial loss class $\mathcal{L}_{dis}$ on $\widehat{S}$ by additionally incorporating the fixed perturbation $\hat{\boldsymbol{\delta}}_c \in \mathcal{C}_\mathcal{B}$ into analysis. We find that the special mechanism of GNNs induces the interaction between the perturbed nodes, compared with neural networks, thus resulting in an additional term affected by the GNNs architecture and perturbation budget.

Now we can obtain the adversarial generalization gap below.

**Theorem 4.8.** *Suppose Assumptions 4.1 and 4.2 hold. Let $\mathcal{F} : \mathbb{R}^{n \times d} \to \mathbb{R}^n$ be the GNNs function class taking values in $[-q, q]$. Let $Q_1 = \frac{1}{u} + \frac{1}{m}$, $Q_2 =$*

$\frac{m+u}{(m+u-1/2)(10-1/2(max(m,u)))}$ *and $c_0 > 5.05$. For any $f \in \mathcal{F}$, with probability of at least $1 - \delta$, we have*

$$Gen(f) \leq Q_{m,u} + \inf_{\mu > 0} \left(\frac{4\mu}{\sqrt{m+u}} + \frac{24C_\ell}{m+u}\right.$$
$$\left. \int_{\frac{\mu}{2C_\ell}}^{\frac{q\sqrt{m+u}}{C_\ell}} \sqrt{\log\left(\frac{3\theta K_f}{\epsilon}\right)^d \mathcal{N}(\widehat{\mathcal{F}}, \epsilon, \|\cdot\|_\infty, S)} d\epsilon\right),$$

*where $Q_{m,u} = qc_0 Q_1 \sqrt{\min(m, u)} + 2q\sqrt{\frac{Q_1 Q_2}{2} \ln \frac{1}{\delta}}$, and $\mu > 0$ is a constant.*

*Remark* 4.9. Theorem 4.8 establishes a GNN-dependent adversarial generalization bound in the context of transductive inference. The term $Q_{m,u}$ inherently exhibit monotonic decrease at a rate of $\mathcal{O}(\max\{\frac{1}{\sqrt{m}}, \frac{1}{\sqrt{u}}\})$ (Oono & Suzuki, 2020; Esser et al., 2021). Compared with the non-robust generalization gap of GNNs (Tang & Liu, 2023), this bound remains another additional difficulty, i.e., the complexity estimate of the perturbed model function class $\widehat{F}$. Considering the perturbed node features combined with the constructed perturbation $\hat{\boldsymbol{\delta}}_c$, we conduct a comprehensive analysis of $\mathcal{N}(\widehat{\mathcal{F}}, \epsilon, \|\cdot\|_\infty, S)$ for various GNNs, and the explicit characterizations will be discussed in the next section. Besides, we can observe that the perturbation budget $\theta$ significantly impairs the capability of GNNs to generalize, which highlights the importance of analyzing the factors influencing GNNs' adversarial robustness under theoretical guidance.

### 4.2. Adversarial Generalization Gap for GNNs

In this part, for different GNNs, we derive the upper bounds of their covering number $\mathcal{N}(\widehat{\mathcal{F}}, \epsilon, \|\cdot\|_\infty, S)$ and Lipschitz constant $K_f$. Three representative GNNs, including GCN, GCNII, and APPNP, are selected for analysis. To establish the generalization gap, we make some necessary assumptions for $L$-layer GNNs first.

**Assumption 4.10.** For the activation function $\sigma_t(\cdot)$ of layer $t \in [1, L]$ and any vector $\boldsymbol{x}$, assume the following inequality holds

$$\|\sigma_t(\boldsymbol{x}) - \sigma_t(\boldsymbol{x}')\|_\infty \leq \rho_t \|\boldsymbol{x} - \boldsymbol{x}'\|_\infty.$$

**Assumption 4.11.** For any input feature $\boldsymbol{x}_i$, assume that $\|\boldsymbol{x}_i\|_2 \leq b$ holds.

**Assumption 4.12.** For the weight matrix $\boldsymbol{W} = \{\boldsymbol{W}_t\}_{t=0}^{L-1}$, where $\boldsymbol{W}_t \in \mathbb{R}^{d_t \times d_{t+1}}$ ($d_0 = d$ and $d_L = |\mathcal{Y}|$), assume that $\|\boldsymbol{W}_t\|_\infty \leq w_t$ holds.

*Remark* 4.13. Many commonly used activation functions satisfy Assumption 4.10, such as Sigmoid, Tanh, and ELU (Exponential Linear Unit). Assumption 4.11 can be satisfied by applying a normalization operation on the input feature, which is demonstrated to help reduce the generalization gap (Verma & Zhang, 2019; Tang & Liu, 2023). The requirement that learning weights remain bounded during training

in Assumption 4.12 is commonly observed in generalization analysis (Garg et al., 2020; Cong et al., 2021).

Next, we analyze the upper bounds of the generalization error of the following model, respectively.

**GCN.** Kipf and Welling (2016) propose an efficient layer-wise propagation rule to learn node-level representations, which encodes the node features in a way useful for semi-supervised classification learning. The output function of a $L$-layer GCN model is

$$
\begin{aligned}
&f(\boldsymbol{A}, \boldsymbol{X}, \boldsymbol{W}) \\
&= \sigma_L(g(\boldsymbol{A})\sigma_{L-1}(g(\boldsymbol{A})\dots\sigma_1(g(\boldsymbol{A})\boldsymbol{X}\boldsymbol{W}_1)\boldsymbol{W}_2)\dots\boldsymbol{W}_L).
\end{aligned}
$$

where $g(\boldsymbol{A}) \in \mathbb{R}^{n \times n}$ is the graph filter. Three types of filters are introduced in this paper, such as the unnormalized filter with self-loops $g(\boldsymbol{A}) = \boldsymbol{A} + \boldsymbol{I}$ (Xu et al., 2018), the symmetric normalized filter $g(\boldsymbol{A}) = (\boldsymbol{D} + \boldsymbol{I})^{-1/2}(\boldsymbol{A} + \boldsymbol{I})(\boldsymbol{D} + \boldsymbol{I})^{-1/2}$ (Kipf & Welling, 2016), and the random walk filter $g(\boldsymbol{A}) = (\boldsymbol{D} + \boldsymbol{I})^{-1}(\boldsymbol{A} + \boldsymbol{I})$ (Zhang et al., 2019). $\boldsymbol{D} \in \mathbb{R}^{n \times n}$ is the degree matrix of graph $G$, where $\boldsymbol{D}_{ii} = \sum_{j=1}^{n} \boldsymbol{A}_{ij}$.

**Proposition 4.14.** *With probability of at least $1 - \delta$, the following inequality holds*

$$
\begin{aligned}
Gen(f) \leq & Q_{m,u} + \frac{4}{m+u} \\
&+ \frac{72C_\ell}{\sqrt{m+u}}\Big(\sqrt{d}\theta K_{GCN} + v\hat{b}g^L \prod_{t=1}^{L} \rho_t w_t\Big),
\end{aligned}
$$

*where $g = \|g(\boldsymbol{A})\|_\infty$, $K_{GCN} = C_\ell g^L \prod_{t=1}^{L} \rho_t w_t$, $\hat{b} = b + \sqrt{d}\theta$, and $v = \max\{d_1, \dots, d_{L+1}\}$. $Q_{m,u}$ is as stated in Theorem 4.8.*

*Remark* 4.15. It's clear that the perturbation budget $\theta$ in the first term $\sqrt{d}\theta K_{\mathrm{GCN}}$ would deteriorate the generalization performance of GCN, where a higher dimensional feature $d$ could exacerbate this adverse effect. Thus, a smaller perturbation budget $\theta$ could reduce the dependence of adversarial generalization on $d$. Moreover, for the unnormalized filter, for a fixed $i \in [n]$ and any $j \in [n]$, we have $\|g(\boldsymbol{A})\|_\infty \leq d_{max} + 1$, where $d_{max}$ denotes the maximum degrees. For the symmetric normalized filter and random walk filter, $\|g(\boldsymbol{A})\|_\infty \leq \sqrt{(d_{max} + 1)/(d_{min} + 1)}$, where $d_{min}$ denotes the minimum degrees. This motivates applying normalized graph filters with an appropriate number of layers to achieve the generalization ability, which is consistent with the empirical findings (Kipf & Welling, 2016; Li et al., 2018). Furthermore, the Lipschitz constant $\rho_t = 1$ is usually satisfied while the activation function is selected properly. Thus, we can mitigate the performance loss caused by the product of weight constraints by applying regularization to the weights.

**APPNP.** Considering the limitations of the propagation procedure for node classification, Gasteiger et al. (2018) derive an improved propagation scheme based on personalized PageRank. This approach constructs a simple model that utilizes a large and adjustable neighborhood for node classification, which avoids the poor performance of GCN caused by more aggregation steps or more layers. The output function of a $L$-layer APPNP model is

$$
\begin{aligned}
&f(\boldsymbol{A}, \boldsymbol{X}, \boldsymbol{W}) \\
&= \sigma_L(\hat{g}(\boldsymbol{A})\sigma_{L-1}(\boldsymbol{W}_{L-1}\sigma_{L-2}(\boldsymbol{W}_{L-2}\dots\sigma_1(\boldsymbol{W}_1\boldsymbol{X})))),
\end{aligned}
$$

where $\hat{g}(\boldsymbol{A}) = \sum_{k=0}^{K-1} \gamma(1-\gamma)^k g(\boldsymbol{A})^k + (1-\gamma)^K g(\boldsymbol{A})^K$. $K$ is the aggregation hop and $\gamma$ is a probability, which is designed to adjust the size of the neighborhood influencing each node.

**Proposition 4.16.** *With probability of at least $1 - \delta$, the following inequality holds*

$$
\begin{aligned}
Gen(f) \leq & Q_{m,u} + \frac{4}{m+u} \\
&+ \frac{72C_\ell}{\sqrt{m+u}}\Big(\sqrt{d}\theta K_{APPNP} + v\hat{b}\hat{g}\rho_L \prod_{t=1}^{L-1} \rho_t w_t\Big),
\end{aligned}
$$

*where $\hat{g} = \|\hat{g}(\boldsymbol{A})\|_\infty \leq \gamma\big(1 + \sum_{k=1}^{K-1} ((1-\gamma)g)^k\big) + ((1-\gamma)g)^K$, and $K_{APPNP} = C_\ell \hat{g}\rho_L \prod_{t=1}^{L-1} \rho_t w_t$. $Q_{m,u}$ is as stated in Theorem 4.8. The definitions of $\hat{b}$, $g$ and $v$ are the same as in Proposition 4.14.*

*Remark* 4.17. Compared to the generalization gap of GCN, the main difference of the bound in Proposition 4.16 lies in the treatment of graph filters. We find that the probability $\gamma \in (0,1)$ is set as a small number, and the graph filter $g(\boldsymbol{A})$ is symmetric normalized, which yields that $\hat{g} \leq g \leq g^L$ (Gasteiger et al., 2018). Therefore, APPNP could achieve a better generalization performance than GCN. Moreover, as a higher probability $\gamma$ could improve the convergence speed and benefit the generalization, such a large size of the neighborhood would lead to a rapid performance degradation (Gasteiger et al., 2018). The neighborhood structures vary across different types of graphs (Grover & Leskovec, 2016; Abu-El-Haija et al., 2020), which has a significant impact on the selection of the parameter $\gamma$. Hence, it is crucial to carefully adjust the parameters $\gamma$ and $K$ to guarantee a trade-off between the generalization behavior and representation ability.

**GCNII.** Although combing the shallow graph neural networks and deep propagation, Chen et al. (2020b) consider APPNP losing the powerful expression ability of deep non-linear architectures. GCNII is proposed to alleviate the over-smoothing in the deep graph model by using two simple techniques, initial residual connection and identity mapping, which enables GCN to express the high-order polynomial filter with arbitrary coefficients. For $l = [1, \dots, L-1]$, the propagation process is

$$
H^l = \sigma_l\Big[\big((1-\alpha)g(\boldsymbol{A})H^{l-1} + \alpha H^0\big)\psi(\boldsymbol{W}_l)\Big],
$$

Table 2. Details of the adopted datasets.

| Dataset | Nodes | Edges | Features | Classes | Training | Validation | Test |
|---------|-------|-------|----------|---------|----------|------------|------|
| Citeseer | 3327 | 9104 | 3703 | 6 | 20 per class | 500 | 1000 |
| Cora | 2708 | 10556 | 1433 | 77 | 20 per class | 500 | 1000 |
| Pubmed | 19717 | 88648 | 500 | 3 | 20 per class | 500 | 1000 |
| DBLP | 17716 | 105734 | 1639 | 4 | 20 per class | 30 per class | Rest |
| CS | 18333 | 163788 | 6805 | 15 | 20 per class | 30 per class | Rest |
| CoraFull | 19793 | 126842 | 8710 | 70 | 20 per class | 30 per class | Rest |

where $H^0 = \sigma_0(\boldsymbol{X}\boldsymbol{W}_0)$, $\psi(\boldsymbol{W}_l) = \left((1-\beta)\boldsymbol{I}_n + \beta\boldsymbol{W}_l\right)$, and $\alpha, \beta \in (0, 1)$. The output function of a $L$-layer GCNII model is $f(\boldsymbol{A}, \boldsymbol{X}, \boldsymbol{W}) = \sigma_L(H^{L-1}\boldsymbol{W}_L)$.

**Proposition 4.18.** *With probability of at least $1 - \delta$, the following inequality holds*

$$Gen(f) \leq Q_{m,u} + \frac{4}{m+u}$$

$$+ \frac{72C_\ell}{\sqrt{m+u}}\left(\sqrt{d}\theta K_{GCNII} + v\hat{b}\beta \sum_{j=0}^{L} T_j \prod_{t=j}^{L} \rho_t \tilde{w}_t\right),$$

*where $\tilde{w}_t \leq 1 - \beta + \beta w_t$, and*

$$K_{GCNII} = C_\ell \rho_0 \tilde{w}_0 \sum_{j=1}^{L}((1-\alpha)g)^{j-1} \prod_{t=L-j}^{L} \rho_t \tilde{w}_t,$$

$$T_j = 2\sum_{j=0}^{L}((1-\alpha)g)^{L-j}\left((1-\alpha)gA_{j-1} + \alpha A_0\right),$$

$$A_p = \hat{b}\rho_0 \tilde{w}_0 \sum_{j=0}^{p}((1-\alpha)g)^j \prod_{t=p-j}^{p} \rho_t \tilde{w}_t.$$

*$Q_{m,u}$ is as stated in Theorem 4.8. The definitions of $\hat{b}$, $g$ and $v$ are the same as in Proposition 4.14.*

*Remark* 4.19. According to Proposition 4.18, the performance of GCNII is mainly related to the choice of $\alpha$ and $\beta$. $\alpha$ is usually set as a small number to avoid the performance drop if we stack many layers, as the final representation of each node retains at least a fraction of $\alpha$ from the input feature (Chen et al., 2020b). However, it is noteworthy that the impact of $\alpha$ and $L$ on generalization is closely relevant to $\|g(\boldsymbol{A})\|_\infty$. A deep model with a smaller $\alpha$ and an unnormalized filter would lead to worse generalization. Moreover, $\beta$ ensures at least the same performance between a deep model and its shallow version (He et al., 2016). By setting $\beta$ relatively small and imposing regularization on the weight can obtain a small weight norm, which is particularly useful to avoid overfitting as well as enhance the generalization.

## 5. Experiments

In this section, we propose an adversarial training algorithm to learn robust GNNs based on our theoretical findings, and

---

**Algorithm 1** Train a robust graph model
---
1: **Input:** Graph $G$, dataset $S$, perturbation budget $\theta$, regularization parameter $\lambda$, initialization $\boldsymbol{W}_0$, learning rate $\eta$, number of iterations $T$.
2: **while** $t < T$ **do**
3:     $\widetilde{S} \leftarrow \emptyset$.
4:     **for** $i = 1, 2, \ldots, n$ **do**
5:         For the input matrix $X_t = [x_{1,t}, \ldots, x_{n,t}]$, perturb $\tilde{X}_t \leftarrow X_t + \mathcal{A}(X_t, A, \theta)$.
6:         For each node in $\tilde{X}_t = [\tilde{x}_{1,t}, \ldots, \tilde{x}_{n,t}]$, append $\{(\tilde{\boldsymbol{x}}_{i,t}, y_{i,t})\}_{i=1}^n$ to $\widetilde{S}_t$ and choose $m$ samples randomly to the training set $\widetilde{S}_{m,t}$.
7:     **end for**
8:     Define a new objective $L(W_{i,t}) = \frac{1}{m}\sum_{X_{i,t}\in\tilde{S}_{m,t}} \ell(f_{i,t}(A, X, W), y_{i,t}) + \lambda\|W_{i,t}\|_\infty$.
9:     For all $i \in [m]$, update $W_t$ using SGD: $W_{i,t+1} \leftarrow W_{i,t} - \eta\nabla L(W_{i,t})$.
10: **end while**
11: **Output:** $\boldsymbol{W}_T$

---

validate our theoretical results by evaluating the effect of several factors.

**Experimental Setup.** We adopt six benchmark datasets provided by PyTorch Geometric, including Citeseer, Cora, Pubmed, DBLP, CS, and CoraFull (see Table 2 for more details). We evaluate the performance of three popular GNN models: GCN, GCNII, and APPNP. Let $\mathcal{A}$ be a gradient-based attack algorithm (e.g., PGD, BIM, Mettack). We present our robust learning method in Algorithm 1. During the training procedure, adversarial examples are generated by PGD algorithm (Bottou, 2010) with a step size of $\theta/128$, where $\theta$ is the perturbation budget. We set training iterations $T$ as 200 and use cross-entropy loss for training. SGD is adopted for optimization with its learning rate $\eta$ set by 0.05 and a weight decay of 1e-3. The regularization parameter $\lambda$ is fixed to 0.1. The generalization gap is approximated by the following accuracy gap based on adversarial training

$$|\text{adversarial\_train\_accuracy} - \text{adversarial\_test\_accuracy}|,$$

which is the absolute difference between the accuracy on

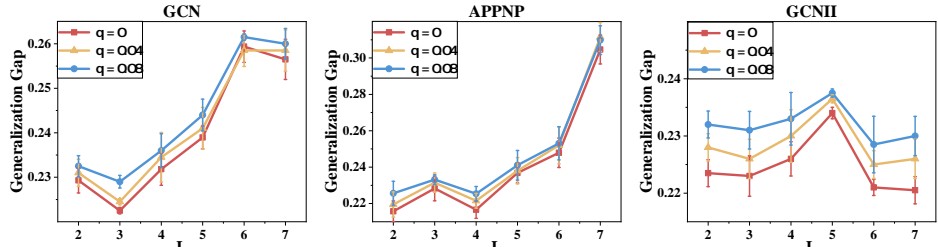

*Figure 1.* The generalization gap for different adversarial perturbations $\theta$ with increased number of layers $L$ (on Cora).

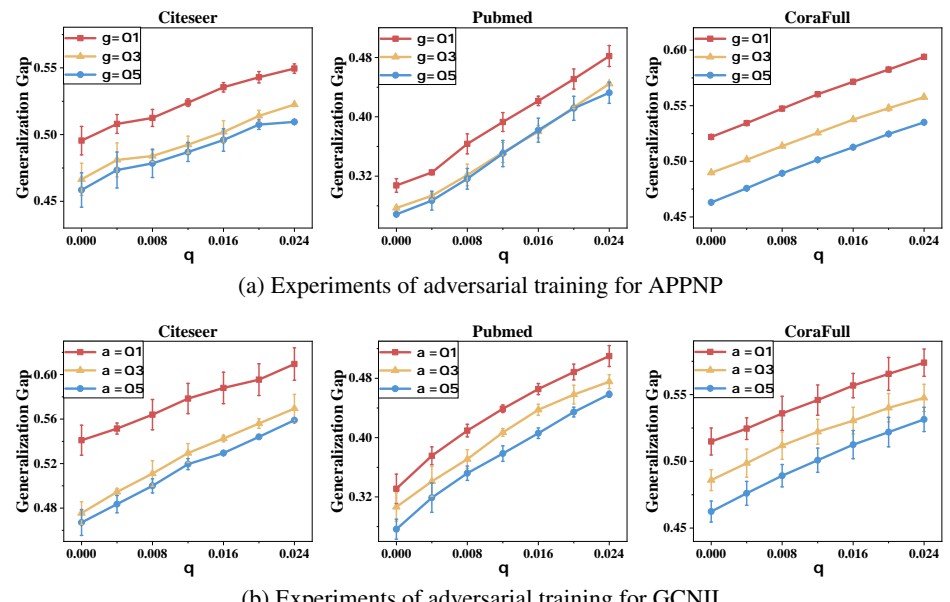

(a) Experiments of adversarial training for APPNP

(b) Experiments of adversarial training for GCNII

*Figure 2.* The generalization gap for different hyper-parameter $\gamma$ in APPNP and $\alpha$ in GCNII with increased perturbations $\theta$.

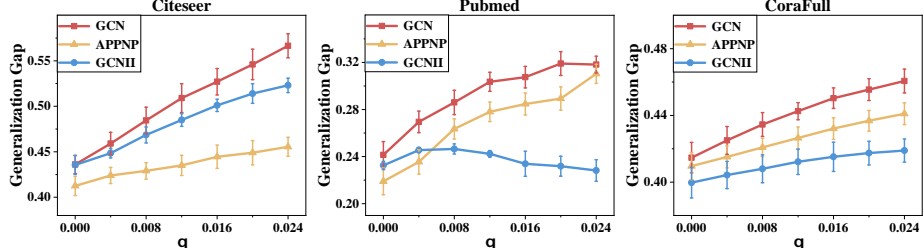

*Figure 3.* The generalization gap for different GNN models with increased perturbations $\theta$.

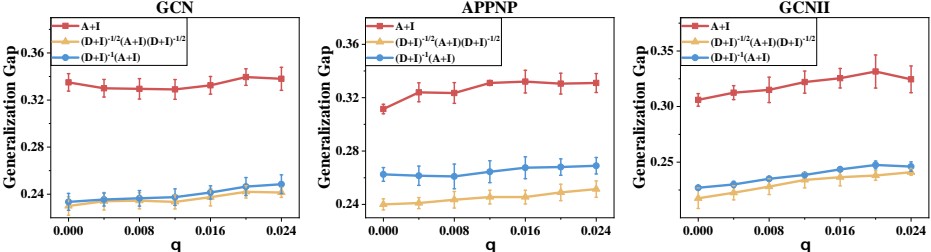

*Figure 4.* The generalization gap for different graph filters $g(\boldsymbol{A})$ with increased perturbations $\theta$ (on Cora).

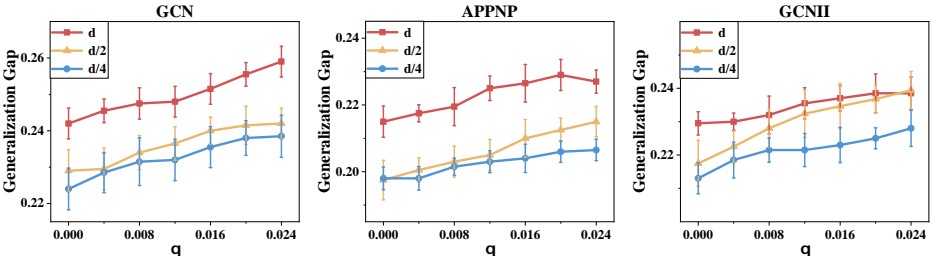

*Figure 5.* The generalization gap for different input feature dimension $d$ with increased perturbations $\theta$ (on Cora).

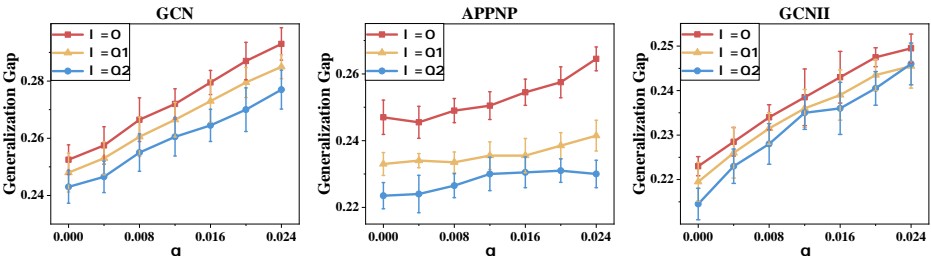

*Figure 6.* The generalization gap for different regularization parameter $\lambda$ with increased perturbations $\theta$ (on Cora).

training and test data.

We repeat each experiment 10 times and report the mean value and standard deviation for output. For more detailed model configuration and experimental results (including results of other datasets, other attack methods, and other influencing factors assessment), please refer to Appendix F.

**Number of layers.** For GCN, Figure 1 reveals that the gap increases as the model gets deeper, which aligns with our theoretical results. And affected by the superposition of the increased nonlinear operations, APPNP performs worse with the higher model depth. However, as stated in Proposition 4.18, the parameter $\alpha$ could adjust the proportion of parameters in deeper layers. The generalization gap of GC-NII presents a slow decreasing trend, which demonstrates its superior performance in a deep version.

**Model architecture.** According to Proposition 4.16, a relatively bigger $\gamma$ could help reduce the adversarial generalization gap, especially when $\|g(\boldsymbol{A})\|_\infty$ of the unnormalized filter is much bigger than 1. Similarly, as stated in Proposition 4.18, a little bigger $\alpha$ benefits the adversarial generalization in a deep GCNII model. Figure 2 shows the results of different hyper-parameters $\gamma$ in APPNP and $\alpha$ in GCNII, which exhibit a similar trend. Moreover, Figure 3 shows that GCNII and APPNP have a smaller generalization gap than GCN, which demonstrates the effectiveness of the improved model architectures.

**Graph filter.** The theoretical results demonstrate that the unnormalized filter hurts the adversarial generalization due to the exponential dependence on model depth. As is shown in Figure 4, the two normalized filters $(\boldsymbol{D} + \boldsymbol{I})^{-1/2}(\boldsymbol{A} +$

$\boldsymbol{I})(\boldsymbol{D} + \boldsymbol{I})^{-1/2}$ and $(\boldsymbol{D} + \boldsymbol{I})^{-1}(\boldsymbol{A} + \boldsymbol{I})$ own better generalization performance than the unnormalized filter $\boldsymbol{A} + \boldsymbol{I}$.

**Feature dimension.** As shown in Figure 5, it is clear that the generalization error with a lower input feature has a better generalization performance, which is consistent with our theoretical analysis as stated in Proposition 1.

**Regularization parameter.** Our theoretical results show that the product of the norm weights deteriorates the generalization, especially when we stack more layers. Figure 6 demonstrates that norm regularization facilitates the adversarial generalization of GNNs.

## 6. Conclusion

In this paper, we establish an adversarial generalization bound of GNNs in the context of transductive learning, providing theoretical support to the empirical advancements of adversarial training. Our approach is modular and easily applicable to a wide range of GNN models. We further showcase our results of generalization analysis on three representative cases (GCN, APPNP, and GCNII), which are validated in our experimental results.

In future work, we consider extending our result to an optimistic fast-rate bound for a smooth assumption of the loss. As the graph topology is vulnerable to adversarial attacks (Li et al., 2022), adversarial generalization analysis against structural perturbations will be our future exploration. The detailed discussions about the extension of our work to topology attacks are included in Appendix E.

## Impact Statement

This paper presents work whose goal is to advance the field of Machine Learning. There are many potential societal consequences of our work, none which we feel must be specifically highlighted here.

## Acknowledgments

This work was supported by the National Natural Science Foundation of China (under Grant Nos. 62276111, 62076041, 62376104, 12426512), and the Open Research Fund of Engineering Research Center of Intelligent Technology for Agriculture, Ministry of Education (No. ERCITA-KF002).

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

## A. Notations

The main notations of this paper are summarized in Table 3.

*Table 3.* Summary of main notations involved in this paper

| Notations | Descriptions |
|---|---|
| $\boldsymbol{A}$ | The adjacency matrix, where $\boldsymbol{A} \in \mathbb{R}^{n \times n}$ |
| $\boldsymbol{X}$ | The node (input) feature matrix, where $\boldsymbol{X} \in \mathbb{R}^{n \times d}$ |
| $\boldsymbol{D}$ | The degree matrix with $D_{i,i} = \sum_j A_{i,j}$ |
| $g(\boldsymbol{A})$ | The graph filter of $G$, where $g(\boldsymbol{A}) \in \mathbb{R}^{n \times n}$ |
| $\boldsymbol{W}$ | The learning parameters of GNNs, where $\boldsymbol{W} = \{\boldsymbol{W}_t\}_{t=1}^L$ |
| $f(\boldsymbol{A}, \boldsymbol{X}, \boldsymbol{W})$ | The output function of GNNs |
| $\boldsymbol{x}_i, y_i$ | The node feature and node label |
| $\ell(f_i(\boldsymbol{A}, \boldsymbol{X}, \boldsymbol{W}), y_i)$ | The loss function |
| $\boldsymbol{\delta}, \theta$ | The perturbations and perturbation budget |
| $\mathcal{B}$ | The noise space $\{\boldsymbol{\delta} : \|\boldsymbol{\delta}\|_\infty \leq \theta\}$ |
| $\mathcal{A}_{\boldsymbol{\delta}}$ | The set of adversarial nodes $\{\widetilde{\boldsymbol{X}} = [\tilde{\boldsymbol{x}}_1, \dots, \tilde{\boldsymbol{x}}_n], \tilde{\boldsymbol{x}}_i = \boldsymbol{x}_i + \boldsymbol{\delta}, i \in [n]\}$ |
| $\max_{\widetilde{\boldsymbol{X}} \in \mathcal{A}_{\boldsymbol{\delta}}} \ell(f_i(\boldsymbol{A}, \widetilde{\boldsymbol{X}}, \boldsymbol{W}), y_i)$ | The adversarial loss function |
| $S, \widehat{S}$ | The training dataset and the extended training dataset with the perturbations |
| $\widetilde{R}_m(f), \widetilde{R}_u(f)$ | The adversarial empirical (population) risk of the adversarial loss function |
| $Gen(f)$ | The generalization error in the adversarial settings |
| $\mathcal{C}_{\mathcal{B}}$ | The cover of the noise set $\mathcal{B}$ |
| $\hat{\boldsymbol{\delta}}, \hat{\boldsymbol{\delta}}_c$ | An arbitrary and fixed perturbations in the cover set $\mathcal{C}_{\mathcal{B}}$ |
| $\widehat{\mathcal{F}}$ | The model function class with the fixed perturbation $\hat{\boldsymbol{\delta}}_c$ |
| $\mathcal{N}(\widehat{\mathcal{F}}, \epsilon, \|\cdot\|_\infty, S)$ | The covering number of perturbed function class $\widehat{\mathcal{F}}$ at scale $\epsilon$ with an $\ell_\infty$ metric |
| $b, \hat{b}$ | The norm constraints of $\|\boldsymbol{x}\|_2$ and $\|\tilde{\boldsymbol{x}}\|_2$ |
| $C_f, K_f$, and $\rho_t$ | The Lipschitz constants of the loss function, output function, and activation function |
| $w_t$ | the norm constraint of the $t$-th layer's weight $\boldsymbol{W}_t$ |
| $\|\boldsymbol{x}\|_2 = \sqrt{\sum_j \|x_j\|^2}$ | The $\ell_2$ norm of a vector $\boldsymbol{x}$ |
| $\|\boldsymbol{x}\|_\infty = \max \sum_j \|x_j\|$ | The $\ell_\infty$ norm of a vector $\boldsymbol{x}$ |
| $\|\boldsymbol{X}\|_\infty = \max \sum_j \|X_{ij}\|$ | The infinity norm of a matrix $\boldsymbol{X}$ |

## B. Proof of Main Results

### B.1. Generalization for GNNs

As discussed in the main test, the main challenges in deriving the adversarial generalization bound for GNNs (Theorem 4.8) can be handled by Lemma 4.4 and Lemma 4.6. Followed by Mustafa, Lei, and Kloft (2022), we first present the proofs of Lemma 4.4, which controls the covering number of the adversarial loss class of GNNs through a covered perturbation class.

*Proof of Lemma 4.4.* For each function $f \in \mathcal{F}$ and a fixed $\boldsymbol{\delta}_c \in \mathcal{B}$, we construct a new function $h : \mathcal{Z} \to (\mathbb{R}^n)^{\mathcal{B}}$ as $h(\boldsymbol{z}_i, \boldsymbol{\delta}_c) = \ell(f_i(\boldsymbol{A}, \widetilde{\boldsymbol{X}}_c, \boldsymbol{W}), y_i)$, where $\widetilde{\boldsymbol{X}}_c = [\boldsymbol{x}_1 + \boldsymbol{\delta}_c, \dots, \boldsymbol{x}_n + \boldsymbol{\delta}_c]$. The corresponding function class is

$$\mathcal{L} := \{\boldsymbol{z}_i \to \ell(f_i(\boldsymbol{A}, \widetilde{\boldsymbol{X}}_c, \boldsymbol{W}), y_i) : f \in \mathcal{F}\}. \tag{1}$$

Recalling the adversarial loss class $\mathcal{L}_{adv}$, let $\max_{\boldsymbol{\delta} \in \mathcal{B}} h(\boldsymbol{z}_i, \boldsymbol{\delta}) = \max_{\widetilde{\boldsymbol{X}} \in \mathcal{A}_{\boldsymbol{\delta}}} \ell(f_i(\boldsymbol{A}, \widetilde{\boldsymbol{X}}, \boldsymbol{W}), y_i)$ for any $\boldsymbol{\delta} \in \mathcal{B}$. In this

part, we use the cover elements of the constructed class $\mathcal{L}$ on dataset $S$ to build a cover for the adversarial loss function class $\mathcal{L}_{adv}$ on the same dataset $S$. Recall from the definition that $\mathcal{N}(\mathcal{L}_{adv}, \epsilon, |\cdot|_{\infty}, S)$ is the cover for

$$\mathcal{L}_{adv}|_S := \left\{ \left( \max_{\boldsymbol{\delta} \in \mathcal{B}} h(\boldsymbol{z}_1, \boldsymbol{\delta}), \dots, \max_{\boldsymbol{\delta} \in \mathcal{B}} h(\boldsymbol{z}_n, \boldsymbol{\delta}) \right) \right\} \subset \mathbb{R}^n.$$

First we construct an $\ell_{\infty}-$cover for $\mathcal{L}_{adv}|_S$ by utilizing an $\ell_{\infty}-$cover for

$$\mathcal{L}|_S := \left\{ \left( h(\boldsymbol{z}_1, \boldsymbol{\delta}_c), \dots, h(\boldsymbol{z}_n, \boldsymbol{\delta}_c) \right) \right\} \subset (\mathbb{R}^{\mathcal{B}})^n,$$

which is $\mathcal{N}(\mathcal{L}, \epsilon, \|\cdot\|_{\infty}, S)$ in definition. The cover of $\mathcal{L}$ of size $N_{\mathcal{L}}$ is denoted by

$$\mathcal{C}_{\mathcal{L}} := \left\{ \left( c_j^1(\boldsymbol{\delta}_c), \dots, c_j^n(\boldsymbol{\delta}_c) \right) : j \in [N_{\mathcal{L}}] \right\} \subset (\mathbb{R}^{\mathcal{B}})^n.$$

Then we can obtain the cover of $\mathcal{L}_{adv}|_S$ as

$$\mathcal{C}_{\mathcal{L}_{adv}} := \left\{ \left( \tilde{c}_j^1 := \max_{\boldsymbol{\delta} \in \mathcal{B}} c_j^1(\boldsymbol{\delta}), \dots, \tilde{c}_j^n := \max_{\boldsymbol{\delta} \in \mathcal{B}} c_j^n(\boldsymbol{\delta}) \right) : j \in [N_{\mathcal{L}}] \right\} \subset \mathbb{R}^n.$$

We have

$$\max_{i \in [n]} |\max_{\boldsymbol{\delta} \in \mathcal{B}} h(\boldsymbol{z}_i, \boldsymbol{\delta}) - \tilde{c}_j^i| = \max_{i \in [n]} |\max_{\boldsymbol{\delta} \in \mathcal{B}} h(\boldsymbol{z}_i, \boldsymbol{\delta}) - \max_{\boldsymbol{\delta} \in \mathcal{B}} c_j^i(\boldsymbol{\delta})| \le \max_{i \in [n]} \max_{\boldsymbol{\delta} \in \mathcal{B}} |h(\boldsymbol{z}_i, \boldsymbol{\delta}) - c_j^i(\boldsymbol{\delta})| \le \epsilon,$$

where the first equality is based on the construction of $\mathcal{C}_{\mathcal{L}_{adv}}$, the first inequality follows from the inequality $|\max_{\boldsymbol{x}} f(\boldsymbol{x}) - \max_{\boldsymbol{x}} g(\boldsymbol{x})| \le \max_{\boldsymbol{x}} |f(\boldsymbol{x}) - g(\boldsymbol{x})|$, and the last inequality is based on the covering number of $\mathcal{L}$. Lastly, we have

$$\mathcal{N}(\mathcal{L}_{adv}, \epsilon, |\cdot|_{\infty}, S) \le \mathcal{N}(\mathcal{L}, \epsilon, \|\cdot\|_{\infty}, S).$$

Now we control the complexity of the adversarial loss class. However, there is still a difficulty in deriving the upper bound of $\mathcal{N}_{\infty}(\mathcal{L}, \epsilon, S)$, as the functions in $\mathcal{L}$ take values in an infinite-dimensional vector space. We approximate the space by a finite discretization version to solve this problem, where the function class of the discrete version is defined by

$$\mathcal{L}_{dis} := \{(\boldsymbol{z}_i, \hat{\boldsymbol{\delta}}) \to \ell(f_i(\boldsymbol{A}, \widehat{\boldsymbol{X}}, \boldsymbol{W}), y_i) : f \in \mathcal{F}\}. \tag{2}$$

In this part, the target is to control the covering number of the infinite-dimensional class $\mathcal{L}$ with the finite-dimensional counterpart $\mathcal{L}_{dis}$. To approximate the functions $h$ with a discrete form, we construct an $\epsilon/2C_{\ell}K_f-$cover for set $\mathcal{B}$

$$\mathcal{C}_{\mathcal{B}} := \{\hat{\boldsymbol{\delta}}_j, j \in [N_{\mathcal{B}}]\} \subset \mathcal{B},$$

where $N_{\mathcal{B}}$ is the size of the cover $\mathcal{C}_{\mathcal{B}}$. Let the function class $\mathcal{L}$ and $\mathcal{L}_{dis}$ be defined in Equation (1) and (2) respectively. Let the extended dataset be expressed by $\widehat{S} = \{(\boldsymbol{x}_i, \hat{\boldsymbol{\delta}}, y_i) : i \in [n], \hat{\boldsymbol{\delta}} \in \mathcal{C}_{\mathcal{B}}\}$. Our goal is to construct an $\epsilon$-cover of $\mathcal{L}$ by utilizing an $\epsilon/2$-cover of $\mathcal{L}_{dis}$. According to the set $\mathcal{C}_{\mathcal{B}}$, we have

$$\mathcal{L}_{dis}|_{\widehat{S}} := \left\{ \left( h(\boldsymbol{z}_1, \hat{\boldsymbol{\delta}}_1), \dots, h(\boldsymbol{z}_1, \hat{\boldsymbol{\delta}}_{N_{\mathcal{B}}}), \dots, h(\boldsymbol{z}_n, \hat{\boldsymbol{\delta}}_1), \dots, h(\boldsymbol{z}_n, \hat{\boldsymbol{\delta}}_{N_{\mathcal{B}}}) \right) \right\} \subset \mathbb{R}^{n \times N_{\mathcal{B}}}.$$

$\mathcal{N}(\mathcal{L}_{dis}, \epsilon, |\cdot|_{\infty}, \widehat{S})$ is defined as an $\epsilon/2$-cover for $\mathcal{L}_{dis}|_{\widehat{S}}$ of size $N_{\mathcal{L}_{dis}}$, which is denoted by

$$\mathcal{C}_{\mathcal{L}_{dis}} := \left\{ \left( \hat{c}_j^1(\hat{\boldsymbol{\delta}}_1) \dots \hat{c}_j^1(\hat{\boldsymbol{\delta}}_{N_{\mathcal{B}}}), \dots, \hat{c}_j^n(\hat{\boldsymbol{\delta}}_1), \dots, \hat{c}_j^n(\hat{\boldsymbol{\delta}}_{N_{\mathcal{B}}}) \right) : j \in [N_{\mathcal{L}_{dis}}] \right\} \subset \mathbb{R}^{n \times N_{\mathcal{B}}}.$$

Next, we construct a cover of $\mathcal{L}|_S$. The key idea is to construct functions $c_j(\hat{\boldsymbol{\delta}})$, where $j \in [N_{\mathcal{L}_{dis}}]$ is a piece-wise constant around each $\hat{\boldsymbol{\delta}} \in \mathcal{C}_{\mathcal{B}}$. Let

$$\mathcal{C}_{\mathcal{L}} := \left\{ \left( c_j^1(\boldsymbol{\delta}) := \hat{c}_j^1(\arg\min_{\hat{\boldsymbol{\delta}} \in \mathcal{C}_{\mathcal{B}}} \|\boldsymbol{\delta} - \hat{\boldsymbol{\delta}}\|), \dots, c_j^n(\boldsymbol{\delta}) := \hat{c}_j^n(\arg\min_{\hat{\boldsymbol{\delta}} \in \mathcal{C}_{\mathcal{B}}} \|\boldsymbol{\delta} - \hat{\boldsymbol{\delta}}\|) \right) : j \in [N_{\mathcal{L}_{dis}}] \right\} \in (\mathbb{R}^{\mathcal{B}})^n$$

be an $\epsilon$-cover of $\mathcal{L}$. Therefore, we have

$$\max_i \max_{\boldsymbol{\delta} \in \mathcal{B}} |h(\boldsymbol{z}_i, \boldsymbol{\delta}) - c_j^i(\boldsymbol{\delta})|$$

$$= \max_i \max_{\boldsymbol{\delta} \in \mathcal{B}} |h(\boldsymbol{z}_i, \boldsymbol{\delta}) - h(\boldsymbol{z}_i, \boldsymbol{\delta}^*) + h(\boldsymbol{z}_i, \boldsymbol{\delta}^*) - c_j^i(\hat{\boldsymbol{\delta}})|$$

$$\leq \max_i \max_{\boldsymbol{\delta} \in \mathcal{B}} \left( |h(\boldsymbol{z}_i, \boldsymbol{\delta}) - h(\boldsymbol{z}_i, \boldsymbol{\delta}^*)| + |h(\boldsymbol{z}_i, \boldsymbol{\delta}^*) - c_j^i(\boldsymbol{\delta})| \right)$$

$$\leq \max_i \max_{\boldsymbol{\delta} \in \mathcal{B}} |h(\boldsymbol{z}_i, \boldsymbol{\delta}) - h(\boldsymbol{z}_i, \boldsymbol{\delta}^*)| + \max_i \max_{\hat{\boldsymbol{\delta}} \in \mathcal{C}_\mathcal{B}} |h(\boldsymbol{z}_i, \hat{\boldsymbol{\delta}}) - \hat{c}_j^i(\boldsymbol{\delta}^*)|$$

$$\leq \max_i \max_{\boldsymbol{\delta} \in \mathcal{B}} C_\ell K_f \|\boldsymbol{\delta} - \boldsymbol{\delta}^*\| + \max_i \max_{\hat{\boldsymbol{\delta}} \in \mathcal{C}_\mathcal{B}} |h(\boldsymbol{z}_i, \hat{\boldsymbol{\delta}}) - \hat{c}_j^i(\hat{\boldsymbol{\delta}})|$$

$$\leq C_\ell K_f \epsilon / 2 C_\ell K_f + \epsilon / 2 = \epsilon,$$

where $\boldsymbol{\delta}^* = \arg\min_{\hat{\boldsymbol{\delta}} \in \mathcal{C}_\mathcal{B}} \|\boldsymbol{\delta} - \hat{\boldsymbol{\delta}}\|$. The first inequality follows from triangle inequality. The second inequality is followed by the construction of $\mathcal{C}_\mathcal{B}$, such that $c_j^i(\boldsymbol{\delta}) = \hat{c}_j^i(\boldsymbol{\delta}^*)$. The third inequality is based on the Lipschitz properties of the loss function with respect to $\boldsymbol{\delta}$. The last inequality is based on the construction of $\boldsymbol{\delta}^*$ and $\mathcal{C}_{\mathcal{L}_{dis}}$. Then we can obtain

$$\mathcal{N}(\mathcal{L}, \epsilon, \|\cdot\|_\infty, S) \leq \mathcal{N}(\mathcal{L}_{dis}, \epsilon/2, |\cdot|_\infty, \widehat{S}),$$

which concludes that

$$\mathcal{N}(\mathcal{L}_{adv}, \epsilon, |\cdot|_\infty, S) \leq \mathcal{N}(\mathcal{L}, \epsilon, \|\cdot\|_\infty, S) \leq \mathcal{N}(\mathcal{L}_{dis}, \epsilon/2, |\cdot|_\infty, \widehat{S}).$$

Now we finish the proofs of Lemma 4.4. $\qquad\square$

Lemma 4.4 utilizes the construction of covering number to control the complexity of $\mathcal{L}_{adv}$, thus removing the maximization on the adversarial loss, which is the first challenge of our work. The second challenge comes from the estimating of the covering number of $\mathcal{L}_{dis}$ based on the extended dataset $\widehat{S}$. Inspired by Bartlett et al. (2017), we break the task of covering a whole function class $\widehat{F}$ on $\widehat{S}$ into a cover term of the first layer and other cover terms for the layers' weights. We proceed with the proofs as follows.

*proof of Lemma 4.6.* Firstly, based on Assumption 4.2 and Lemma C.1, we have

$$|\max_{\widetilde{\boldsymbol{X}} \in \mathcal{A}_{\boldsymbol{\delta}}} \ell(f_i(\boldsymbol{A}, \widetilde{\boldsymbol{X}}, \boldsymbol{W}), y_i) - \max_{\widetilde{\boldsymbol{X}} \in \mathcal{A}_{\boldsymbol{\delta}}} \ell(f_i'(\boldsymbol{A}, \widetilde{\boldsymbol{X}}, \boldsymbol{W}), y_i)| \leq C_\ell \|f_i(\boldsymbol{A}, \widetilde{\boldsymbol{X}}, \boldsymbol{W}) - f_i'(\boldsymbol{A}, \widetilde{\boldsymbol{X}}, \boldsymbol{W})\|_\infty.$$

By the definition of covering number, we can obtain

$$\mathcal{N}(\mathcal{L}_{dis}, \epsilon/2, |\cdot|_\infty, \widehat{S}) = \mathcal{N}(\tilde{\ell} \circ \mathcal{F}, \epsilon/2, |\cdot|_\infty, \widehat{S}) \leq \mathcal{N}(\widehat{\mathcal{F}}, \epsilon/2C_\ell, \|\cdot\|_\infty, \widehat{S}),$$

where $\tilde{\ell} = \max_{\widetilde{\boldsymbol{X}} \in \mathcal{A}_{\boldsymbol{\delta}}} \ell(f_i(\boldsymbol{A}, \widetilde{\boldsymbol{X}}, \boldsymbol{W}), y_i)$, and $\widehat{\mathcal{F}} := \{(\boldsymbol{x}_i, \hat{\boldsymbol{\delta}}) \to f_i(\boldsymbol{A}, \widehat{\boldsymbol{X}}, \boldsymbol{W}) : \hat{\boldsymbol{\delta}} \in \mathcal{C}_\mathcal{B}\}$. To decompose the covering number of function class $\widehat{\mathcal{F}}$ under dataset $\widehat{S}$, we construct covers $E_0$ and $E_1$ of the initial layer and the rest layers, respectively, where the cover of the first part depends on the cardinality of $\mathcal{C}_\mathcal{B}$, and covers of the rest parts depend on each choice $\{\boldsymbol{W}_1, \ldots, \boldsymbol{W}_L\}$. To facilitate our analysis, we consider the rest as a whole there, which will be analyzed in the next section.

- For any $\boldsymbol{x}_i \in \mathcal{X}$ and $\hat{\boldsymbol{\delta}} \in \mathcal{C}_\mathcal{B}$, choose an $\epsilon_0$-cover $E_0$ of $\{\boldsymbol{x}_i + \hat{\boldsymbol{\delta}} : \hat{\boldsymbol{\delta}} \in \mathcal{C}_\mathcal{B}\}$, thus

$$|E_0| \leq \mathcal{N}(\{\boldsymbol{x}_i + \hat{\boldsymbol{\delta}} : \hat{\boldsymbol{\delta}} \in \mathcal{C}_\mathcal{B}\}, \epsilon_0, \|\cdot\|_\infty, \widehat{S}) = |\mathcal{C}_\mathcal{B}(\epsilon/2C_\ell K_f)|,$$

  as the complexity of the adversarial examples $\{\widehat{\boldsymbol{X}}\}$ is controled by the cover set of the adversarial perturbations $\mathcal{C}_\mathcal{B}$.

- For any $\boldsymbol{x}_i \in \mathcal{X}$ and fixed $\hat{\boldsymbol{\delta}}_c \in \mathcal{C}_\mathcal{B}$, choose an $\epsilon_1$-cover $E_1$ of $\widehat{\mathcal{F}}$. Recalling that

$$\widehat{\mathcal{F}} := \{(\boldsymbol{x}_i, \hat{\boldsymbol{\delta}}_c) \to f_i(\boldsymbol{A}, \widehat{\boldsymbol{X}}, \boldsymbol{W}) : \hat{\boldsymbol{\delta}}_c \in \mathcal{C}_\mathcal{B}\},$$

  thus

$$|E_1| \leq \mathcal{N}(\widehat{\mathcal{F}}, \epsilon_1, \|\cdot\|_\infty, S) = \mathcal{N}(\widehat{\mathcal{F}}, \epsilon/2C_\ell, \|\cdot\|_\infty, S),$$

  where the last equality is because the changes of the perturbation cover won't change the model function class.

Lastly, as the complexity of class $\widetilde{\mathcal{F}}$ depends on the choices of the two parts, the following inequalities hold

$$
\begin{aligned}
\mathcal{N}(\mathcal{L}_{dis}, \epsilon/2, \| \cdot \|_\infty, \widehat{S}) \leq & \mathcal{N}(\widehat{\mathcal{F}}, \epsilon/2C_\ell, | \cdot |_\infty, \widetilde{S}) \\
\leq & |E_0||E_1| \\
\leq & |\mathcal{C}_\mathcal{B}(\epsilon/2C_\ell K_f)|\mathcal{N}(\widehat{\mathcal{F}}, \epsilon/2C_\ell, \| \cdot \|_\infty, S) \\
\leq & \left(\frac{6\theta C_\ell K_f}{\epsilon}\right)^d \mathcal{N}(\widehat{\mathcal{F}}, \epsilon/2C_\ell, \| \cdot \|_\infty, S),
\end{aligned}
$$

where the last inequality is due to Lemma C.2. □

Lemma 4.6 finally gets the covering number of the function class $\widehat{F}$ with fixed perturbations controlled by the constructed cover $\mathcal{C}_\mathcal{B}$ on $S$. Next, we utilize transductive Rademacher complexity to derive the covering number-based generalization bound over the adversarial loss.

**Definition B.1** (Transductive Rademacher complexity). Let $\mathcal{G}$ be a set of vectors from $\mathbb{R}^{m+u}$ and $p \in [0, \frac{1}{2}]$. Let $\boldsymbol{g} = (g_1, \ldots, g_{m+u}) \in \mathbb{R}^{m+u}$ and $\boldsymbol{\sigma} = (\sigma_1, \ldots, \sigma_{m+u})^T$ be a vector of i.i.d. random variables such that

$$
\sigma_i = \begin{cases} 1 & with probability & p; \\ -1 & with probability & p; \\ 0 & with probability & 1 - 2p. \end{cases}
$$

The transductive Rademacher complexity with parameter $p$ is

$$
\mathfrak{R}_{m+u}(\mathcal{G}) = \left(\frac{1}{m} + \frac{1}{u}\right)\mathbb{E}_{\boldsymbol{\sigma}}\left\{ \sup_{g_i \in \mathcal{G}} \sum_{i=1}^{m+u} \sigma_i g_i \right\}.
$$

*Proof of Theorem 4.8.* Combining Lemma C.3 with C.4 and considering the adversarial learning, we can obtain

$$
\widetilde{R}_u(f) \leq \widetilde{R}_m(f) + \inf_{\mu>0}\left(\frac{4\mu}{\sqrt{n}} + \frac{12}{n}\int_\mu^{2q\sqrt{n}} \sqrt{\log \mathcal{N}(\mathcal{L}_{adv}, \epsilon, | \cdot |_\infty, S)}d\epsilon\right) + qc_0 Q_1 \sqrt{\min(m, u)} + 2q\sqrt{\frac{Q_1 Q_2}{2}\ln\frac{1}{\delta}}.
$$

Then, we apply Lemma 4.4 and Lemma 4.6, the following inequalities hold

$$
\begin{aligned}
& \widetilde{R}_u(f) - \widetilde{R}_m(f) \\
& \leq \inf_{\mu>0}\left(\frac{4\mu}{\sqrt{n}} + \frac{12}{n}\int_\mu^{2q\sqrt{m+u}} \sqrt{\log \mathcal{N}(\mathcal{L}_{adv}, \epsilon, | \cdot |_\infty, S)}d\epsilon + qc_0 Q_1 \sqrt{\min(m, u)} + 2q\sqrt{\frac{Q_1 Q_2}{2}\ln\frac{1}{\delta}}\right. \\
& \leq \inf_{\mu>0}\left(\frac{4\mu}{\sqrt{n}} + \frac{12}{n}\int_\mu^{2q\sqrt{m+u}} \sqrt{\log \mathcal{N}(\mathcal{L}_{dis}, \epsilon/2, | \cdot |_\infty, \widetilde{S})}d\epsilon + qc_0 Q_1 \sqrt{\min(m, u)} + 2q\sqrt{\frac{Q_1 Q_2}{2}\ln\frac{1}{\delta}}\right. \\
& \leq \inf_{\mu>0}\left(\frac{4\mu}{\sqrt{n}} + \frac{12}{n}\int_\mu^{2q\sqrt{m+u}} \sqrt{\log \left(\frac{6\theta C_\ell K_f}{\epsilon}\right)^d \mathcal{N}(\widehat{\mathcal{F}}, \epsilon/2C_\ell, \| \cdot \|_\infty, S)}d\epsilon + qc_0 Q_1 \sqrt{\min(m, u)} + 2q\sqrt{\frac{Q_1 Q_2}{2}\ln\frac{1}{\delta}}\right. \\
& \leq \inf_{\mu>0}\left(\frac{4\mu}{\sqrt{n}} + \frac{24C_\ell}{n}\int_{\frac{\mu}{2C_\ell}}^{\frac{q\sqrt{m+u}}{C_\ell}} \sqrt{\log \left(\frac{3\theta K_f}{\epsilon}\right)^d \mathcal{N}(\widehat{\mathcal{F}}, \epsilon, \| \cdot \|_\infty, S)}d\epsilon + qc_0 Q_1 \sqrt{\min(m, u)} + 2q\sqrt{\frac{Q_1 Q_2}{2}\ln\frac{1}{\delta}}\right,
\end{aligned}
$$

where the last inequality replaces $\epsilon/2C_\ell$ with $\epsilon$ and remains the complexity estimate of $\widehat{\mathcal{F}}$ in the next section. □

## B.2. Proofs of Case Study.

Based on Theorem 4.8, we analyze the adversarial generalization of several classical GNNs via the lens of covering number.

*Proof of Proposition 4.14.* We first derive the Lipschitz constant $K_f$ of GCN. Let the update rule of node $\mathbf{x}_i$ with $l \in [L]$ is denoted by $Z_{i*}(\widetilde{\boldsymbol{X}}, \boldsymbol{W}_1, \ldots, \boldsymbol{W}_l) = \sigma_l(\sum_{j=1}^n [g(\boldsymbol{A})]_{ij} Z_{j*}(\widetilde{\boldsymbol{X}}, \boldsymbol{W}_1, \ldots, \boldsymbol{W}_{l-1})\boldsymbol{W}_l)$, where $Z_{i*}(\cdot)$ means the $i$-th line of

the output matrix. Thus we have $h(\boldsymbol{z}_i, \boldsymbol{\delta}) = \ell(f_i(\boldsymbol{A}, \widetilde{\boldsymbol{X}}, \boldsymbol{W}), y_i) = \ell(Z_{i*}(\widetilde{\boldsymbol{X}}, \boldsymbol{W}_1, \dots, \boldsymbol{W}_L), y_i)$ hold for any $\boldsymbol{\delta} \in \mathcal{B}$. The following inequalities hold

$$
\begin{aligned}
&|h(\boldsymbol{z}_i, \boldsymbol{\delta}) - h(\boldsymbol{z}_i, \boldsymbol{\delta}')| \\
\leq& C_\ell \|Z_{i*}(\widetilde{\boldsymbol{X}}, \boldsymbol{W}_1, \dots, \boldsymbol{W}_L) - Z_{i*}(\widetilde{\boldsymbol{X}}', \boldsymbol{W}_1, \dots, \boldsymbol{W}_L)\|_\infty \\
\leq& C_\ell \|\sigma_L\big(\sum_{j=1}^n [g(\boldsymbol{A})]_{ij} Z_{j*}(\widetilde{\boldsymbol{X}}, \boldsymbol{W}_1, \dots, \boldsymbol{W}_{L-1}) \boldsymbol{W}_L\big) - \sigma_L\big(\sum_{j=1}^n [g(\boldsymbol{A})]_{ij} Z_{j*}(\widetilde{\boldsymbol{X}}', \boldsymbol{W}_1, \dots, \boldsymbol{W}_{L-1}) \boldsymbol{W}_L\big)\|_\infty \\
\leq& C_\ell \rho_L \sum_{j=1}^n [g(\boldsymbol{A})]_{ij} \|Z_{j*}(\widetilde{\boldsymbol{X}}, \boldsymbol{W}_1, \dots, \boldsymbol{W}_{L-1}) \boldsymbol{W}_L - Z_{j*}(\widetilde{\boldsymbol{X}}', \boldsymbol{W}_1, \dots, \boldsymbol{W}_{L-1}) \boldsymbol{W}_L\|_\infty \\
\leq& C_\ell \rho_L \|g(\boldsymbol{A})\|_\infty \max_j \|Z_{j*}(\widetilde{\boldsymbol{X}}, \boldsymbol{W}_1, \dots, \boldsymbol{W}_{L-1}) - Z_{j*}(\widetilde{\boldsymbol{X}}', \boldsymbol{W}_1, \dots, \boldsymbol{W}_{L-1})\|_\infty \|\boldsymbol{W}_L\|_\infty \\
\leq& C_\ell g^L \rho_1 \prod_{t=2}^L w_t \rho_t \max_j \|\widetilde{\boldsymbol{X}}_{j*} \boldsymbol{W}_1 - \widetilde{\boldsymbol{X}}'_{j*} \boldsymbol{W}_1\|_\infty \\
\leq& C_\ell g^L \prod_{t=1}^L w_t \rho_t \max_j \|\widetilde{\boldsymbol{X}}_{j*} - \widetilde{\boldsymbol{X}}'_{j*}\|_\infty \\
\leq& C_\ell g^L \prod_{t=1}^L w_t \rho_t \max_j \|\widetilde{\boldsymbol{x}}_j + \boldsymbol{\delta} - \widetilde{\boldsymbol{x}}_j - \boldsymbol{\delta}'\|_\infty \\
\leq& C_\ell g^L \prod_{t=1}^L w_t \rho_t \|\boldsymbol{\delta} - \boldsymbol{\delta}'\|_\infty \\
\triangleq& K_{\mathrm{GCN}} \|\boldsymbol{\delta} - \boldsymbol{\delta}'\|_\infty,
\end{aligned}
$$

where the first inequality is due to Assumption 4.2, the third inequality is due to Assumption 4.10, the fourth inequality is according to the compatibility of the matrix norm constraint, and the fifth inequality is based on Assumption 4.11.

Now we analyze the covering number of model function class $\widehat{\mathcal{F}}$. Denote by $\Delta_l = \|Z_{i*}(\widehat{\boldsymbol{X}}_c, \boldsymbol{W}_1, \dots, \boldsymbol{W}_l) - Z_{i*}(\widehat{\boldsymbol{X}}_c, \boldsymbol{W}'_1, \dots, \boldsymbol{W}'_l)\|_\infty$, where $i \in [n]$ and $l \in [L]$. Based on the Lipschitz property of activation function , we first observe that

$$
\begin{aligned}
\Delta_1 =& \|Z_{i*}(\widehat{\boldsymbol{X}}_c, \boldsymbol{W}_1) - Z_{i*}(\widehat{\boldsymbol{X}}_c, \boldsymbol{W}'_1)\|_\infty \\
\leq& \|\sigma_1\big(\sum_{j=1}^n [g(\boldsymbol{A})]_{ij} \widehat{\boldsymbol{X}}_{c,j*} \boldsymbol{W}_1\big) - \sigma_1\big(\sum_{j=1}^n [g(\boldsymbol{A})]_{ij} \widehat{\boldsymbol{X}}_{c,j*} \boldsymbol{W}'_1\big)\|_\infty \\
\leq& \rho_1 \sum_{j=1}^n [g(\boldsymbol{A})]_{ij} \|\widehat{\boldsymbol{X}}_{c,j*}(\boldsymbol{W}_1 - \boldsymbol{W}'_1)\|_\infty \\
\leq& \rho_1 g \max_j \|\widehat{\boldsymbol{X}}_{c,j*}\|_2 \|(\boldsymbol{W}_1 - \boldsymbol{W}'_1)\|_\infty \\
\leq& \rho_1 g \hat{b} \|\boldsymbol{W}_1 - \boldsymbol{W}'_1\|_\infty \\
=& \hat{b} g \rho_1 w_1 \frac{\|\boldsymbol{W}_1 - \boldsymbol{W}_1\|_\infty}{w_1},
\end{aligned}
$$

where the third inequality is due to the fact that $\|\boldsymbol{x}\|_\infty \leq \|\boldsymbol{x}\|_2$ holds for any vector $\boldsymbol{x}$. For $\widehat{\boldsymbol{X}}_{c,j*} = (\boldsymbol{x}_j, \hat{\boldsymbol{\delta}}_c)$, we have

$$
\hat{b} = \|\widehat{\boldsymbol{X}}_{c,j*}\|_2 \leq \|\boldsymbol{x}_j\|_2 + \|\hat{\boldsymbol{\delta}}_c\|_2 \leq b + \sqrt{d}\|\boldsymbol{\delta}\|_\infty = b + \sqrt{d}\theta. \tag{3}
$$

Similarly, we have

$$
\begin{aligned}
\Delta_2 =& \|Z_{i*}(\widehat{\boldsymbol{X}}_c, \boldsymbol{W}_1, \boldsymbol{W}_2) - Z_{i*}(\widehat{\boldsymbol{X}}_c, \boldsymbol{W}'_1, \boldsymbol{W}'_2)\|_\infty \\
\leq& \|\sigma_2\big(\sum_{j=1}^n [g(\boldsymbol{A})]_{ij} Z_{j*}(\widehat{\boldsymbol{X}}_c, \boldsymbol{W}_1)\boldsymbol{W}_2\big) - \sigma_2\big(\sum_{j=1}^n [g(\boldsymbol{A})]_{ij} Z_{j*}(\widehat{\boldsymbol{X}}_c, \boldsymbol{W}'_1)\boldsymbol{W}'_2\big)\|_\infty \\
\leq& \rho_2 \sum_{j=1}^n [g(\boldsymbol{A})]_{ij} \|Z_{j*}(\widehat{\boldsymbol{X}}_c, \boldsymbol{W}_1)(\boldsymbol{W}_2 - \boldsymbol{W}'_2) + (Z_{j*}(\widehat{\boldsymbol{X}}_c, \boldsymbol{W}_1) - Z_{j*}(\widehat{\boldsymbol{X}}_c, \boldsymbol{W}'_1))\boldsymbol{W}'_2\|_\infty \\
\leq& \rho_2 \|g(\boldsymbol{A})\|_\infty \big(\|\sigma_1\big(\sum_{j=1}^n [g(\boldsymbol{A})]_{ij}\widehat{\boldsymbol{X}}_{c,j*}\boldsymbol{W}_1\big)(\boldsymbol{W}_2 - \boldsymbol{W}'_2)\|_\infty + \Delta_1 w_2\big) \\
\leq& \rho_2 \|g(\boldsymbol{A})\|_\infty \big(\|\sigma_1\big(\sum_{j=1}^n [g(\boldsymbol{A})]_{ij}\widehat{\boldsymbol{X}}_{c,j*}\boldsymbol{W}_1\big)\|_\infty \|\boldsymbol{W}_2 - \boldsymbol{W}'_2\|_\infty + \Delta_1 w_2\big) \\
\leq& \rho_2 \|g(\boldsymbol{A})\|_\infty \big(\hat{b}g\rho_1 w_1 \|\boldsymbol{W}_2 - \boldsymbol{W}'_2\|_\infty + \hat{b}g\rho_1 w_2 \|\boldsymbol{W}_1 - \boldsymbol{W}'_1\|_\infty\big) \\
=& \hat{b}g^2 \rho_1 \rho_2 w_1 w_2 \big(\frac{\|\boldsymbol{W}_1 - \boldsymbol{W}'_1\|_\infty}{w_1} + \frac{\|\boldsymbol{W}_2 - \boldsymbol{W}'_2\|_\infty}{w_2}\big),
\end{aligned}
$$

where the second inequality follows by the triangle inequality and the last inequality is followed by the proofs of $K_{GCN}$. Then, for any $l > 0$, using the induction step, we have

$$
\begin{aligned}
\Delta_{l+1} =& \|Z_{i*}(\widehat{\boldsymbol{X}}, \boldsymbol{W}_1, \ldots, \boldsymbol{W}_{l+1}) - Z_{i*}(\widehat{\boldsymbol{X}}, \boldsymbol{W}'_1, \ldots, \boldsymbol{W}'_{l+1})\|_\infty \\
\leq& \|\sigma_{l+1}\big(\sum_{j=1}^n [g(\boldsymbol{A})]_{ij} Z_{j*}(\widehat{\boldsymbol{X}}_c, \boldsymbol{W}_1, \ldots, \boldsymbol{W}_l)\boldsymbol{W}_{l+1}\big) - \sigma_{l+1}\big(\sum_{j=1}^n [g(\boldsymbol{A})]_{ij} Z_{j*}(\widehat{\boldsymbol{X}}_c, \boldsymbol{W}'_1, \ldots, \boldsymbol{W}'_l)\boldsymbol{W}'_{l+1}\big)\|_\infty \\
\leq& \rho_{l+1} g \Big(\|Z_{j*}(\widehat{\boldsymbol{X}}_c, \boldsymbol{W}_1, \ldots, \boldsymbol{W}_l)\boldsymbol{W}_{l+1} - Z_{j*}(\widehat{\boldsymbol{X}}_c, \boldsymbol{W}_1, \ldots, \boldsymbol{W}_l)\boldsymbol{W}'_{l+1}\|_\infty \\
& + \|Z_{j*}(\widehat{\boldsymbol{X}}_c, \boldsymbol{W}_1, \ldots, \boldsymbol{W}_l)\boldsymbol{W}'_{l+1} - Z_{j*}(\widehat{\boldsymbol{X}}_c, \boldsymbol{W}'_1, \ldots, \boldsymbol{W}'_l)\boldsymbol{W}'_{l+1}\|_\infty\Big) \\
\leq& \hat{b}\rho_{l+1} g^{l+1} \prod_{t=1}^l \rho_t w_t \|\boldsymbol{W}_{l+1} - \boldsymbol{W}'_{l+1}\|_\infty + \rho_{l+1} g \Delta_l \\
\leq& \hat{b}g^{l+1} \prod_{t=1}^{l+1} \rho_t w_t \sum_{i=1}^{l+1} \frac{\|\boldsymbol{W}_i - \boldsymbol{W}'_i\|_\infty}{w_i}.
\end{aligned}
$$

Now we proceed to upper bound the covering number for $\widehat{\mathcal{F}}$. For any $f_i(\cdot)$ and $f'_i(\cdot)$, applying the above inequality, the following hold

$$
\|f_i(\boldsymbol{A}, \widehat{\boldsymbol{X}}, \boldsymbol{W}) - f'_i(\boldsymbol{A}, \widehat{\boldsymbol{X}}, \boldsymbol{W})\|_\infty = \Delta_L \leq \hat{b}g^L \prod_{t=1}^L \rho_t w_t \sum_{j=1}^L \frac{\|\boldsymbol{W}_j - \boldsymbol{W}'_j\|_\infty}{w_j}.
$$

Then, based on the definition of the covering number, we have the following relation of the cover between the class $\widehat{\mathcal{F}}$ and $\mathcal{W} = \{\boldsymbol{W}_j : \|\boldsymbol{W}_j\|_\infty \leq w_j\}$

$$
\epsilon \leq \hat{b}g^L \prod_{t=1}^L \rho_t w_t \sum_{j=1}^L \frac{\epsilon_j}{w_j}. \tag{4}
$$

Thus, according to Theorem 4.8, we focus on

$$
\int_{\frac{\mu}{2C_\ell}}^{\frac{q\sqrt{m+u}}{C_\ell}} \sqrt{\log\big(\frac{3\theta K_{\text{GCN}}}{\epsilon}\big)^d \mathcal{N}(\widehat{\mathcal{F}}, \epsilon, \|\cdot\|_\infty, S)} d\epsilon
$$

$$
\leq \int_{\frac{\mu}{2C_\ell}}^{\frac{q\sqrt{m+u}}{C_\ell}} \sqrt{\log\big(\frac{3\theta K_{\text{GCN}}}{\epsilon}\big)^d + \log \mathcal{N}(\widehat{\mathcal{F}}, \epsilon, |\cdot|_\infty, S)} d\epsilon
$$

$$
\leq \int_{\frac{\mu}{2C_\ell}}^{\frac{q\sqrt{m+u}}{C_\ell}} \left\{ \sqrt{\log\big(\frac{3\theta K_{\text{GCN}}}{\epsilon}\big)^d} + \sqrt{\log \mathcal{N}(\widehat{\mathcal{F}}, \epsilon, |\cdot|_\infty, S)} d\epsilon \right\}
$$

$$
\leq \int_{\frac{\mu}{2C_\ell}}^{\frac{q\sqrt{m+u}}{C_\ell}} \left\{ \sqrt{\log\big(\frac{3\theta K_{\text{GCN}}}{\epsilon}\big)^d} + \sqrt{\log \prod_{j=1}^{L} \mathcal{N}(\{\boldsymbol{W}_j : \|\boldsymbol{W}_j\|_\infty \leq w_j\}, \|\cdot\|_\infty, \epsilon_j)} d\epsilon \right\}
$$

$$
\leq \int_{\frac{\mu}{2C_\ell}}^{\frac{q\sqrt{m+u}}{C_\ell}} \sqrt{d\log\big(\frac{3\theta K_{\text{GCN}}}{\epsilon}\big)} d\epsilon + \int_{\frac{\mu}{2C_\ell}}^{\frac{q\sqrt{m+u}}{C_\ell}} \sqrt{\sum_{j=1}^{L} \log\{\mathcal{N}(\boldsymbol{W}_j : \|\boldsymbol{W}_j\|_\infty \leq w_j, \|\cdot\|_\infty, \epsilon_j)\}} d\epsilon. \tag{5}
$$

where the second inequality follows by $(\sqrt{a+b})^2 \leq (\sqrt{a} + \sqrt{b})^2$, and the third inequality is followed by (Bartlett et al., 2017; Tu et al., 2019). Next, we proceed to the proof of the first term as follows

$$
\sqrt{d} \int_{\frac{\mu}{2C_\ell}}^{\frac{q\sqrt{m+u}}{C_\ell}} \sqrt{\log\big(\frac{3\theta K_{\text{GCN}}}{\epsilon}\big)} d\epsilon \leq \sqrt{d} \int_{0}^{K_{\text{GCN}}\theta} \sqrt{\log\big(\frac{3\theta K_{\text{GCN}}}{\epsilon}\big)} d\epsilon \leq 3\sqrt{d}\theta K_{\text{GCN}} \int_{0}^{1/3} \sqrt{\log\frac{1}{\epsilon}} d\epsilon, \tag{6}
$$

where the first inequality is based on the fact that $\|\boldsymbol{\delta} - \boldsymbol{\delta}'\|_\infty \leq \|\boldsymbol{\delta}\|_\infty$. Since $\boldsymbol{W}_j \in \mathbb{R}^{d_j \times d_{j-1}}$, we can regard it as a vector in $\mathbb{R}^{v^2}$, where $v = \max\{d_0, \ldots, d_L\}$. Then the set $\{\boldsymbol{W}_j : \|\boldsymbol{W}_j\|_\infty \leq w_j\}$ forms a $w_j$-ball in $\mathbb{R}^{v^2}$, and the covering number for this ball can be upper bounded by the following inequality for $0 \leq \epsilon_j \leq w_j$ (Long & Sedghi, 2019)

$$
\mathcal{N}(\boldsymbol{W}_j : \|\boldsymbol{W}_j\|_\infty \leq w_j, \|\cdot\|_\infty, \epsilon_j) \leq \big(\frac{3w_j}{\epsilon_j}\big)^{v^2}, \tag{7}
$$

and $\mathcal{N}(\boldsymbol{W}_j : \|\boldsymbol{W}_j\|_\infty \leq w_j, \|\cdot\|_\infty, \epsilon_j) = 1$ for $\epsilon_j \geq w_j$. Then, by applying equation (7), we focus on the second term in equation (5) as follows.

$$
\int_{\frac{\mu}{2C_\ell}}^{\frac{q\sqrt{m+u}}{C_\ell}} \sqrt{\sum_{j=1}^{L} \log\{\mathcal{N}(\boldsymbol{W}_j : \|\boldsymbol{W}_j\|_\infty \leq w_j, \|\cdot\|_\infty, \epsilon_j)\}} d\epsilon
$$

$$
\leq \sum_{j=1}^{L} \int_{\frac{\mu}{2C_\ell}}^{\frac{q\sqrt{m+u}}{C_\ell}} \sqrt{v^2 \log \frac{3w_j}{\epsilon_j}} d\epsilon
$$

$$
\leq \sum_{j=1}^{L} v \int_{\frac{\mu}{2C_\ell}}^{\frac{q\sqrt{m+u}}{C_\ell}} \sqrt{\log \frac{3w_j}{\epsilon_j}} d\epsilon_j \frac{\hat{b}g^L \prod_{t=1}^{L} \rho_t w_t}{w_j}
$$

$$
\leq \sum_{j=1}^{L} \frac{v\hat{b}g^L \prod_{t=1}^{L} \rho_t w_t}{w_j} \int_{0}^{\infty} \sqrt{\log \frac{3w_j}{\epsilon_j}} d\epsilon_j
$$

$$
\leq \sum_{j=1}^{L} \frac{v\hat{b}g^L \prod_{t=1}^{L} \rho_t w_t}{w_j} \int_{0}^{s_j} \sqrt{\log \frac{3w_j}{\epsilon_j}} d\epsilon_j
$$

$$
\leq 3v\hat{b}g^L \prod_{t=1}^{L} \rho_t w_t \sum_{j=1}^{L} \frac{w_j}{w_j} \int_{0}^{1/3} \sqrt{\log \frac{1}{\epsilon_j}} d\epsilon_j
$$

$$
\leq 3v\hat{b}g^L \prod_{t=1}^{L} \rho_t w_t \int_{0}^{1/3} \sqrt{\log \frac{1}{\epsilon_j}} d\epsilon_j, \tag{8}
$$

where the second inequality is based on equation (4) and the forth inequality is due to the condition that $\epsilon_j \leq w_j$. Then combining equation (6) and (8), we can obtain

$$\int_{\frac{\mu}{2C_\ell}}^{\frac{q\sqrt{m+u}}{C_\ell}} \sqrt{\log\big(\frac{3\theta K_{\text{GCN}}}{\epsilon}\big)^d \mathcal{N}(\widehat{\mathcal{F}}, \epsilon, |\cdot|_\infty, S)} d\epsilon$$

$$\leq 3\Big(\sqrt{d}\theta K_{\text{GCN}} \int_0^{1/3} \sqrt{\log\frac{1}{\epsilon}} d\epsilon + v\hat{b}g^L \prod_{t=1}^L \rho_t w_t \int_0^{1/3} \sqrt{\log\frac{1}{\epsilon_j}} d\epsilon_j\Big)$$

$$\leq 3\Big(\sqrt{d}\theta K_{\text{GCN}} + v\hat{b}g^L \prod_{t=1}^L \rho_t w_t\Big),$$

where the last inequality uses $\int_0^{1/3} \sqrt{\log\frac{1}{\epsilon}} d\epsilon = \frac{1}{6}\big(2\sqrt{\log 3} + 2\sqrt{\pi}\text{erfc}\big(\sqrt{\log 3}\big)\big) \leq 1$ (Tu et al., 2019). Finally, according to Theorem 4.8, by setting $\mu$ to $\frac{1}{n}$, we can obtain the final generalization gap

$$\widetilde{R}_u(f) \leq \widetilde{R}_m(f) + \Big(\frac{4\mu}{\sqrt{m+u}} + \frac{72C_\ell}{\sqrt{m+u}}\big(\sqrt{d}\theta K_{\text{GCN}} + v\hat{b}g^L \prod_{t=1}^L \rho_t w_t\big) + qc_0 Q_1 \sqrt{\min(m,u)} + 2q\sqrt{\frac{Q_1 Q_2}{2} \ln\frac{1}{\delta}}$$

$$\leq \widetilde{R}_m(f) + C_\ell\Big(\frac{4}{m+u} + \frac{72C_\ell}{\sqrt{m+u}}\big(\sqrt{d}\theta K_{\text{GCN}} + v\hat{b}g^L \prod_{t=1}^L \rho_t w_t\big) + qc_0 Q_1 \sqrt{\min(m,u)} + 2q\sqrt{\frac{Q_1 Q_2}{2} \ln\frac{1}{\delta}}.$$

Now we complete the proofs of Proposition 4.14. $\qquad\qquad\qquad\qquad\qquad\qquad\qquad\qquad\qquad\qquad\qquad\qquad\square$

*Proof of Proposition 4.16.* Denote by $Z_{i*}(\widehat{\boldsymbol{X}}_c, \boldsymbol{W}_1, \ldots, \boldsymbol{W}_l) = \sigma_l(Z_{i*}(\widehat{\boldsymbol{X}}_c, \boldsymbol{W}_1, \ldots, \boldsymbol{W}_{l-1})\boldsymbol{W}_l)$ for $l \in [L]$. Note that $f_i(\boldsymbol{A}, \widehat{\boldsymbol{X}}_c, \boldsymbol{W}) = \sigma_L(\sum_{j=1}^n [\hat{g}(\boldsymbol{A})]_{ij} Z_{j*}(\widehat{\boldsymbol{X}}_c, \boldsymbol{W}_1, \ldots, \boldsymbol{W}_{L-1}))$, where $\sum_{j=1}^n [\hat{g}(\boldsymbol{A})]_{ij} = \sum_{k=0}^{K-1} \gamma(1-\gamma)^k \hat{g}(\boldsymbol{A})^k + (1-\gamma)^K \hat{g}(\boldsymbol{A})^K$. Firstly, the Lipschitz constant $K_{\text{APPNP}}$ is derived by

$$|h(\boldsymbol{z}, \boldsymbol{\delta}) - h(\boldsymbol{z}, \boldsymbol{\delta}')|$$

$$\leq C_\ell \|\sigma_L\big(\sum_{j=1}^n [\hat{g}(\boldsymbol{A})]_{ij} Z_{j*}(\widetilde{\boldsymbol{X}}, \boldsymbol{W}_1, \ldots, \boldsymbol{W}_{L-1})\big) - \sigma_L\big(\sum_{j=1}^n [\hat{g}(\boldsymbol{A})]_{ij} Z_{j*}(\widetilde{\boldsymbol{X}}', \boldsymbol{W}_1, \ldots, \boldsymbol{W}_{L-1})\big)\|_\infty$$

$$\leq C_\ell \rho_L \hat{g} \|Z_{j*}(\widetilde{\boldsymbol{X}}, \boldsymbol{W}_1, \ldots, \boldsymbol{W}_{L-1}) - Z_{j*}(\widetilde{\boldsymbol{X}}', \boldsymbol{W}_1, \ldots, \boldsymbol{W}_{L-1})\|_\infty$$

$$\leq C_\ell \rho_L \hat{g} \|\sigma_{L-1}(Z_{j*}(\widetilde{\boldsymbol{X}}, \boldsymbol{W}_1, \ldots, \boldsymbol{W}_{L-2})\boldsymbol{W}_{L-1}) - \sigma_{L-1}(Z_{j*}(\widetilde{\boldsymbol{X}}', \boldsymbol{W}_1, \ldots, \boldsymbol{W}_{L-2})\boldsymbol{W}_{L-1})\|_\infty$$

$$\leq C_\ell \hat{g} \rho_L \prod_{t=1}^{L-1} \rho_t w_t \|\boldsymbol{\delta} - \boldsymbol{\delta}'\|_\infty$$

$$\triangleq K_{\text{APPNP}} \|\boldsymbol{\delta} - \boldsymbol{\delta}'\|_\infty,$$

where $\hat{g} = \|\hat{g}(\boldsymbol{A})\|_\infty \leq \gamma\big(1 + \sum_{k=1}^{K-1} \big((1-\gamma)g\big)^k\big) + \big((1-\gamma)g\big)^K$. Now we proceed to the covering number of $\widehat{\mathcal{F}}$. Let $\Delta_l = \|Z_{i*}(\widehat{\boldsymbol{X}}_c, \boldsymbol{W}_1, \ldots, \boldsymbol{W}_l) - Z_{i*}(\widehat{\boldsymbol{X}}_c, \boldsymbol{W}_1', \ldots, \boldsymbol{W}_l')\|_\infty$, we have

$$\Delta_1 = \|Z_{i*}(\widehat{\boldsymbol{X}}_c, \boldsymbol{W}_1) - Z_{i*}(\widehat{\boldsymbol{X}}_c, \boldsymbol{W}_1')\|$$

$$= \|\sigma_1(\widehat{\boldsymbol{X}}_{c,i*}\boldsymbol{W}_1) - \sigma_1(\widehat{\boldsymbol{X}}_{c,i*}\boldsymbol{W}_1')\|_\infty$$

$$\leq \rho_1 \|\widehat{\boldsymbol{X}}_{c,i*}(\boldsymbol{W}_1 - \boldsymbol{W}_1')\|_\infty$$

$$\leq \hat{b}\rho_1 \|\boldsymbol{W}_1 - \boldsymbol{W}_1'\|_\infty.$$

Similar to the proofs of Proposition 4.14, using the induction step, we derive that

$$\Delta_{l+1} = \|Z_{i*}(\widehat{\boldsymbol{X}}_c, \boldsymbol{W}_1, \ldots, \boldsymbol{W}_{l+1}) - Z_{i*}(\widehat{\boldsymbol{X}}_c, \boldsymbol{W}_1', \ldots, \boldsymbol{W}_{l+1}')\|_\infty$$

$$\leq \rho_{l+1} \|Z_{i*}(\widehat{\boldsymbol{X}}_c, \boldsymbol{W}_1, \ldots, \boldsymbol{W}_l)\boldsymbol{W}_{l+1} - Z_{i*}(\widehat{\boldsymbol{X}}_c, \boldsymbol{W}_1', \ldots, \boldsymbol{W}_l')\boldsymbol{W}_{l+1}'\|_\infty$$

$$\leq \hat{b} \prod_{t=1}^{l+1} \rho_t w_t \sum_{j=1}^{l+1} \frac{\|\boldsymbol{W}_j - \boldsymbol{W}_j'\|_\infty}{s_j}.$$

Then we can obtain the following inequalities

$$
\begin{aligned}
&\|f_i(\boldsymbol{A}, \widehat{\boldsymbol{X}}_c, \boldsymbol{W}) - f_i'(\boldsymbol{A}, \widehat{\boldsymbol{X}}_c, \boldsymbol{W})\|_\infty \\
=&\|\sigma_L\big(\sum_{j=1}^n [\hat{g}(\boldsymbol{A})]_{ij} Z_{j*}(\widehat{\boldsymbol{X}}_c, \boldsymbol{W}_1, \ldots, \boldsymbol{W}_{L-1})\big) - \sigma_L\big(\sum_{j=1}^n [\hat{g}(\boldsymbol{A})]_{ij} Z_{j*}(\widehat{\boldsymbol{X}}_c, \boldsymbol{W}_1', \ldots, \boldsymbol{W}_{L-1}')\big)\|_\infty \\
\leq& \rho_L \|\hat{g}(\boldsymbol{A})\|_\infty \Delta_{L-1} \\
\leq& \hat{b}\hat{g}\rho_L \prod_{t=1}^{L-1} \rho_t w_t \sum_{j=1}^{L-1} \frac{\|\boldsymbol{W}_j - \boldsymbol{W}_j'\|_\infty}{s_j},
\end{aligned}
$$

which implies the following relation between $\epsilon$ and $\epsilon_j$

$$
\epsilon \leq \hat{b}\hat{g}\rho_L \prod_{t=1}^{L-1} \rho_t w_t \sum_{j=1}^{L-1} \frac{\epsilon_j}{s_j}. \tag{9}
$$

By applying Equation (9), we can derive the covering number of $\widehat{F}$ below

$$
\begin{aligned}
&\int_{\frac{\mu}{2C_\ell}}^{\frac{q\sqrt{m+u}}{C_\ell}} \sqrt{\log\big(\frac{3\theta K_{\text{APPNP}}}{\epsilon}\big)^d \mathcal{N}(\widehat{\mathcal{F}}, \epsilon, \|\cdot\|_\infty, S)} d\epsilon \\
\leq& \sqrt{d} \int_{\frac{\mu}{2C_\ell}}^{\frac{q\sqrt{m+u}}{C_\ell}} \sqrt{\big(\log\frac{3\theta K_{\text{APPNP}}}{\epsilon}\big)} d\epsilon + \sum_{j=1}^L \int_{\frac{\mu}{2C_\ell}}^{\frac{q\sqrt{m+u}}{C_\ell}} \sqrt{v^2 \log\frac{3w_j}{\epsilon_j}} d\epsilon \\
\leq& \sqrt{d} \int_0^{K_{\text{APPNP}}\theta} \sqrt{\log\big(\frac{3\theta K_{\text{APPNP}}}{\epsilon}\big)} d\epsilon + \sum_{j=1}^{L-1} v \int_{\frac{\mu}{2C_\ell}}^{\frac{q\sqrt{m+u}}{C_\ell}} \sqrt{\log\frac{3w_j}{\epsilon_j}} d\epsilon_j \frac{\hat{b}\hat{g}\rho_L \prod_{t=1}^{L-1}\rho_t w_t}{s_j} \\
\leq& 3\sqrt{d}\theta K_{\text{APPNP}} \int_0^{1/3} \sqrt{\log\frac{1}{\epsilon}} d\epsilon + \sum_{j=1}^{L-1} \frac{v\hat{b}\hat{g}\rho_L \prod_{t=1}^{L-1}\rho_t w_t}{s_j} \int_0^{s_j} \sqrt{\log\frac{3w_j}{\epsilon_j}} d\epsilon_i \\
\leq& 3\sqrt{d}\theta K_{\text{APPNP}} \int_0^{1/3} \sqrt{\log\frac{1}{\epsilon}} d\epsilon + 3v\hat{b}\hat{g}\rho_L \prod_{t=1}^{L-1}\rho_t w_t \int_0^{1/3} \sqrt{\log\frac{1}{\epsilon_j}} d\epsilon_j \\
\leq& 3\Big(\sqrt{d}\theta K_{\text{APPNP}} + v\hat{b}\hat{g}\rho_L \prod_{t=1}^{L-1}\rho_t w_t\Big).
\end{aligned}
$$

where the first inequality follows by the proofs of GCN. Finally, the adversarial generalization gap of APPNP is obtained by applying the covering number of $\widehat{F}$ and setting $\mu$ to $\frac{1}{n}$

$$
\begin{aligned}
\widetilde{R}_u(f) \leq& \widetilde{R}_m(f) + \frac{4\mu}{\sqrt{m+u}} + \frac{72C_\ell}{\sqrt{m+u}}\Big(\sqrt{d}\theta K_{\text{APPNP}} + v\hat{b}\hat{g}\rho_L \prod_{t=1}^{L-1}\rho_t w_t\Big) + qc_0 Q_1 \sqrt{\min(m,u)} + 2q\sqrt{\frac{Q_1 Q_2}{2} \ln\frac{1}{\delta}} \\
\leq& \widetilde{R}_m(f) + \frac{4}{m+u} + \frac{72C_\ell}{\sqrt{m+u}}\Big(\sqrt{d}\theta K_{\text{APPNP}} + v\hat{b}\hat{g}\rho_L \prod_{t=1}^{L-1}\rho_t w_t\Big) + qc_0 Q_1 \sqrt{\min(m,u)} + 2q\sqrt{\frac{Q_1 Q_2}{2} \ln\frac{1}{\delta}}.
\end{aligned}
$$

Now the proofs of Proposition 4.16 are finished. $\qquad\square$

*Proof of Proposition 4.18.* Denote by $Z_{j*}(\widehat{\boldsymbol{X}}_c, \boldsymbol{W}_0, \ldots, \boldsymbol{W}_l) = \sigma_l\Big[\big((1-\alpha)\sum_{j=1}^n [g(\boldsymbol{A})]_{ij} Z_{j*}(\widehat{\boldsymbol{X}}_c, \boldsymbol{W}_0, \ldots, \boldsymbol{W}_{l-1}) + \alpha Z_{j*}(\widehat{\boldsymbol{X}}_c, \boldsymbol{W}_0)\big)\psi(\boldsymbol{W}_l)\Big]$, where $Z_{i*}(\widehat{\boldsymbol{X}}_c, \boldsymbol{W}_0) = \sigma_0(\widehat{\boldsymbol{X}}_c \boldsymbol{W}_0)$ and $\psi(\boldsymbol{W}_l) = (1-\beta)\boldsymbol{I}_n + \beta\boldsymbol{W}_l$. We define $f_i(\boldsymbol{A}, \widehat{\boldsymbol{X}}_c, \boldsymbol{W}) = \sigma_L(Z_{i*}(\widehat{\boldsymbol{X}}_c, \boldsymbol{W}_0, \ldots, \boldsymbol{W}_{L-1})\boldsymbol{W}_L)$. Now we first derive the Lipschitz constant $K_{\text{GCNII}}$ in the fol-

lowing

$$
\begin{aligned}
&|h(\boldsymbol{z}, \boldsymbol{\delta}) - h(\boldsymbol{z}, \boldsymbol{\delta}')| \\
&\leq C_\ell \Big\| \Big[ \sigma_L(Z_{j*}(\widetilde{\boldsymbol{X}}, \boldsymbol{W}_0, \ldots, \boldsymbol{W}_{L-1}) \boldsymbol{W}_L) - \sigma_L(Z_{j*}(\widetilde{\boldsymbol{X}}', \boldsymbol{W}_0, \ldots, \boldsymbol{W}_{L-1}) \boldsymbol{W}_L) \Big] \Big\|_\infty \\
&\leq C_\ell \rho_L \tilde{w}_L \Big\| \sigma_{L-1} \Big[ \Big( (1-\alpha) \sum_{j=1}^n [g(\boldsymbol{A})]_{ij} Z_{j*}(\widetilde{\boldsymbol{X}}, \boldsymbol{W}_0, \ldots, \boldsymbol{W}_{L-2}) + \alpha Z_{j*}(\widetilde{\boldsymbol{X}}, \boldsymbol{W}_0) \Big) \psi(\boldsymbol{W}_{L-1}) \Big] \\
&\qquad - \sigma_{L-1} \Big[ \Big( (1-\alpha) \sum_{j=1}^n [g(\boldsymbol{A})]_{ij} Z_{j*}(\widetilde{\boldsymbol{X}}', \boldsymbol{W}_0, \ldots, \boldsymbol{W}_{L-2}) + \alpha Z_{j*}(\widetilde{\boldsymbol{X}}', \boldsymbol{W}_0) \Big) \psi(\boldsymbol{W}_{L-1}) \Big] \Big\|_\infty \\
&\leq C_\ell \prod_{t=L-1}^L \rho_t \tilde{w}_t \Big[ (1-\alpha) g \Big\| Z_{j*}(\widetilde{\boldsymbol{X}}, \boldsymbol{W}_0, \ldots, \boldsymbol{W}_{L-2}) - Z_{j*}(\widetilde{\boldsymbol{X}}', \boldsymbol{W}_0, \ldots, \boldsymbol{W}_{L-2}) \Big\|_\infty \\
&\qquad + \alpha \big\| Z_{j*}(\widetilde{\boldsymbol{X}}, \boldsymbol{W}_0) - Z_{j*}(\widetilde{\boldsymbol{X}}', \boldsymbol{W}_0) \big\|_\infty \Big],
\end{aligned}
$$

where the second inequality is based on Assumption 4.10 and 4.12, and the third inequality can be further expanded as

$$
\begin{aligned}
&\Big\| Z_{j*}(\widetilde{\boldsymbol{X}}, \boldsymbol{W}_0, \ldots, \boldsymbol{W}_{L-2}) - Z_{j*}(\widetilde{\boldsymbol{X}}', \boldsymbol{W}_0, \ldots, \boldsymbol{W}_{L-2}) \Big\|_\infty \\
&\leq \Big\| \sigma_{L-2} \Big[ \Big( (1-\alpha) \sum_{j=1}^n [g(\boldsymbol{A})]_{ij} Z_{j*}(\widetilde{\boldsymbol{X}}, \boldsymbol{W}_0, \ldots, \boldsymbol{W}_{L-3}) + \alpha Z_{j*}(\widetilde{\boldsymbol{X}}, \boldsymbol{W}_0) \Big) \psi(\boldsymbol{W}_{L-2}) \Big] \\
&\qquad - \sigma_{L-1} \Big[ \Big( (1-\alpha) \sum_{j=1}^n [g(\boldsymbol{A})]_{ij} Z_{j*}(\widetilde{\boldsymbol{X}}', \boldsymbol{W}_0, \ldots, \boldsymbol{W}_{L-3}) + \alpha Z_{j*}(\widetilde{\boldsymbol{X}}, \boldsymbol{W}_0) \Big) \psi(\boldsymbol{W}_{L-2}) \Big] \Big\|_\infty \\
&\leq \rho_{L-2} \tilde{w}_{L-2} \Big[ (1-\alpha) g \Big\| Z_{j*}(\widetilde{\boldsymbol{X}}, \boldsymbol{W}_0, \ldots, \boldsymbol{W}_{L-3}) - Z_{j*}(\widetilde{\boldsymbol{X}}', \boldsymbol{W}_0, \ldots, \boldsymbol{W}_{L-3}) \Big\|_\infty \\
&\qquad + \alpha \big\| Z_{j*}(\widetilde{\boldsymbol{X}}, \boldsymbol{W}_0) - Z_{j*}(\widetilde{\boldsymbol{X}}', \boldsymbol{W}_0) \big\|_\infty \Big].
\end{aligned}
$$

Substituting the inequality back can obtain the final results as

$$
\begin{aligned}
&|h(\boldsymbol{z}, \boldsymbol{\delta}) - h(\boldsymbol{z}, \boldsymbol{\delta}')| \\
&\leq C_\ell \Big[ \prod_{t=L-2}^L \rho_t \tilde{w}_t (1-\alpha)^2 g^2 \Big\| Z_{j*}(\widetilde{\boldsymbol{X}}, \boldsymbol{W}_0, \ldots, \boldsymbol{W}_{L-3}) - Z_{j*}(\widetilde{\boldsymbol{X}}', \boldsymbol{W}_0, \ldots, \boldsymbol{W}_{L-3}) \Big\|_\infty \\
&\quad + \prod_{t=L-2}^L \rho_t \tilde{w}_t (1-\alpha) g \alpha \big\| Z_{j*}(\widetilde{\boldsymbol{X}}, \boldsymbol{W}_0) - Z_{j*}(\widetilde{\boldsymbol{X}}', \boldsymbol{W}_0) \big\|_\infty + \prod_{t=L-1}^L \rho_t \tilde{w}_t \alpha \big\| Z_{j*}(\widetilde{\boldsymbol{X}}, \boldsymbol{W}_0) - Z_{j*}(\widetilde{\boldsymbol{X}}', \boldsymbol{W}_0) \big\|_\infty \Big] \\
&\leq C_\ell \Big[ \prod_{t=1}^L \rho_t \tilde{w}_t (1-\alpha)^{L-1} g^{L-1} + \alpha \sum_{j=1}^{L-1} (1-\alpha)^{j-1} g^{j-1} \prod_{t=L-j}^L \rho_t \tilde{w}_t \Big] \big\| Z_{j*}(\widetilde{\boldsymbol{X}}, \boldsymbol{W}_0) - Z_{j*}(\widetilde{\boldsymbol{X}}', \boldsymbol{W}_0) \big\|_\infty \\
&\leq C_\ell \Big[ \prod_{t=0}^L \rho_t \tilde{w}_t (1-\alpha)^{L-1} g^{L-1} + \alpha \rho_0 \tilde{w}_0 \sum_{j=1}^{L-1} (1-\alpha)^{j-1} g^{j-1} \prod_{t=L-j}^L \rho_t \tilde{w}_t \Big] \| \boldsymbol{\delta} - \boldsymbol{\delta}' \|_\infty \\
&\leq C_\ell \rho_0 \tilde{w}_0 \sum_{j=1}^L (1-\alpha)^{j-1} g^{j-1} \prod_{t=L-j}^L \rho_t \tilde{w}_t \| \boldsymbol{\delta} - \boldsymbol{\delta}' \|_\infty \\
&\leq K_{\text{GCNII}} \| \boldsymbol{\delta} - \boldsymbol{\delta}' \|_\infty,
\end{aligned}
$$

where the third inequality is due to that

$$
\big\| Z_{j*}(\widetilde{\boldsymbol{X}}, \boldsymbol{W}_0) - Z_{j*}(\widetilde{\boldsymbol{X}}', \boldsymbol{W}_0) \big\|_\infty \leq \rho_0 \tilde{w}_0 \| \widetilde{X} - \widetilde{X}' \| \leq \rho_0 \tilde{w}_0 \| \boldsymbol{\delta} - \boldsymbol{\delta}' \|_\infty,
$$

and $\tilde{w}_t$ is denoted by

$$
\| \psi(\boldsymbol{W}_t) \|_\sigma = \| (1-\beta) \boldsymbol{I}_n + \beta \boldsymbol{W}_t \|_\sigma \leq 1 - \beta + \beta w_t.
$$

Notably, we set $\beta = 1$ in $\tilde{w}_0$ and $\tilde{w}_L$. Then we turn to the covering number of $\widehat{\mathcal{F}}$. Let $\Delta_l = \|Z_{i*}(\widehat{\boldsymbol{X}}_c, \boldsymbol{W}_0, \ldots, \boldsymbol{W}_l) - Z_{i*}(\widehat{\boldsymbol{X}}_c, \boldsymbol{W}_0, \ldots, \boldsymbol{W}_l)\|_\infty$. For brevity, let $A_p = \|Z_{i*}(\widehat{\boldsymbol{X}}_c, \boldsymbol{W}_0, \ldots, \boldsymbol{W}_p)\|_\infty$, and $B_p^q = \|Z_{i*}(\widehat{\boldsymbol{X}}_c, \boldsymbol{W}_0, \ldots, \boldsymbol{W}_q, \ldots, \boldsymbol{W}_p) - Z_{i*}(\widehat{\boldsymbol{X}}_c, \boldsymbol{W}_0, \ldots, \boldsymbol{W}_q', \ldots, \boldsymbol{W}_p)\|_\infty$, where $q \leq p$. We make some preparation first to facilitate the further derivations

$$
\begin{aligned}
A_p =& \|Z_{j*}(\widehat{\boldsymbol{X}}_c, \boldsymbol{W}_0, \ldots, \boldsymbol{W}_p)\|_\infty \\
=& \left\| \sigma_p \Big[ \Big( (1-\alpha) \sum_{j=1}^n [g(\boldsymbol{A})]_{ij} Z_{j*}(\widehat{\boldsymbol{X}}_c, \boldsymbol{W}_0, \ldots, \boldsymbol{W}_{p-1}) + \alpha Z_{j*}(\widehat{\boldsymbol{X}}_c, \boldsymbol{W}_0) \Big) \psi(\boldsymbol{W}_p) \Big] \right\|_\infty \\
\leq& \rho_p \tilde{w}_p (1-\alpha) g \|Z_{j*}(\widehat{\boldsymbol{X}}_c, \boldsymbol{W}_0, \ldots, \boldsymbol{W}_{p-1})\|_\infty + \rho_p \tilde{w}_p \alpha \|Z_{j*}(\widehat{\boldsymbol{X}}_c, \boldsymbol{W}_0)\|_\infty \\
\leq& \rho_p \tilde{w}_p \rho_{p-1} \tilde{w}_{p-1} (1-\alpha)^2 g^2 \|Z_{j*}(\widehat{\boldsymbol{X}}_c, \boldsymbol{W}_0, \ldots, \boldsymbol{W}_{p-2})\|_\infty + \rho_p \tilde{w}_p \rho_{p-1} \tilde{w}_{p-1} (1-\alpha) g \|Z_{j*}(\widehat{\boldsymbol{X}}_c, \boldsymbol{W}_0)\|_\infty \\
& + \rho_p \tilde{w}_p \|Z_{j*}(\widehat{\boldsymbol{X}}_c, \boldsymbol{W}_0)\|_\infty \\
\leq& ((1-\alpha)g)^p \prod_{t=1}^p \rho_t \tilde{w}_t \|Z_{j*}(\widehat{\boldsymbol{X}}_c, \boldsymbol{W}_0)\| + \sum_{j=0}^{p-1} ((1-\alpha)g)^j \prod_{t=p-j}^p \rho_t \tilde{w}_t \|Z_{j*}(\widehat{\boldsymbol{X}}_c, \boldsymbol{W}_0)\|_\infty \\
\leq& \hat{b} \rho_0 \tilde{w}_0 \sum_{j=0}^p ((1-\alpha)g)^j \prod_{t=p-j}^p \rho_t \tilde{w}_t,
\end{aligned}
$$

where the last inequality is according to $\|Z_{j*}(\widehat{\boldsymbol{X}}_c, \boldsymbol{W}_0)\|_\infty \leq \rho_0 \tilde{w}_0 \|\widehat{\boldsymbol{X}}_c\| \leq \hat{b} \rho_0 \tilde{w}_0$. Then, we repeat the proof procedure, which starts with

$$
\begin{aligned}
\Delta_1 =& \|Z_{j*}(\widehat{\boldsymbol{X}}_c, \boldsymbol{W}_0, \boldsymbol{W}_1) - Z_{j*}(\widehat{\boldsymbol{X}}_c, \boldsymbol{W}_0', \boldsymbol{W}_1')\|_\infty \\
\leq& \|Z_{j*}(\widehat{\boldsymbol{X}}_c, \boldsymbol{W}_0, \boldsymbol{W}_1) - Z_{j*}(\widehat{\boldsymbol{X}}_c, \boldsymbol{W}_0, \boldsymbol{W}_1') + Z_{j*}(\widehat{\boldsymbol{X}}_c, \boldsymbol{W}_0, \boldsymbol{W}_1') - Z_{j*}(\widehat{\boldsymbol{X}}_c, \boldsymbol{W}_0', \boldsymbol{W}_1')\|_\infty \\
=& B_1^1 + B_1^0 \\
\leq& \rho_1 \Big[ \Big\| (1-\alpha) \sum_{j=1}^n [g(\boldsymbol{A})]_{ij} Z_{j*}(\widehat{\boldsymbol{X}}_c, \boldsymbol{W}_0) + \alpha Z_{j*}(\widehat{\boldsymbol{X}}_c, \boldsymbol{W}_0) \Big\|_\infty \|\psi(\boldsymbol{W}_1) - \psi(\boldsymbol{W}_1')\|_\infty + \tilde{w}_1 \Big( \Big\| ((1-\alpha) \\
& \sum_{j=1}^n [g(\boldsymbol{A})]_{ij} Z_{j*}(\widehat{\boldsymbol{X}}_c, \boldsymbol{W}_0) + \alpha Z_{j*}(\widehat{\boldsymbol{X}}_c, \boldsymbol{W}_0)) - ((1-\alpha) \sum_{j=1}^n [g(\boldsymbol{A})]_{ij} Z_{j*}(\widehat{\boldsymbol{X}}_c, \boldsymbol{W}_0') + \alpha Z_{j*}(\widehat{\boldsymbol{X}}_c, \boldsymbol{W}_0')) \Big\|_\infty \Big) \Big] \\
\leq& \rho_1 \Big[ ((1-\alpha)g A_0 + \alpha A_0) \|\psi(\boldsymbol{W}_1) - \psi(\boldsymbol{W}_1')\|_\infty + \tilde{w}_1 ((1-\alpha)g \Delta_0 + \alpha B_0^0) \Big] \\
\leq& \rho_1 \Big[ ((1-\alpha)g A_0 + \alpha A_0) \beta \|\boldsymbol{W}_1 - \boldsymbol{W}_1'\|_\infty + \tilde{w}_1 ((1-\alpha)g \Delta_0 + \alpha B_0^0) \Big].
\end{aligned}
$$

where the last inequality is due to $\|\psi(\boldsymbol{W}_1) - \psi(\boldsymbol{W}_1')\|_\infty \leq \|\beta_1 \boldsymbol{W}_1 - \beta_1 \boldsymbol{W}_1'\|_\infty$. Next, we derive that

$$
\begin{aligned}
\Delta_2 =& \|Z_{j*}(\widehat{\boldsymbol{X}}_c, \boldsymbol{W}_0, \boldsymbol{W}_1, \boldsymbol{W}_2) - Z_{j*}(\widehat{\boldsymbol{X}}_c, \boldsymbol{W}_0', \boldsymbol{W}_1', \boldsymbol{W}_2')\|_\infty = \sum_{t=0}^2 B_2^t \\
\leq& \rho_2 \Big[ \Big\| (1-\alpha) \sum_{j=1}^n [g(\boldsymbol{A})]_{ij} Z_{j*}(\widehat{\boldsymbol{X}}_c, \boldsymbol{W}_0, \boldsymbol{W}_1) + \alpha Z_{j*}(\widehat{\boldsymbol{X}}_c, \boldsymbol{W}_0) \Big\|_\infty \|\psi(\boldsymbol{W}_2) - \psi(\boldsymbol{W}_2')\|_\infty \\
& + \tilde{w}_2 \Big\| (1-\alpha) \sum_{j=1}^n [g(\boldsymbol{A})]_{ij} Z_{j*}(\widehat{\boldsymbol{X}}_c, \boldsymbol{W}_0, \boldsymbol{W}_1) - (1-\alpha) \sum_{j=1}^n [g(\boldsymbol{A})]_{ij} Z_{j*}(\widehat{\boldsymbol{X}}_c, \boldsymbol{W}_0, \boldsymbol{W}_1') \Big\|_\infty \\
& + \tilde{w}_2 \Big\| (1-\alpha) \sum_{j=1}^n [g(\boldsymbol{A})]_{ij} Z_{j*}(\widehat{\boldsymbol{X}}_c, \boldsymbol{W}_0, \boldsymbol{W}_1') - (1-\alpha) \sum_{j=1}^n [g(\boldsymbol{A})]_{ij} Z_{j*}(\widehat{\boldsymbol{X}}_c, \boldsymbol{W}_0', \boldsymbol{W}_1') \Big\|_\infty \\
& + \alpha \Big\| Z_{j*}(\widehat{\boldsymbol{X}}_c, \boldsymbol{W}_0) - Z_{j*}(\widehat{\boldsymbol{X}}_c, \boldsymbol{W}_0') \Big\|_\infty \Big] \\
\leq& \rho_2 \Big[ ((1-\alpha)g A_1 + \alpha A_0) \|\psi(\boldsymbol{W}_2) - \psi(\boldsymbol{W}_2')\|_\infty + \tilde{w}_2 (1-\alpha)g (\Delta_1 - B_1^0) + \tilde{w}_2 ((1-\alpha)g B_1^0 + \alpha B_0^0) \Big] \\
\leq& \rho_2 \Big[ ((1-\alpha)g A_1 + \alpha A_0) \beta \|\boldsymbol{W}_2 - \boldsymbol{W}_2'\|_\infty + \tilde{w}_2 ((1-\alpha)g \Delta_1 + \alpha B_0^0) \Big].
\end{aligned}
$$

For the inductive step, the following inequalities hold with the fact that $\Delta_0 = B_0^0$

$$
\begin{aligned}
\Delta_{l+1} &= \sum_{t=0}^{l+1} B_{l+1}^t \\
&\leq \rho_{l+1}\Big[\big((1-\alpha)gA_l + \alpha A_0\big)\beta\|\boldsymbol{W}_{l+1} - \boldsymbol{W}'_{l+1}\|_\infty + \tilde{w}_{l+1}\big((1-\alpha)g\Delta_l + \alpha B_0^0\big)\Big] \\
&\leq \rho_{l+1}\big((1-\alpha)gA_l + \alpha A_0\big)\beta\|\boldsymbol{W}_{l+1} - \boldsymbol{W}'_{l+1}\|_\infty + \rho_{l+1}\rho_l\tilde{w}_{l+1}(1-\alpha)g\big((1-\alpha)gA_{l-1} + \alpha A_0\big)\beta\|\boldsymbol{W}_l - \boldsymbol{W}'_l\|_\infty \\
&\quad + \rho_{l+1}\rho_l\tilde{w}_{l+1}\tilde{w}_l(1-\alpha)g\Big((1-\alpha)g\Delta_{l-1} + \alpha B_0^0\Big) + \rho_{l+1}\tilde{w}_{l+1}\alpha B_0^0 \\
&\leq \rho_{l+1}\tilde{w}_{l+1}\big((1-\alpha)gA_l + \alpha A_0\big)\beta\frac{\|\boldsymbol{W}_{l+1} - \boldsymbol{W}'_{l+1}\|_\infty}{\tilde{w}_{l+1}} \\
&\quad + \rho_{l+1}\rho_l\tilde{w}_{l+1}\tilde{w}_l(1-\alpha)g\big[(1-\alpha_{l-1})gA_{l-1} + \alpha A_0\big)\beta\frac{\|\boldsymbol{W}_l - \boldsymbol{W}'_l\|_\infty}{\tilde{w}_l} \\
&\quad + \rho_{l+1}\rho_l\tilde{w}_{l+1}\tilde{w}_l(1-\alpha)^2g^2\Delta_{l-1} + \big(\rho_{l+1}\rho_l\tilde{w}_{l+1}\tilde{w}_l(1-\alpha)g + \rho_{l+1}\tilde{w}_{l+1}\big)\alpha B_0^0 \\
&\leq \sum_{j=1}^{l+1}\big((1-\alpha)g\big)^{l+1-j}\big((1-\alpha)gA_{j-1} + \alpha A_0\big)\prod_{t=j}^{l+1}\rho_t\tilde{w}_t\beta\frac{\|\boldsymbol{W}_j - \boldsymbol{W}'_j\|_\infty}{\tilde{w}_j} + \big((1-\alpha)g\big)^{l+1}\prod_{t=1}^{l+1}\rho_t\tilde{w}_t\Delta_0 \\
&\quad + \sum_{j=1}^{l+1}\big((1-\alpha)g\big)^{l+1-j}\prod_{t=j}^{l+1}\rho_t\tilde{w}_t\alpha B_0^0 \\
&\leq \sum_{j=0}^{l+1}\big((1-\alpha)g\big)^{l+1-j}\big((1-\alpha)gA_{j-1} + \alpha A_0\big)\prod_{t=j}^{l+1}\rho_t\tilde{w}_t\beta\frac{\|\boldsymbol{W}_j - \boldsymbol{W}'_j\|_\infty}{\tilde{w}_j} + \sum_{j=1}^{l+1}\big((1-\alpha)g\big)^{l+1-j}\prod_{t=j}^{l+1}\rho_t\tilde{w}_t\alpha B_0^0 \\
&\leq 2\hat{b}\sum_{j=0}^{l+1}\big((1-\alpha)g\big)^{l+1-j}\big((1-\alpha)gA_{j-1} + \alpha A_0\big)\prod_{t=j}^{l+1}\rho_t\tilde{w}_t\beta\frac{\|\boldsymbol{W}_j - \boldsymbol{W}'_j\|_\infty}{\tilde{w}_j}.
\end{aligned}
$$

Finally, we can obtain

$$
\begin{aligned}
&\|f_i(\boldsymbol{A}, \widehat{\boldsymbol{X}}_c, \boldsymbol{W}) - f_i(\boldsymbol{A}, \widehat{\boldsymbol{X}}_c, \boldsymbol{W})'\|_\infty \\
&= \|\sigma_L(Z_{i*}(\widehat{\boldsymbol{X}}_c, \boldsymbol{W}_0, \ldots, \boldsymbol{W}_{L-1})\boldsymbol{W}_L) - \sigma_L(Z_{i*}(\widehat{\boldsymbol{X}}_c, \boldsymbol{W}'_0, \ldots, \boldsymbol{W}'_{L-1})\boldsymbol{W}'_L)\|_\infty \\
&\leq \rho_L(\tilde{w}_L\Delta_{L-1} + A_{L-1}\|\boldsymbol{W}_L - \boldsymbol{W}'_L\|_\infty) \\
&\leq \rho_L\tilde{w}_L\Big(2\hat{b}\sum_{j=0}^{L-1}\big((1-\alpha)g\big)^{L-j}\big((1-\alpha)gA_{j-1} + \alpha A_0\big)\prod_{t=j}^{L-1}\frac{\beta\|\boldsymbol{W}_j - \boldsymbol{W}'_j\|_\infty}{\tilde{w}_j}\Big) + A_{L-1}\rho_L\tilde{w}_L\frac{\|\boldsymbol{W}_L - \boldsymbol{W}'_L\|_\infty}{\tilde{w}_L} \\
&\leq 2\hat{b}\sum_{j=0}^{L}\big((1-\alpha)g\big)^{L-j}\big((1-\alpha)gA_{j-1} + \alpha A_0\big)\prod_{t=j}^{L}\rho_t\tilde{w}_t\frac{\beta\|\boldsymbol{W}_j - \boldsymbol{W}'_j\|_\infty}{\tilde{w}_j},
\end{aligned}
$$

Similarly, by defining that

$$
T_j = \sum_{j=0}^{L}\big((1-\alpha)g\big)^{L-j}\big((1-\alpha)gA_{j-1} + \alpha A_0\big),
$$

we have the relation between $\epsilon$ and $\epsilon_j$ as

$$
\epsilon \leq 2\hat{b}\beta\sum_{j=0}^{L}g^{L-j}\big((1-\alpha)gA_{j-1} + \alpha A_0\big)\prod_{t=j}^{L}\rho_t\tilde{w}_t\frac{\epsilon_j}{\tilde{w}_j} = 2\hat{b}\beta\sum_{j=1}^{L}T_j\prod_{t=j}^{L}\rho_t\tilde{w}_t\frac{\epsilon_j}{\tilde{w}_j}. \tag{10}
$$

Then, the following inequalities hold

$$\int_{\frac{\mu}{2C_\ell}}^{\frac{q\sqrt{m+u}}{C_\ell}} \sqrt{\log\Big(\frac{3\theta K_{\mathrm{GCNII}}}{\epsilon}\Big)^d \mathcal{N}(\widehat{\mathcal{F}}, \epsilon, \|\cdot\|_\infty, S)} d\epsilon$$

$$\leq \sqrt{d}\int_{\frac{\mu}{2C_\ell}}^{\frac{q\sqrt{m+u}}{C_\ell}} \sqrt{\Big(\log\frac{3\theta K_{\mathrm{GCNII}}}{\epsilon}\Big)} d\epsilon + \sum_{i=1}^{L}\int_{\frac{\mu}{2C_\ell}}^{\frac{q\sqrt{m+u}}{C_\ell}} \sqrt{v^2\log\frac{3s_i}{\epsilon_i}} d\epsilon$$

$$\leq 3\sqrt{d}\theta K_{\mathrm{GCNII}} + \sum_{i=1}^{L}\int_{\frac{\mu}{2C_\ell}}^{\frac{q\sqrt{m+u}}{C_\ell}} \sqrt{v^2\log\frac{3s_i}{\epsilon_i}} d\epsilon$$

$$\leq 3\sqrt{d}\theta K_{\mathrm{GCNII}} + \sum_{i=1}^{L} v\int_{\frac{\mu}{2C_\ell}}^{\frac{q\sqrt{m+u}}{C_\ell}} \sqrt{\log\frac{3s_i}{\epsilon_i}} d\epsilon_i \frac{\hat{b}\beta\sum_{j=1}^{L}T_j\prod_{t=j}^{L}\rho_t\tilde{w}_t}{\tilde{w}_j}$$

$$\leq 3\sqrt{d}\theta K_{\mathrm{GCNII}} + \sum_{i=1}^{L}\frac{v\hat{b}\beta\sum_{j=1}^{L}T_j\prod_{t=j}^{L}\rho_t\tilde{w}_t}{\tilde{w}_j}\int_{\frac{\mu}{2C_\ell}}^{\frac{q\sqrt{m+u}}{C_\ell}} \sqrt{\log\frac{3s_i}{\epsilon_i}} d\epsilon_i$$

$$\leq 3\Big(\sqrt{d}\theta K_{\mathrm{GCNII}} + v\hat{b}\beta\sum_{j=1}^{L}T_j\prod_{t=j}^{L}\rho_t\tilde{w}_t\int_{0}^{1/3} \sqrt{\log\frac{1}{\epsilon_i}} d\epsilon_i\Big)$$

$$\leq 3(\sqrt{d}\theta K_{\mathrm{GCNII}} + v\hat{b}\beta\sum_{j=1}^{L}T_j\prod_{t=j}^{L}\rho_t\tilde{w}_t).$$

where the third inequality is due to Equation (10). Finally, the generalization gap holds with $\mu = \frac{1}{n}$

$$\widetilde{R}_u(f) \leq \widetilde{R}_m(f) + \frac{4\mu}{\sqrt{m+u}} + \frac{72C_\ell}{\sqrt{m+u}}\Big(\sqrt{d}\theta K_{\mathrm{GCNII}} + v\hat{b}\beta\sum_{j=1}^{L}T_j\prod_{t=j}^{L}\rho_t\tilde{w}_t\Big) + qc_0Q_1\sqrt{\min(m,u)} + 2q\sqrt{\frac{Q_1Q_2}{2}\ln\frac{1}{\delta}}$$

$$\leq \widetilde{R}_m(f) + \frac{4}{m+u} + \frac{72C_\ell}{\sqrt{m+u}}\Big(\sqrt{d}\theta K_{\mathrm{GCNII}} + v\hat{b}\beta\sum_{j=1}^{L}T_j\prod_{t=j}^{L}\rho_t\tilde{w}_t\Big) + qc_0Q_1\sqrt{\min(m,u)} + 2q\sqrt{\frac{Q_1Q_2}{2}\ln\frac{1}{\delta}}.$$

Now we complete the proofs of Proposition 4.18. $\qquad\square$

## C. Additional Lemmas

**Lemma C.1.** *(Xiao et al., 2022) Let $g(\boldsymbol{w}, \boldsymbol{z})$ be the loss function and $h(\boldsymbol{w}, \boldsymbol{z}) = \max_{\|\boldsymbol{z}-\boldsymbol{z}'\|\leq\beta} g(\boldsymbol{w}, \boldsymbol{z}')$ be the adversarial loss. Assume that $\boldsymbol{w} \to g(\boldsymbol{w}, \boldsymbol{z})$ is $\|\cdot\|$-Lipschitz with constant L, we have*

$$\|h(\boldsymbol{w}, \boldsymbol{z}) - h(\boldsymbol{w}', \boldsymbol{z})\| \leq L\|\boldsymbol{w} - \boldsymbol{w}'\|.$$

**Lemma C.2.** *(Mohri et al., 2018) Let $\|\cdot\|$ be an arbitrary norm and $\mathcal{B}$ be a ball of radius $\eta$ in $\mathbb{R}^d$. Let $\mathcal{C}$ be a smallest possible $\xi$-cover of $\mathcal{B}$. Then,*

$$|\mathcal{C}| \leq \Big(\frac{3\eta}{\xi}\Big)^d.$$

**Lemma C.3.** *(El-Yaniv & Pechyony, 2009) Let $\mathcal{G}$ be a set of real-valued vectors in $[-q, q]^{m+u}$. Let $Q_1 = \frac{1}{u} + \frac{1}{m}$, $Q_2 = \frac{m+u}{(m+u-1/2)(10-1/2(\max(m,u)))}$ and $c_0 = \sqrt{\frac{32\log(4e)}{3}} < 5.05$. Then with probability of at least $1 - \delta$, for all $g \in \mathcal{G}$, we have*

$$R_u(g) \leq R_m(g) + \mathfrak{R}_{m+u}(\mathcal{G}, \frac{mu}{(m+u)^2}) + qc_0Q_1\sqrt{\min(m,u)} + 2q\sqrt{\frac{Q_1Q_2}{2}\ln\frac{1}{\delta}}.$$

**Lemma C.4.** *(Bartlett et al., 2017) Let $\mathcal{G}$ be a real-valued function class taking values in $[-q, q]$, where $q > 0$ is a constant, and assume that $\mathbf{0} \in \mathcal{G}$. Then the transductive Rademacher complexity of $\mathcal{G}$ can be bounded as*

$$\mathcal{R}_{m+u}(\mathcal{G}) \leq \inf_{\mu>0}\Big(\frac{4\mu}{\sqrt{n}} + \frac{12}{n}\int_{\mu}^{2q\sqrt{n}} \sqrt{\log\mathcal{N}(\mathcal{G}, \epsilon, \|\cdot\|_\infty, S)} d\epsilon\Big).$$

*Table 4.* Summary of adversarial generalization analysis ($m$-number of training samples; $u$-number of test samples; NNs-Neural Networks; $\star$-expectation bound; $\sharp$-optimization bound).

| Reference | Model | Learning mode | Analysis tool | Convergence rate |
|---|---|---|---|---|
| Xing et al. (2021) | Multi-layer NNs | Inductive | Uniform argument stability | $\star\mathcal{O}(1/m)$ |
| Xiao et al. (2022) | Multi-layer NNs | Inductive | Uniform stability | $\star\mathcal{O}(1/m)$ |
| Yin et al. (2020) | One-layer NN | Inductive | Rademacher complexity | $\mathcal{O}(1/\sqrt{m})$ |
| Awasthi et al. (2020) | Multi-layer NNs | Inductive | Rademacher complexity | $\mathcal{O}(1/\sqrt{m})$ |
| Tu,Zhang, and Tao (2019) | Multi-layer NNs | Inductive | Covering number | $\mathcal{O}(1/\sqrt{m})$ |
| Mustafa, Lei, and Kloft (2022) | Multi-layer NNs | Inductive | Covering number | $\sharp\mathcal{O}(1/m)$ |
| Ours | Multi-layer GNNs | Transductive | Covering number | $\mathcal{O}\big(\max\{\frac{1}{\sqrt{m}}, \frac{1}{\sqrt{u}}\}\big)$ |

## D. Additional Related Work.

**Adversarial generalization analysis.** As a powerful tool to explain generalization in adversarial training, classical learning-theoretic measures can provide encouraging inspiration in adversarial generalization analysis for GNNs. Xing et al. (2021) and Xiao et al. (2022) utilize algorithm stability to derive adversarial stability bounds for NN models, which are based on certain algorithms such as SGD ( stochastic gradient descent). Awasthi et al. (2020) and Yin et al. (2020) provide an adversarial generalization bound via the lens of Rademacher complexity with a restricted NN model structure. By using covering number, Tu, Zhang and Tao (2019) transform the adversarial expected risk over a distribution to the standard expected risk over a new distribution; Mustafa, Lei and Kloft (2022) approximate the complexity of the adversarial loss class by a finite discrete space. Table 4 summarizes the related works in adversarial learning using various techniques.

## E. Limitations

Though this paper does not involve topology attacks, given the similar adversarial generation settings for topology attacks and node attacks, our analytical framework could be expanded upon the topology attacks.

To be more specific, let the adversarial graph be generated from $\{\tilde{\mathbf{A}} : \|\tilde{\mathbf{A}}\| \leq \gamma\}$, where $\tilde{\mathbf{A}} = \mathbf{A} - \mathbf{A}'$ denotes the perturbation matrix added to the original adjacency matrix. The adversarial loss w.r.t. adversarial graph is defined by $\max_{\|\tilde{\mathbf{A}}\| \leq \gamma} \ell(f_i(\tilde{\mathbf{A}}, \mathbf{X}, \mathbf{W}), y_i)$. Analyzing analogously to the node attacks, we could measure the complexity of the adversarial loss function class by utilizing the covering number techniques, which is the main methodology developed in this paper (Lemma 4.6). This requires an additional assumption that the adversarial loss is $L_A$-Lipschitz continuous regarding the adjacency matrix $\mathbf{A}$, where the constant $L_A$ can be derived if given specific GNN models. Finally we can solve the measurement difficulty caused by the graph topology perturbations and apply it to our main results (Theorem 4.8). The remaining analysis will be left to future work.

## F. Additional Experiments.

This section provides the detailed experimental configuration and results for adversarial training on GNNs. Unless otherwise indicated, we adopt a two-layer GNN with the ELU activation in each layer and log-softmax activation for output, where the number of hidden units is fixed to 64. $\gamma$ and $K$ in APPNP are set as 0.5 and 10, respectively. $\alpha$ in GCNII is set as 0.1 and $\beta = \log(\xi/L + 1)$, where $L$ is the number of layers and $\xi$ is fixed to 1. Notably, the learning rate of CoraFull is set as 0.2 with a weight decay of 1e-4. The implement is GeForce RTX 3080 GPU.

Next, we provide the numerical discussion for the experimental results of the remained datasets from the main text, other influencing factors assessment (comparison of different GNNs and different feature dimensions), and other classical attack methods (FGSM (Goodfellow et al., 2015) and BIM (Kurakin et al., 2017)).

**Graph filter.** Based on the default settings, when the unnormalized filter $g(\mathbf{A}) = \mathbf{A} + \mathbf{I}$ is adopted, the learning rate $\eta$ is set by 0.01, where $\eta = 0.001$ for CS and $T = 400$ for CoraFull, and the aggregation hop $K$ of APPNP is set as 1. Figure 7-8 show the experimental results of the remaining datasets from the main text, Figure 9-11 present the results of adversarial training attacked by BIM, which exhibit a similar trend with PGD.

**Model architecture.** For a clearer comparison, we adopt the unnormalized filter, and the experimental setup is similar to the experiments of the graph filter. Figure 12 shows the results of hyper-parameter $\gamma$ in APPNP and $\alpha$ in GCNII of the remained

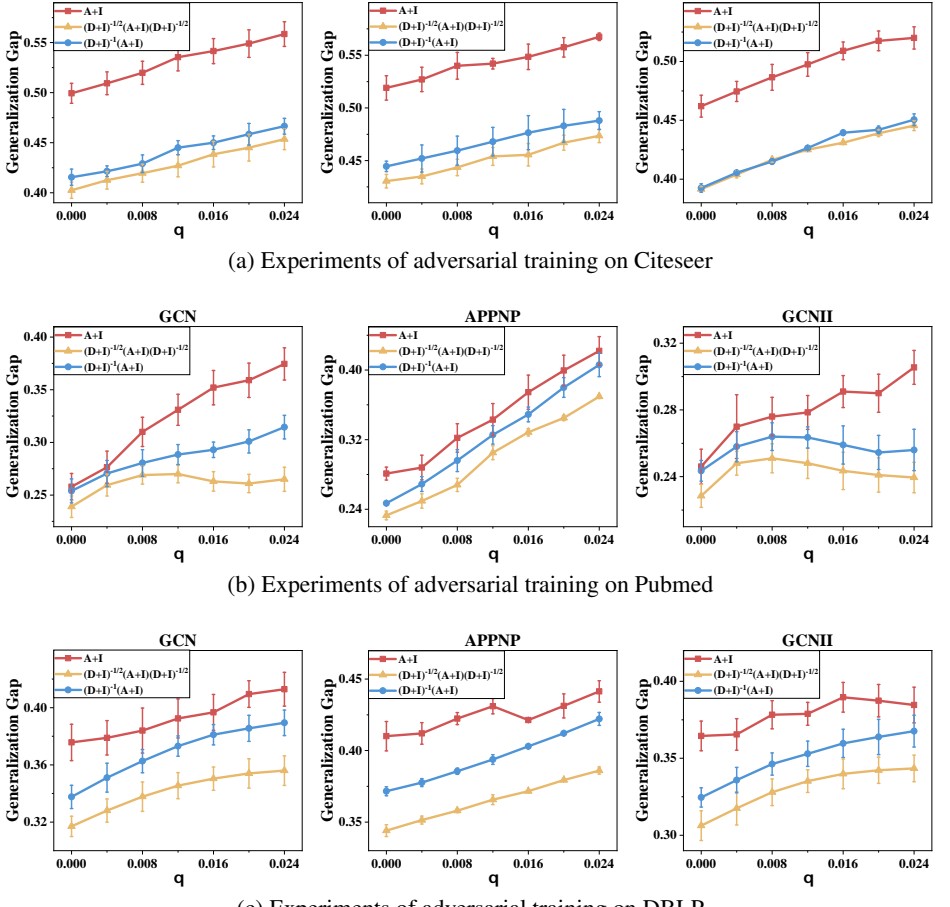

*Figure 7.* The generalization gap for different graph filters $g(\boldsymbol{A})$ with increased perturbations $\theta$ (attacked by PGD).

datasets from the main text. Moreover, Figure 13-14 show the results of adversarial training attacked by FGSM, which exhibit a similar trend with PGD.

**Model selection.** We compare the adversarial generalization gaps of three different GNN models. Figure 3 reveals that GCN has the largest generalization gap in its deep version, as GCNII and APPNP can improve their performance by their well-crafted architectures, especially by their optimizable parameters. Figure 16 and 17 present the results of adversarial training attacked by BIM and FGSM, respectively, indicating that our analysis has a wide range of applications to attack methods.

**Number of layers.** Based on the default settings, when the number of layers $L$ is bigger than 5, the iteration $T$ is set as 400 for CoraFull. Figure 18-19 display the experimental results of the remaining datasets from the main text, Figure 20-22 show the results of adversarial training attacked by BIM.

**Regularization parameter.** Figure 23-24 present the experimental results of the remaining datasets from the main text, Figure 25-27 show the results of adversarial training attacked by FGSM, which exhibit a similar trend with PGD.

**Feature dimension.** Based on the default settings, the number of hidden units is set as 128. Figure 28-29 present the experimental results of the remaining datasets from the main text, Figure 30-32 show the results of adversarial training attacked by BIM.

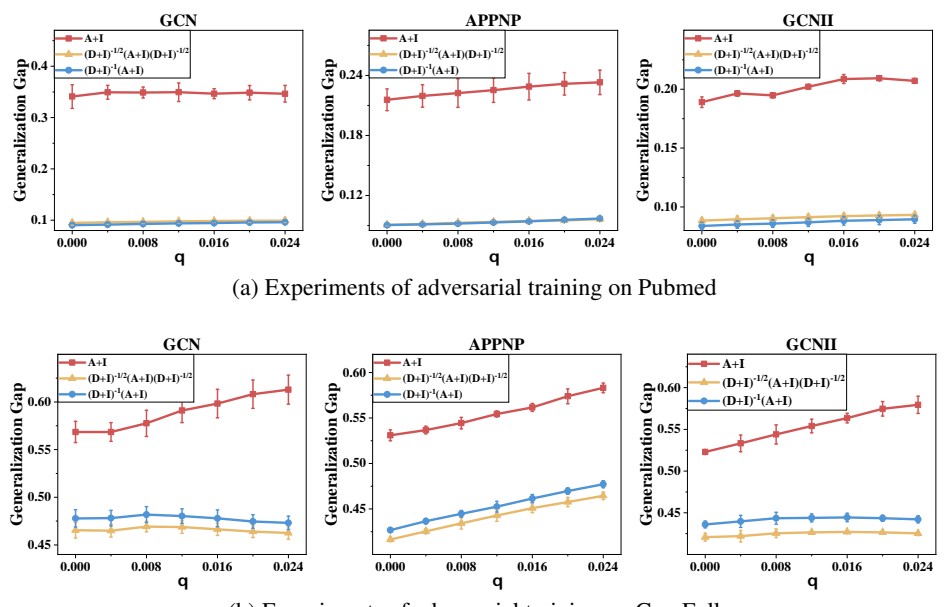

(a) Experiments of adversarial training on Pubmed

(b) Experiments of adversarial training on CoraFull

*Figure 8.* The generalization gap for different graph filters $g(\boldsymbol{A})$ with increased perturbations $\theta$ (attacked by PGD).

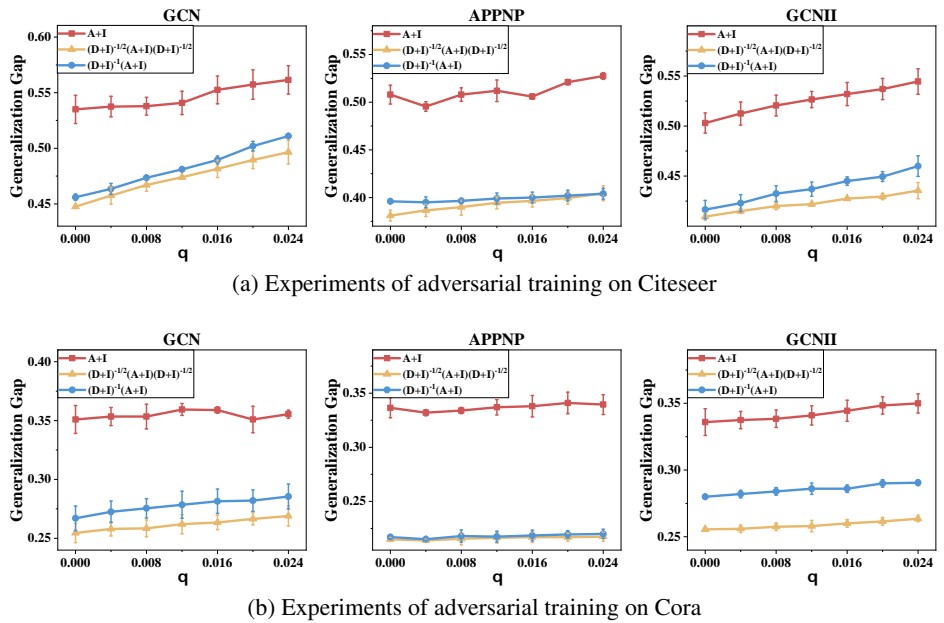

(a) Experiments of adversarial training on Citeseer

(b) Experiments of adversarial training on Cora

*Figure 9.* The generalization gap for different graph filters $g(\boldsymbol{A})$ with increased perturbations $\theta$ (attacked by BIM).

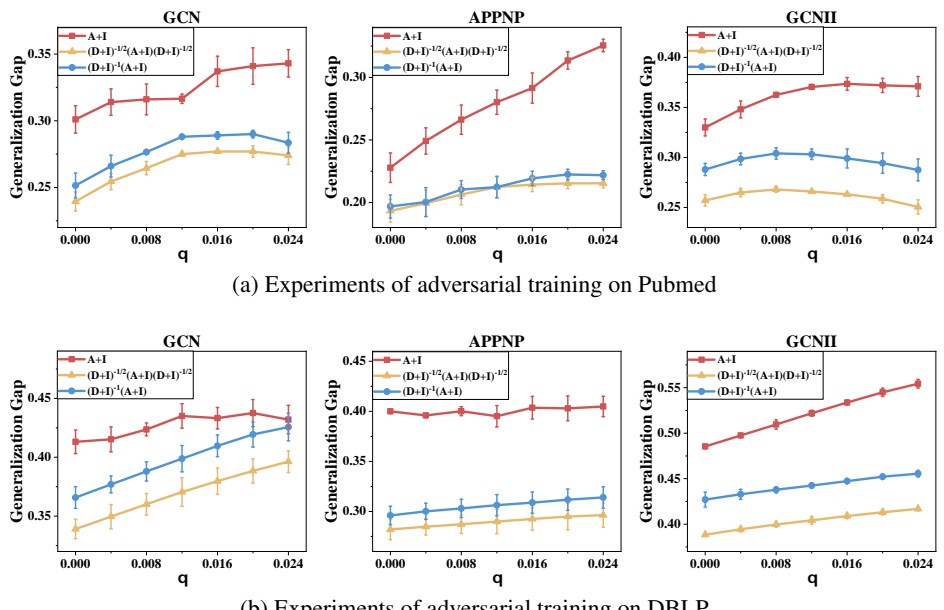

(a) Experiments of adversarial training on Pubmed

(b) Experiments of adversarial training on DBLP

*Figure 10.* The generalization gap for different graph filters $g(\boldsymbol{A})$ with increased perturbations $\theta$ (attacked by BIM).

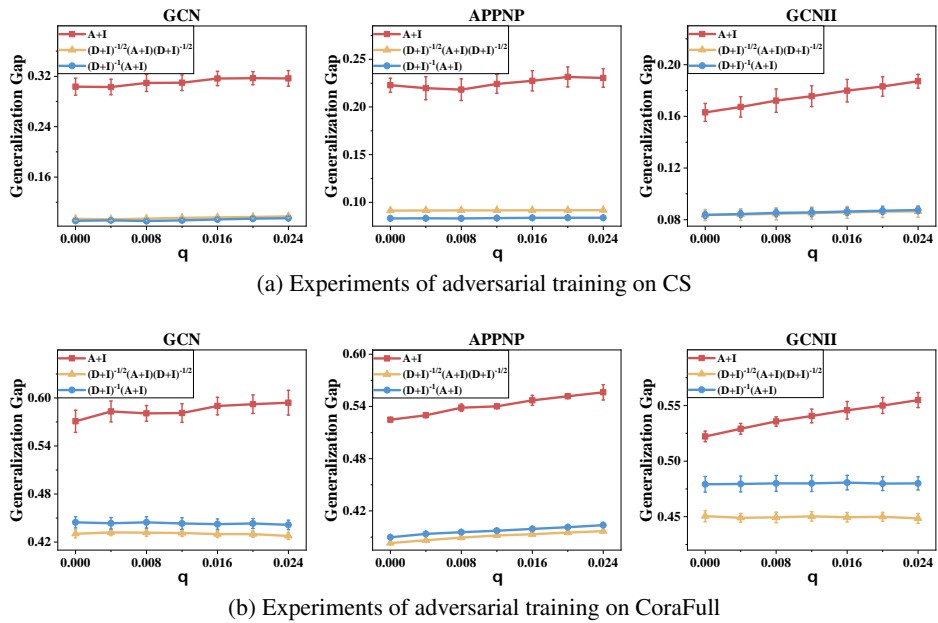

(a) Experiments of adversarial training on CS

(b) Experiments of adversarial training on CoraFull

*Figure 11.* The generalization gap for different graph filters $g(\boldsymbol{A})$ with increased perturbations $\theta$ (attacked by BIM).

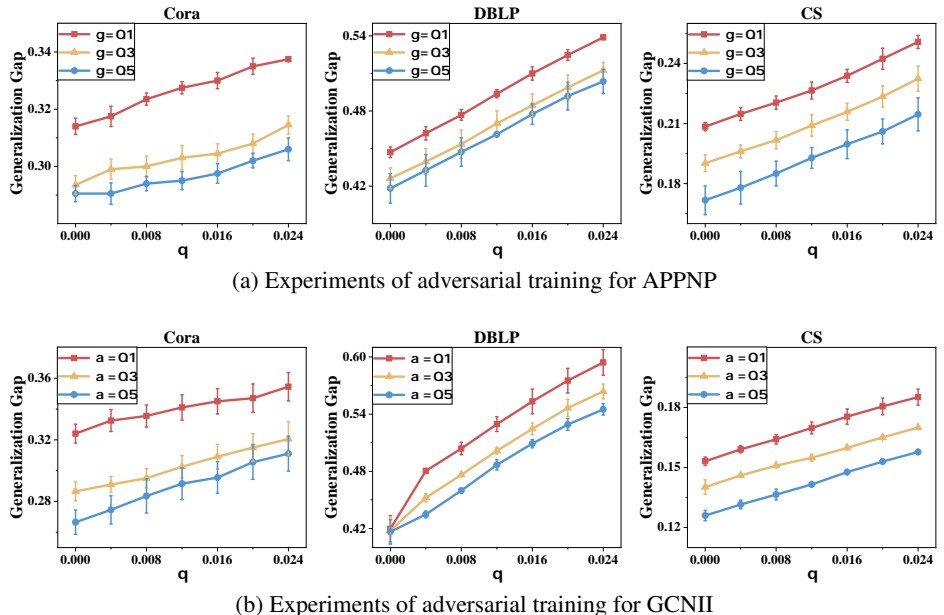

(a) Experiments of adversarial training for APPNP

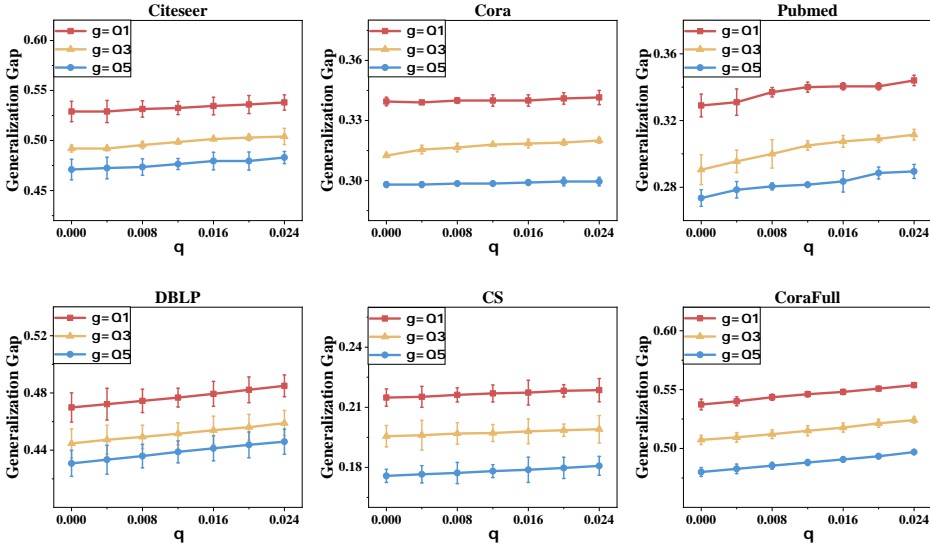

(b) Experiments of adversarial training for GCNII

*Figure 12.* The generalization gap for different hyper-parameter $\gamma$ in APPNP and $\alpha$ in GCNII with increased perturbations $\theta$ (attacked by PGD).

*Figure 13.* The generalization gap for different hyper-parameter $\gamma$ in APPNP with increased perturbations $\theta$ (attacked by FGSM).

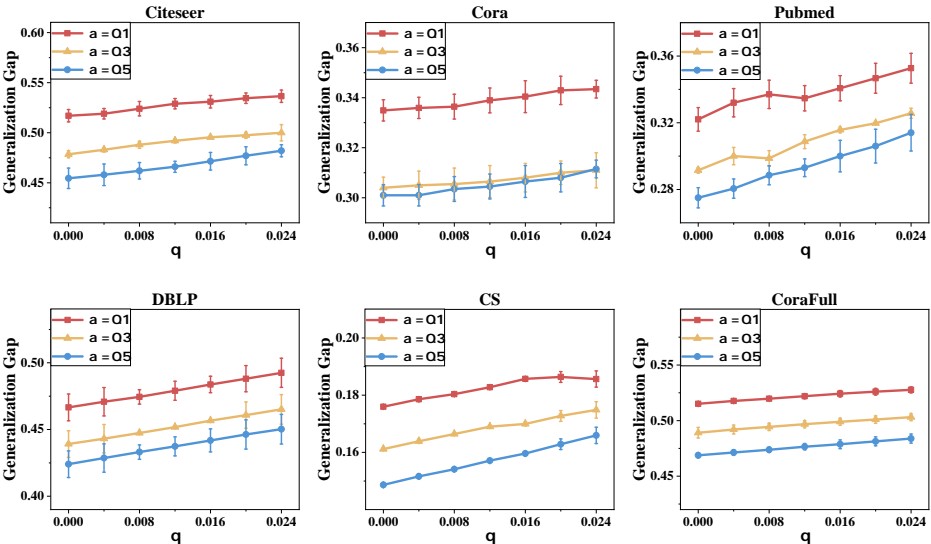

*Figure 14.* The generalization gap for different hyper-parameter $\alpha$ in GCNII with increased perturbations $\theta$ (attacked by FGSM).

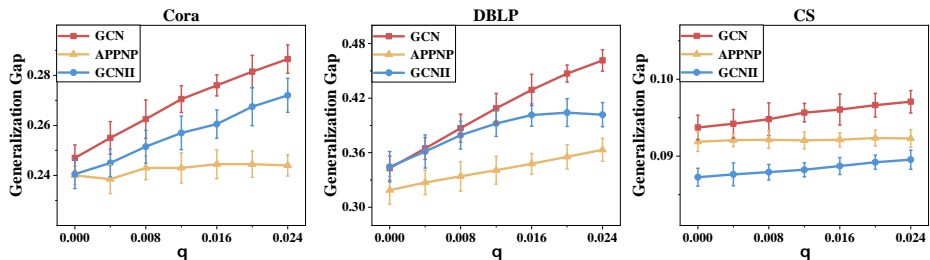

*Figure 15.* The generalization gap for different GNN models with increased perturbations $\theta$ (attacked by PGD).

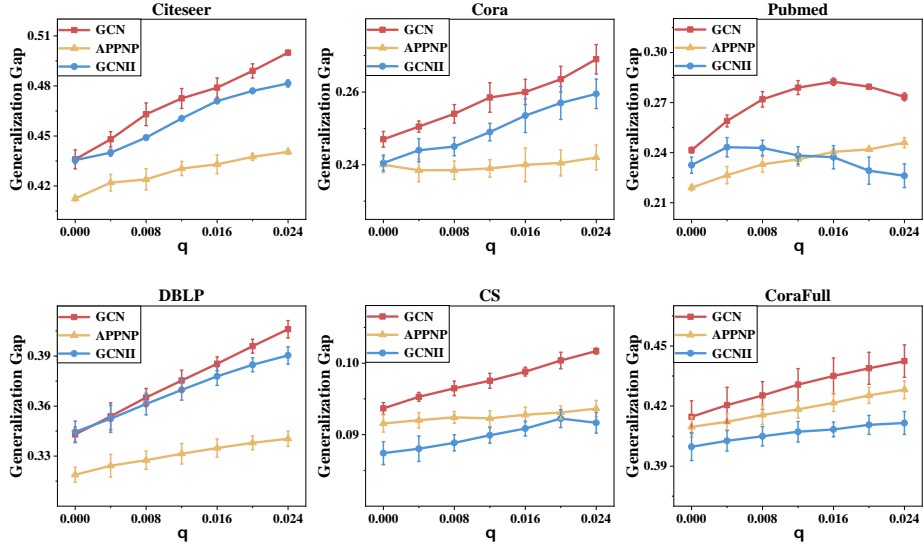

*Figure 16.* The generalization gap for different GNN models with increased perturbations $\theta$ (attacked by BIM).

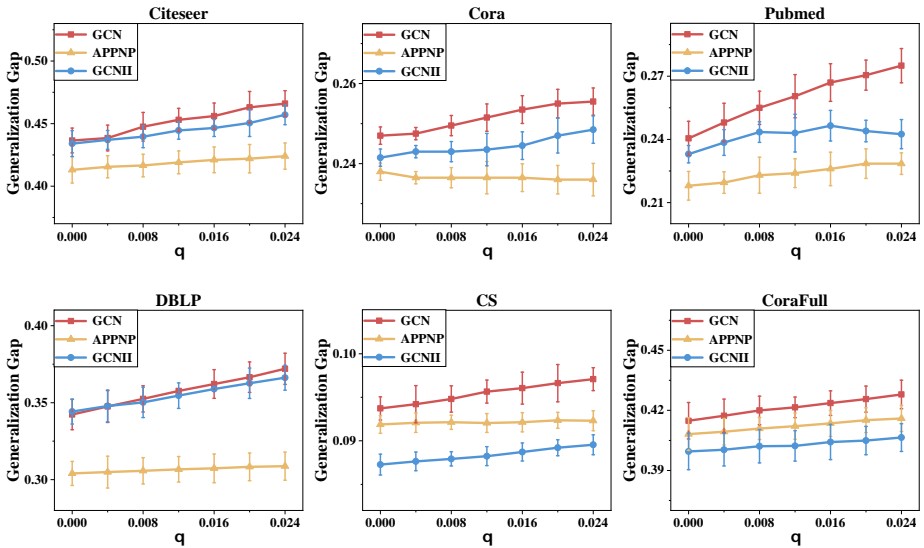

*Figure 17.* The generalization gap for different GNN models with increased perturbations $\theta$ (attacked by FGSM).

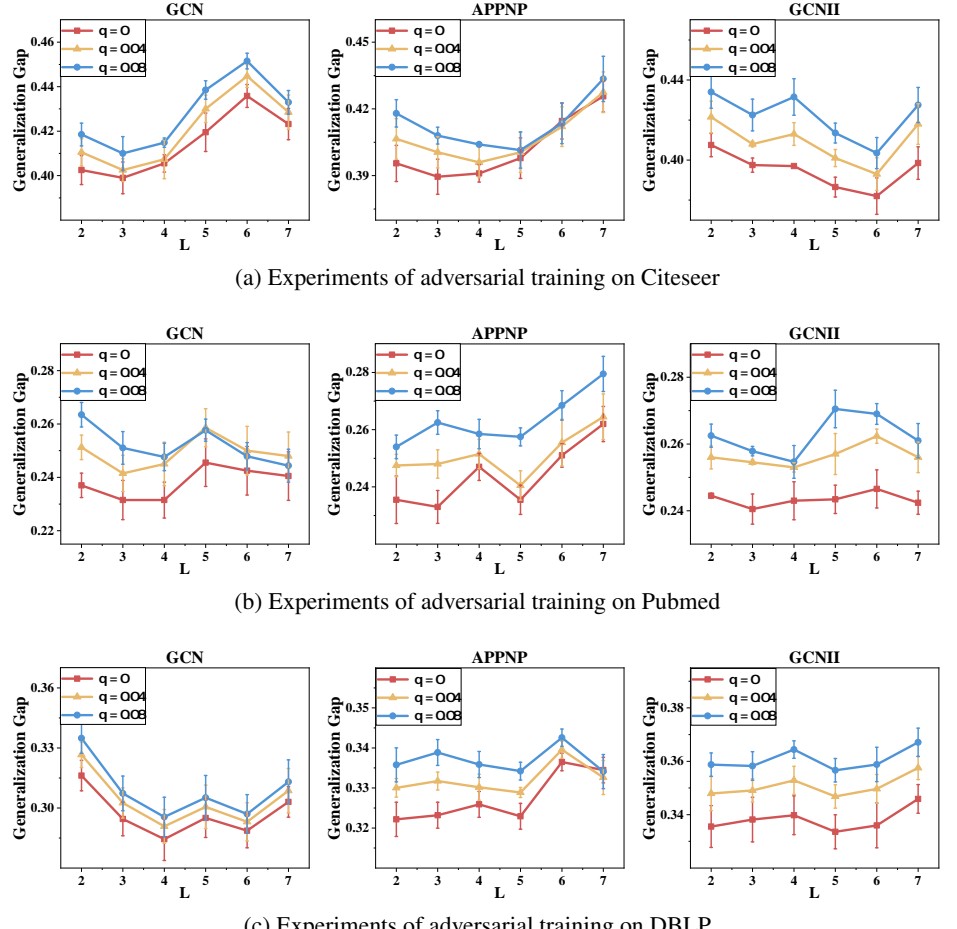

*Figure 18.* The generalization gap for different adversarial perturbations $\theta$ (attacked by PGD) with increased number of layers $L$.

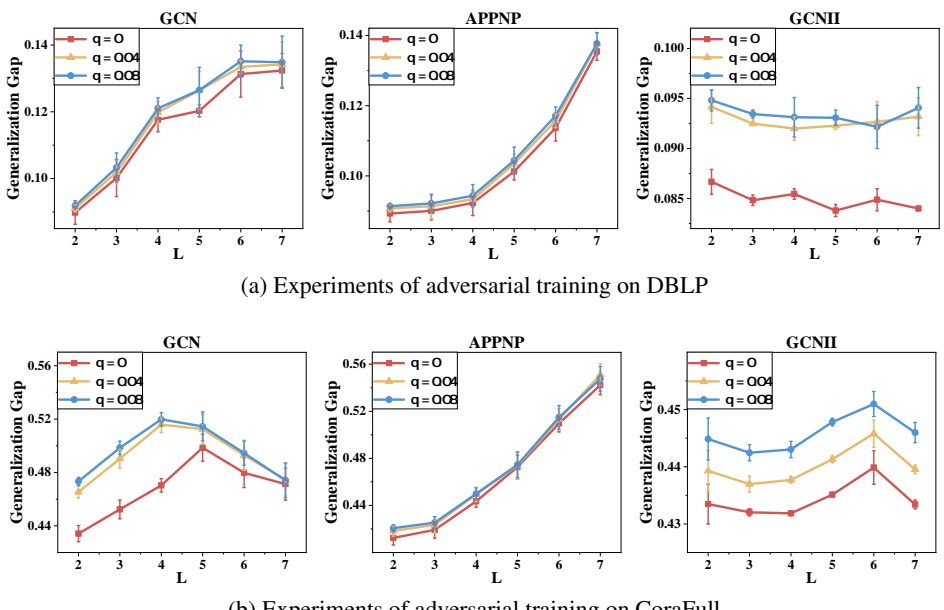

(a) Experiments of adversarial training on DBLP

(b) Experiments of adversarial training on CoraFull

*Figure 19.* The generalization gap for different adversarial perturbations $\theta$ (attacked by PGD) with increased number of layers $L$.

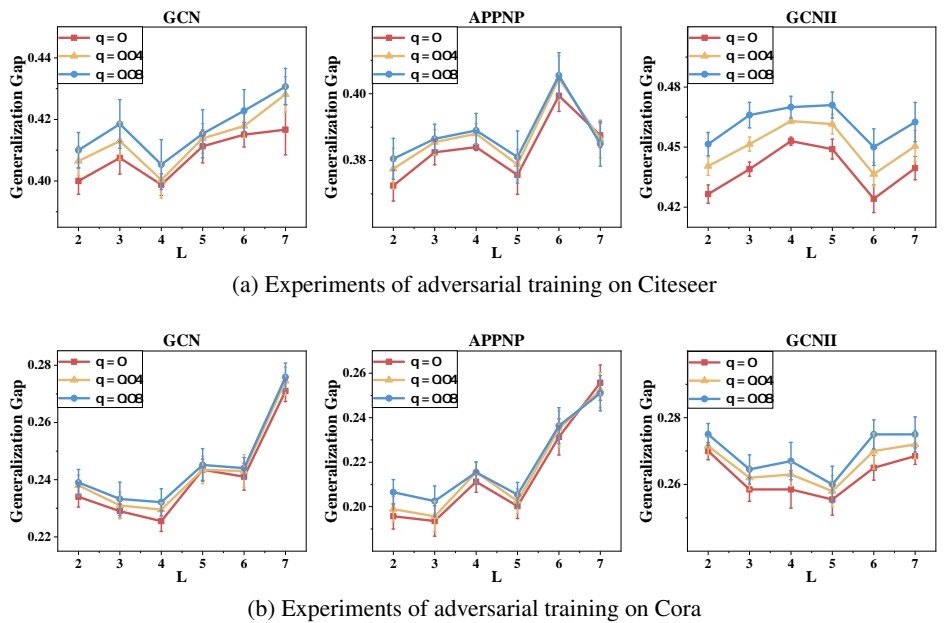

(a) Experiments of adversarial training on Citeseer

(b) Experiments of adversarial training on Cora

*Figure 20.* The generalization gap for different adversarial perturbations $\theta$ (attacked by BIM) with increased number of layers $L$.

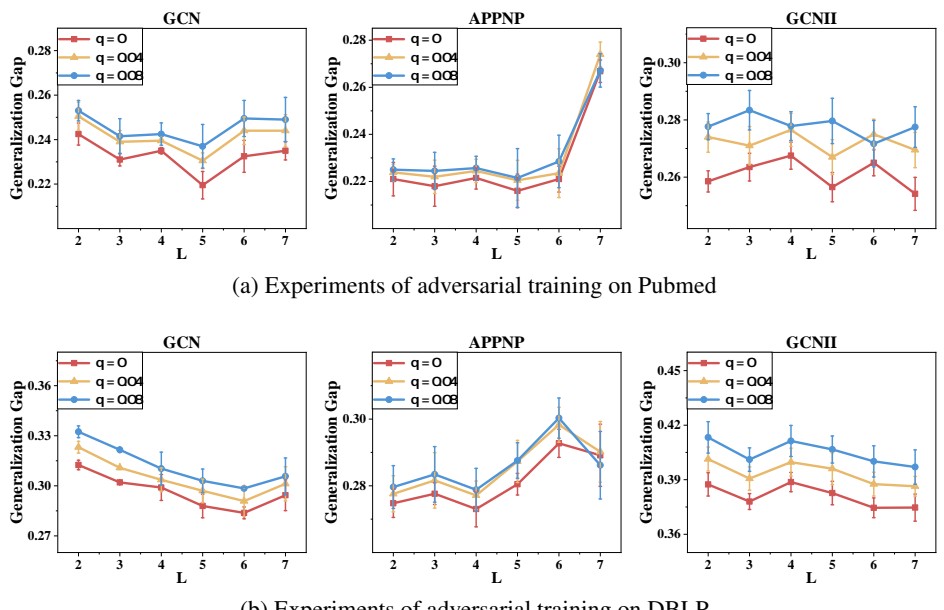

(a) Experiments of adversarial training on Pubmed

(b) Experiments of adversarial training on DBLP

*Figure 21.* The generalization gap for different adversarial perturbations $\theta$ (attacked by BIM) with increased number of layers $L$.

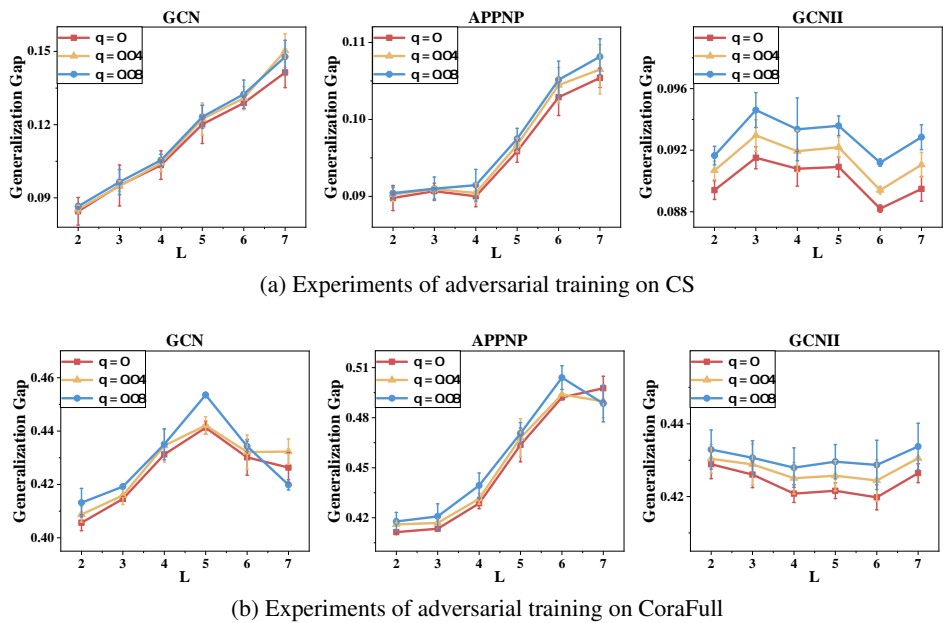

(a) Experiments of adversarial training on CS

(b) Experiments of adversarial training on CoraFull

*Figure 22.* The generalization gap for different adversarial perturbations $\theta$ (attacked by BIM) with increased number of layers $L$.

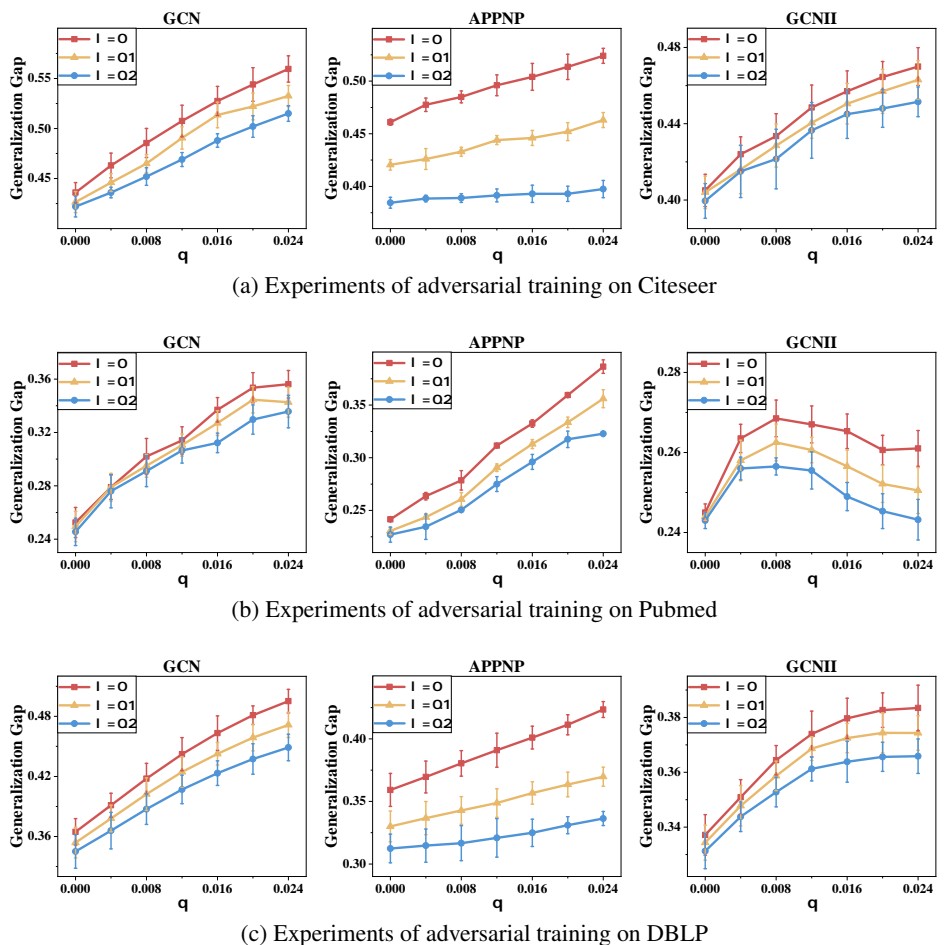

(a) Experiments of adversarial training on Citeseer

(b) Experiments of adversarial training on Pubmed

(c) Experiments of adversarial training on DBLP

*Figure 23.* The generalization gap for different regularization parameter $\lambda$ with increased perturbations $\theta$ (attacked by PGD).

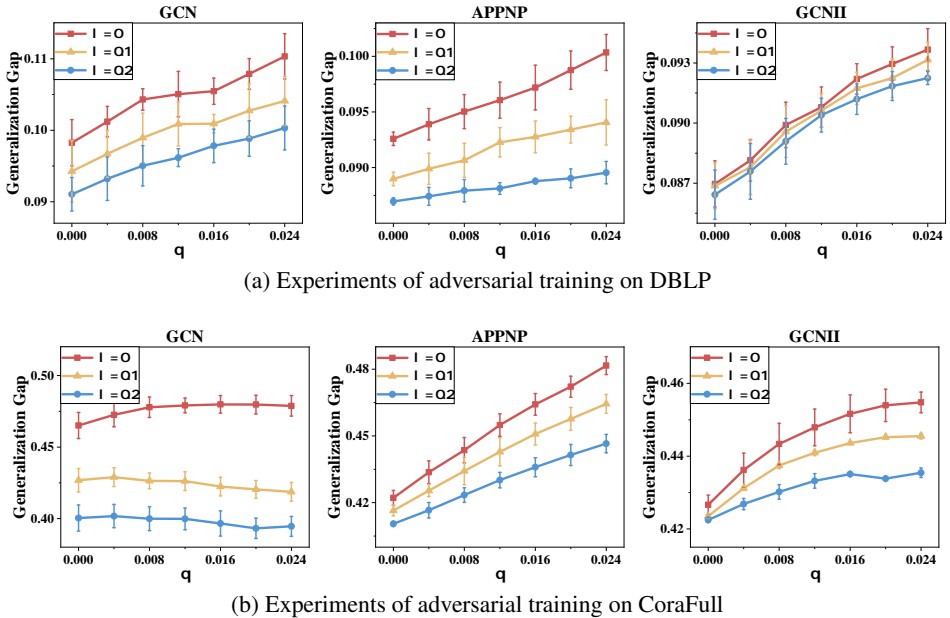

(a) Experiments of adversarial training on DBLP

(b) Experiments of adversarial training on CoraFull

*Figure 24.* The generalization gap for different regularization parameter $\lambda$ with increased perturbations $\theta$ (attacked by PGD).

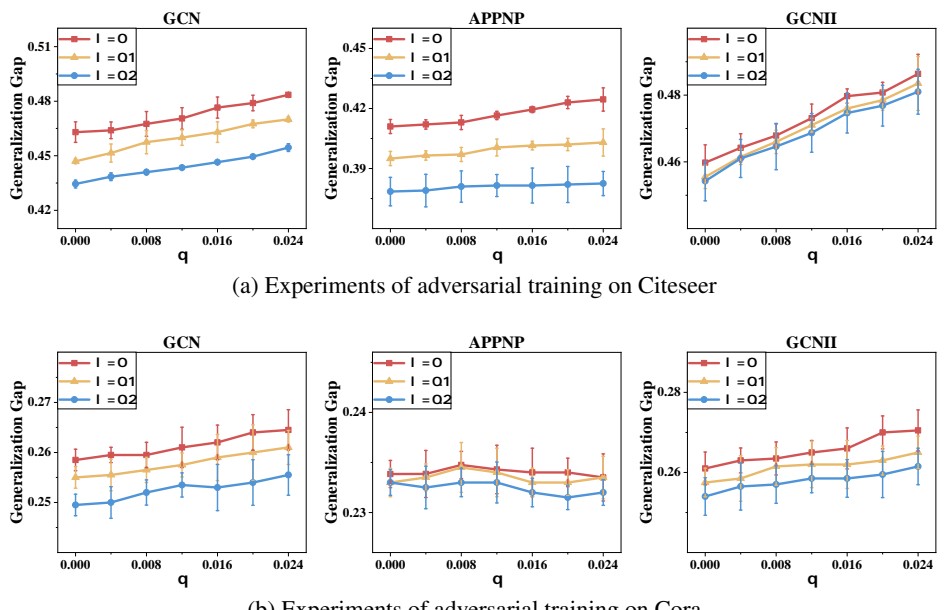

Figure 25. The generalization gap for different regularization parameter $\lambda$ with increased perturbations $\theta$ (attacked by FGSM).

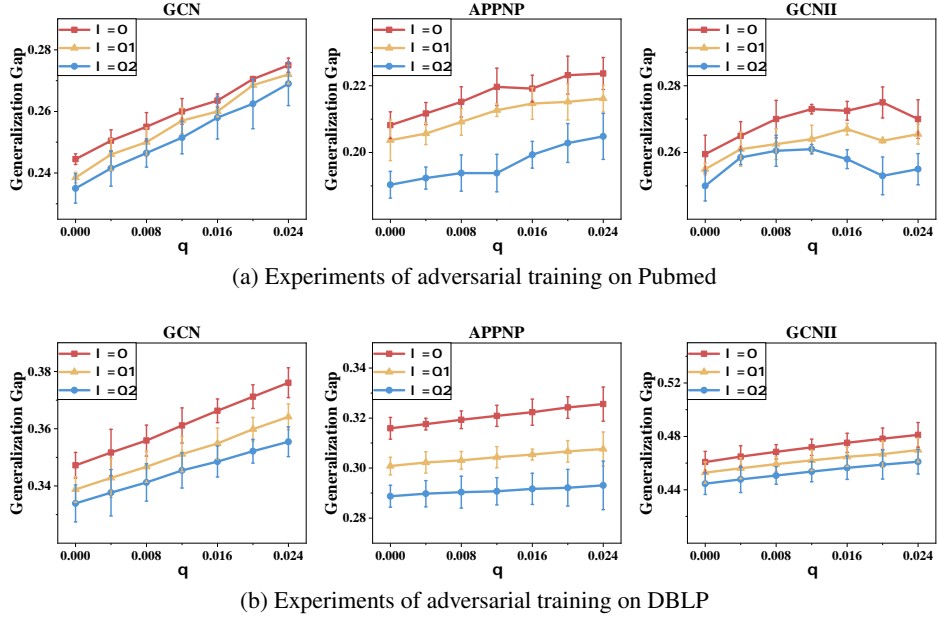

Figure 26. The generalization gap for different regularization parameter $\lambda$ with increased perturbations $\theta$ (attacked by FGSM).

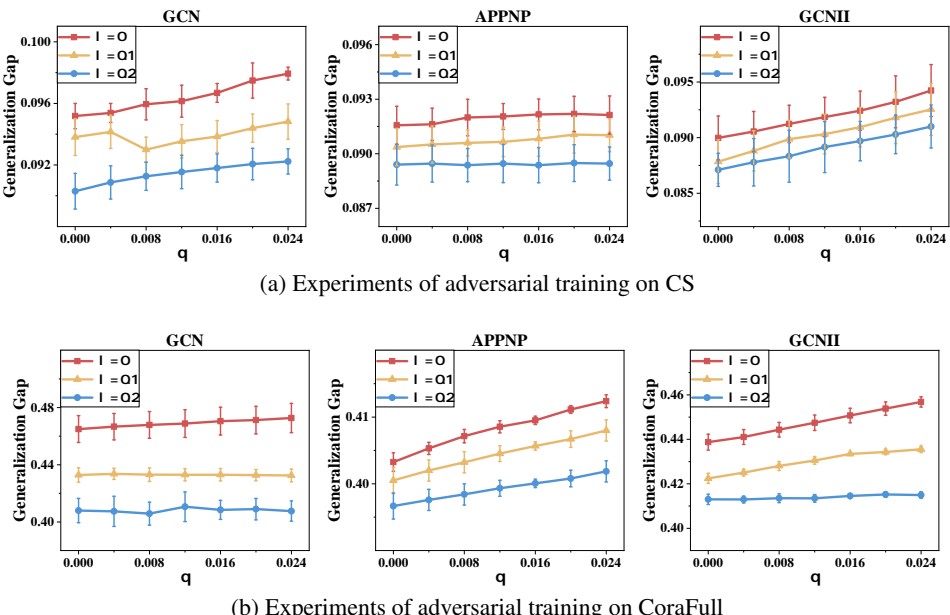

(a) Experiments of adversarial training on CS

(b) Experiments of adversarial training on CoraFull

*Figure 27.* The generalization gap for different regularization parameter $\lambda$ with increased perturbations $\theta$ (attacked by FGSM).

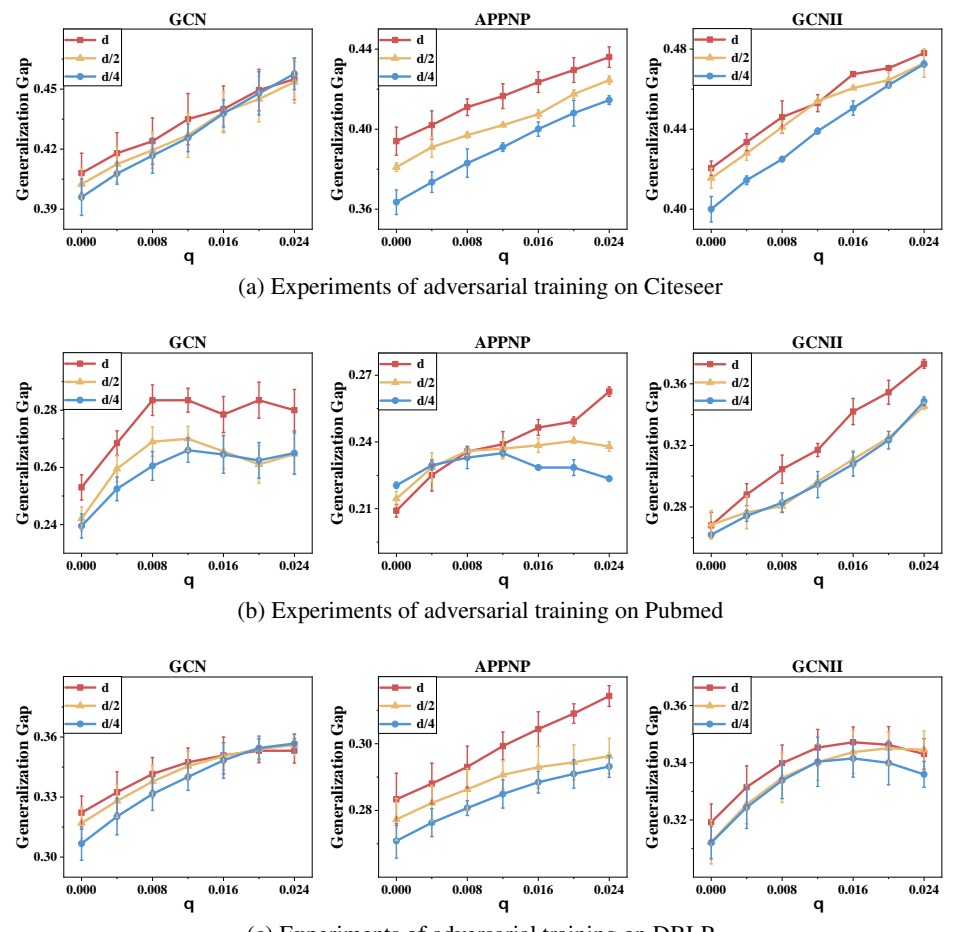

(a) Experiments of adversarial training on Citeseer

(b) Experiments of adversarial training on Pubmed

(c) Experiments of adversarial training on DBLP

*Figure 28.* The generalization gap for different input feature dimension $d$ with increased perturbations $\theta$ (attacked by PGD).

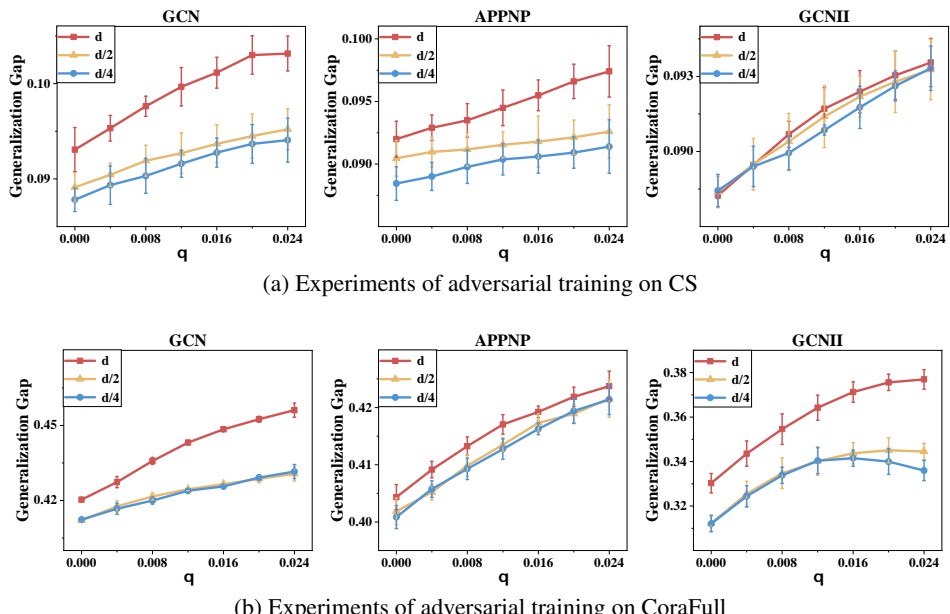

(a) Experiments of adversarial training on CS

(b) Experiments of adversarial training on CoraFull

*Figure 29.* The generalization gap for different input feature dimension $d$ with increased perturbations $\theta$ (attacked by PGD).

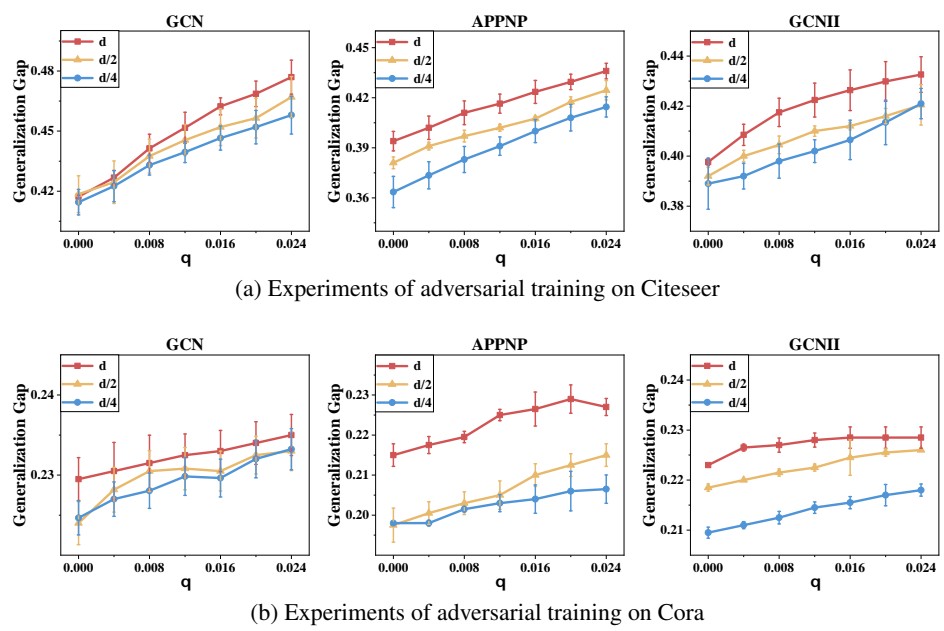

(a) Experiments of adversarial training on Citeseer

(b) Experiments of adversarial training on Cora

*Figure 30.* The generalization gap for different input feature dimension $d$ with increased perturbations $\theta$ (attacked by BIM).

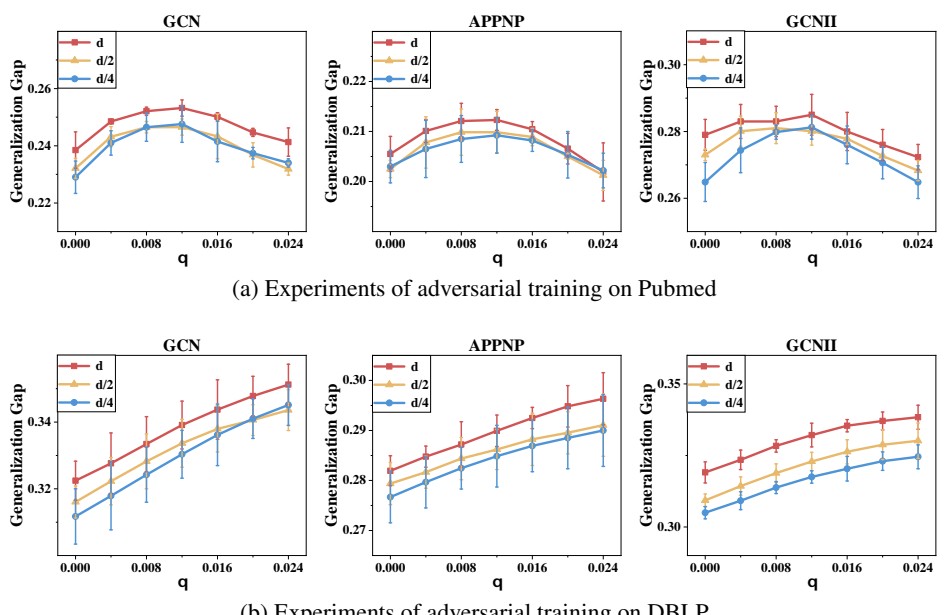

(a) Experiments of adversarial training on Pubmed

(b) Experiments of adversarial training on DBLP

*Figure 31.* The generalization gap for different input feature dimension $d$ with increased perturbations $\theta$ (attacked by BIM).

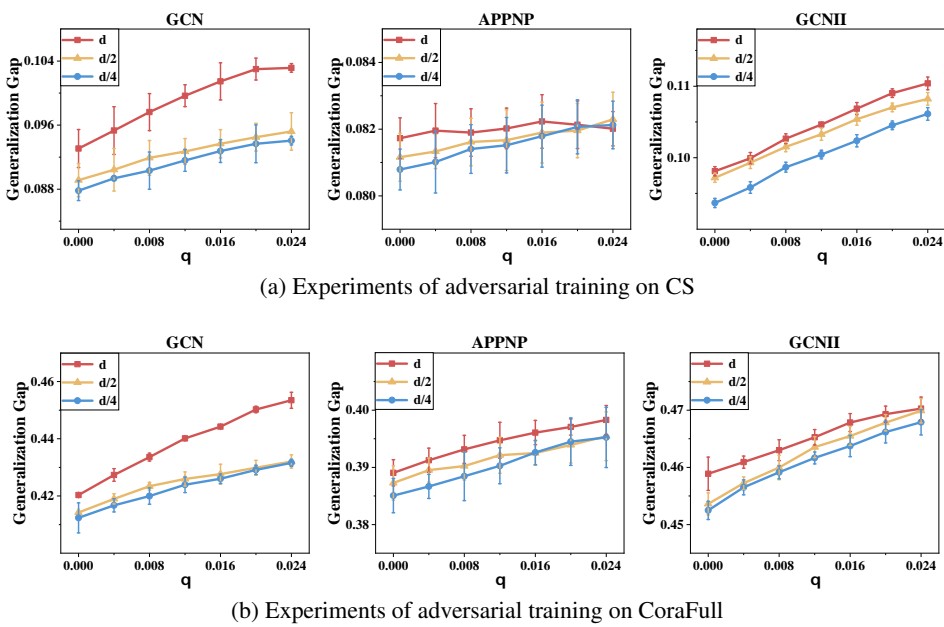

(a) Experiments of adversarial training on CS

(b) Experiments of adversarial training on CoraFull

*Figure 32.* The generalization gap for different input feature dimension $d$ with increased perturbations $\theta$ (attacked by BIM).

