# OpenReview forum: "Adversarial Robust Generalization of Graph Neural Networks"
_ICML.cc/2025/Conference — ICML 2025 poster_

### Official Review · Reviewer_5KXc · 2025-03-11

**Overall Recommendation:** 1

**Summary:**

The paper investigates the adversarial robustness of Graph Neural Networks (GNNs) in node classification tasks. The authors propose a high-probability generalization bound for GNNs under adversarial attacks using covering number analysis. They derive bounds for several popular GNN models (GCN, GCNII, APPNP) and analyze the impact of architectural factors on adversarial generalization. The paper also provides experimental results on benchmark datasets to validate the theoretical findings, showing that factors like model architecture, graph filters, and regularization parameters influence the generalization gap under adversarial attacks.

**Claims And Evidence:**

The paper makes several claims regarding the generalization bounds of GNNs under adversarial attacks. While the theoretical framework is well-structured, the evidence supporting these claims is not entirely convincing.

The experimental results, though consistent with the theoretical predictions, are limited in scope and do not fully validate the broad applicability of the proposed bounds.

The authors rely heavily on synthetic or controlled settings, and the generalization to real-world scenarios remains unclear.

Additionally, the paper lacks a thorough comparison with state-of-the-art adversarial training methods, which weakens the claim of providing a comprehensive understanding of adversarial robustness in GNNs.

**Essential References Not Discussed:**

The referred papers (Szegedyetal.,2013; Goodfellowetal.,2014) are irrelevant to GNN applications

The paper misses many relevant papers on GNN attacks and defenses

**Experimental Designs Or Analyses:**

The experimental design is reasonable but lacks depth. The authors evaluate the generalization gap under adversarial attacks for different GNN models, but the experiments are limited to a few datasets and attack methods. The results, while consistent with the theoretical predictions, do not provide strong empirical evidence for the robustness of the proposed bounds. The paper would benefit from more extensive experiments, including comparisons with state-of-the-art adversarial training methods and evaluations on larger, more diverse datasets.

**Methods And Evaluation Criteria:**

The methods proposed in the paper, particularly the covering number analysis, are theoretically sound and appropriate for analyzing the adversarial robustness of GNNs. However, the evaluation criteria are somewhat limited. The experiments are conducted on standard benchmark datasets, but the adversarial attacks used (e.g., PGD) are relatively simple and do not cover the full spectrum of possible adversarial perturbations. The paper would benefit from evaluating the proposed bounds against more diverse and challenging attack scenarios, as well as comparing with other adversarial training techniques.

**Other Comments Or Suggestions:**

Whats the key challenges/difficulties of the derived theoretical results, compared with the existing theoretical results on non-graph data?

Can the proposed theoretical result be applied to graph structure perturbation?

What type of GNN architecture is suited to the derived theoretical results?

How to calculate the generalization gap in the evaluations? The paper uses many assumptions which makes me doubtful on calculating the values of variables in the theoretical gap.

**Other Strengths And Weaknesses:**

Strengths:
* The paper addresses an important and timely problem in the field of adversarial robustness for GNNs.
* The theoretical framework is well-structured and provides a solid foundation for analyzing the generalization properties of GNNs under adversarial attacks.

Weaknesses:
* The empirical evaluation is limited in scope and does not fully validate the broad applicability of the proposed bounds.
* The paper lacks a thorough comparison with state-of-the-art adversarial training methods, which weakens its claim of providing a comprehensive understanding of adversarial robustness in GNNs.
* The assumptions made in the theoretical analysis (e.g., Lipschitz continuity) are not thoroughly discussed, and their practical implications are not explored in depth.
* The paper does not clearly articulate how its contributions advance the state-of-the-art or provide new insights beyond existing work.
* The theoretical results is only for node feature perturbation, while graph structure perturbation is more common against GNNs.

* The paper misses many relevant work on GNN attacks and defenses

**Questions For Authors:**

While the paper presents an interesting theoretical framework for analyzing the adversarial robustness of GNNs, the empirical evaluation is insufficient to support the broad claims made by the authors. The lack of comparison with state-of-the-art methods and the limited scope of the experiments weaken the paper's contribution to the field. Additionally, the assumptions made in the theoretical analysis are not thoroughly discussed, and their practical implications are not explored in depth. For these reasons, I recommend rejecting the paper in its current form.

**Relation To Broader Scientific Literature:**

The paper is well-situated within the broader literature on adversarial robustness and GNNs. It builds on prior work in adversarial training and generalization analysis, particularly in the context of graph-structured data. However, the paper does not sufficiently highlight how its contributions advance the state-of-the-art.

While the theoretical bounds are novel, the practical implications and applications of these bounds are not clearly articulated. The paper would benefit from a more detailed discussion of how the proposed bounds compare to existing methods and what new insights they provide.

**Theoretical Claims:**

The theoretical claims are based on covering number analysis, which is a well-established tool in statistical learning theory. The proofs provided in the appendix appear to be correct, but the paper lacks a detailed discussion of the assumptions made (e.g., Lipschitz continuity of the loss function and model architecture).

These assumptions may not hold in practice, especially for more complex GNN architectures or non-smooth loss functions. For instance, though Assumption4.1 maybe satisfied in standard non-graph neural networks, it could be violated in graph neural networks due to the message passing on the interdependent graph data.

The paper would benefit from a more thorough exploration of the limitations of these assumptions and their impact on the generalization bounds.

---

> ### Author Rebuttal · Authors · 2025-03-29
>
> We sincerely appreciate your thorough review and valuable suggestions. However, **we would like to clarify the first misunderstanding below**:
>
> > 1. Lack of new insights, comparison with existing methods, and application to real-world scenarios.
>
> **A1**: Generally speaking, our focus **does not lie in proposing a competitive algorithm tailored for a real-world application scenario** (including real-world datasets and attack scenarios). Instead, this paper **aims at a broader theoretical exploration of the robust overfitting in a general adversarial scenario**. Our work not only develops a **novel analytical framework** for general GNNs (Theorem 4.8), but also provides **helpful insights** into model construction and algorithm designs (Proposition 4.14~4.18).
>
> So, it is imperative for us to further clarify that  **" The lack of comparison with state-of-the-art methods and the limited scope of the experiments weaken the paper's contribution to the field" is not valid.**
>
> To be specific, this paper focuses on the robust overfitting phenomenon of GNN and provides theoretical guidance for improving their robust generalization in a general adversarial scenario.  Based on our theoretical results, our empirical evaluation focuses on the influencing factors (some model architecture-related factors, like graph filter, weight norm, hyperparameters, etc.) and demonstrates their important roles in improving (or deteriorating) the adversarial generalization.
>
> > 2. Analytical challenges introduced by graph data.
>
> **A2**:  **Challenge 1**:  The information interaction of nodes leads to the correlation of perturbations between different nodes, making the adversarial perturbation set of graph data different from that of non-graph data. **Solvement 1**: In adversarial settings, we search for the worst perturbation vector $\delta$ from all node features that consists of a perturbation matrix. Then, by incorporating the worst perturbation vector  $\delta$  into coverage analysis, Lemma 4.6 reveals an additional term $(\frac{6\theta C_{\ell}K_f}{\epsilon})^d$ influencing generalization caused by the interaction between perturbed nodes.
>
>  **Challenge 2**: Each node in GNN aggregates messages from its neighbor nodes through the message-passing mechanism, making the complexity measure of GNN model class different from NN. **Solvement 2**: In the decomposition of the propagation process (Proposition 4.14~4.18), we pay attention to the information interaction of graph data in the propagation process, which is reflected in the graph filter $\sum_{j=1}^n[g(A)]_{ij}$.
>
> > 3. Lack of discussion of the assumptions made.
>
> **A3**: Our analytical framework doesn't require smoothness assumptions of the loss function. This paper only needs the Lipschitz continuity assumption of loss function (Assumption 4.2) and activation function (Assumption 4.10), which can be easily satisfied by some commonly used functions (eg, cross-entropy and hinge loss; Sigmoid and ELU).  Other assumptions about the norm constraint of input feature and weight matrix are also usually used in literature [1, 2].
>
> In particular, for **Assumption 4.1**, we give the specific Lipschitz constant of each GNN model. For example, for a two-layer GCN, and $X(\delta)=[x_1+\delta,\dots,x_n+\delta]$, we have
>
> $\Vert f_i(A,X(\delta),W)-f_i(A,X(\delta'),W)\Vert$
>
> $\leq\rho_2\Vert\sum_{j=1}^n[g(A)] _ {ij}[\sigma(g(A)X(\delta)W_1] _ {j*}W_2-\sum_{j=1}^n[g(A)] _ {ij}[\sigma(g(A)X(\delta')W_1] _ {j*}W_2\Vert$
>
> $\leq \rho_2w_2 \Vert g(A)\Vert\max_j\Vert[\sigma(g(A)X(\delta)W_1] _ {j*}-[\sigma(g(A)X(\delta')W_1] _ {j*}\Vert$
>
> $\leq\rho_1\rho_2w_1w_2\Vert g(A)\Vert^2\max_j\Vert [X(\delta)] _ {j*}-[X(\delta')] _ {j*}\Vert$
>
> $\leq K_{GCN}\Vert\delta-\delta'\Vert$.
>
> Given the GCN with ELU activation and normalized filter ($\rho=1$, $\Vert g(A)\Vert=1$), conducting adversarial training with Algorithm 1 ($\Vert W\Vert$ is controlled) could lead to a controllable and small Lipschitz constant.
>
> > 4. Irrelevance and misses of some referred papers.
>
> **A4**: Thanks for pointing out the inaccuracy and misses. We will rectify the incorrect references (see [3,4]).  As this paper focuses more on the theoretical analysis for adversarial training of GNNs, we will add additional related works about GNN attacks and defenses in the appendix for reference.
>
> > 5. Applicable GNN types
> > 6. Extension to structure perturbations.
>
> **A5/A6**: Please refer to  **A2/A4** to the first reviewer (jE3b).
>
> ------
>
> [1] Zhou, X. and Wang, H. The generalization error of graph convolutional networks may enlarge with more layers. Neurocomputing, 424:97–106, 2021.
>
> [2] Tang, H. and Liu, Y. Towards understanding generalization of graph neural networks. ICML, pp. 33674–33719. PMLR, 2023.
>
> [3] Ben, F., et al. Single-node attacks for fooling graph neural networks. Neurocomputing, 513:1–12, 2022.
>
> [4] Liu, T., et al. Nt-gnn: Network traffic graph for 5g mobile iot android malware detection. Electronics, 12(4):789, 2023.

---

### Official Review · Reviewer_fe5j · 2025-03-12

**Overall Recommendation:** 3

**Summary:**

This paper investigates the generalization ability of graph neural networks (GNNs) under adversarial training, which is an important and widely interested research direction. The paper first proposes a high probability generalization limit and analyzes the generalization ability of GNN under adversarial training through covering number analysis. This provides theoretical support for understanding the behavior of GNN under adversarial learning. The paper selected three representative GNN variants for experiments and proposed a new adversarial training algorithm, which proved its effectiveness in improving the stability of GNN training through experiments.

**Claims And Evidence:**

This paper has rigorous formula derivation, which can theoretically support the method proposed in this article. Clear logic in experimental procedures and methods.

**Essential References Not Discussed:**

I'm not clear enough.

**Experimental Designs Or Analyses:**

In the main text and appendix of the paper, the author conducted extensive experiments on six benchmark datasets to support the claims of this article. However, the description of Algorithm 1 seems somewhat vague.

**Methods And Evaluation Criteria:**

The proposed methods and evaluation criteria mainly focus on analyzing the adversarial generalization ability of GNNs, which can be applied to the robustness research of GNNs and improving their generalization ability.

**Other Comments Or Suggestions:**

nothing

**Other Strengths And Weaknesses:**

Advantages
1.The paper has a clear structure, rigorous logic, and standardized use of symbols.
2.This paper establishes the high probability generalization limit of GNNs in adversarial learning, providing theoretical guidance for the design and training of GNNs.
Weekness
The derivation process of the coverage analysis method is relatively complex and may be difficult to understand and apply.

**Questions For Authors:**

1.You established a high probability generalization limit for GNNs in adversarial learning in your paper. Can the derivation process of this boundary be further simplified?
2.You have proposed an adversarial training algorithm to learn a robust GNNs, but it seems that there is no clear description of the algorithm.
3. Why does the experimental part of this paper seem to lack a comparative study with previous work?

**Relation To Broader Scientific Literature:**

I'm not clear enough.

**Theoretical Claims:**

This paper has rigorous and extensive formula reasoning, and in my reading, I did not see any obvious errors.

---

> ### Author Rebuttal · Authors · 2025-03-29
>
> We deeply thank you for acknowledging the rigorous logic, clear structure, and extensive formula reasoning of our work.  Below are our detailed responses.
>
> > 1. Why does the experimental part of this paper seem to lack a comparative study with previous work?
>
> **A1**: Generally speaking, our focus **does not lie in proposing a competitive algorithm tailored for a specific application scenario**. Instead, this paper **aims at a broader theoretical exploration of robust overfitting in a general adversarial scenario**.
>
> To be specific, this paper focuses on the robust overfitting phenomenon of GNNs and provides theoretical guidance for improving their robust generalization in a general adversarial scenario.  Our work not only develops a **novel analytical framework** for general GNNs (Theorem 4.8), but also provides **helpful insights** into model construction and algorithm designs (Proposition 4.14~4.18). Based on our theoretical results,  our empirical evaluation focuses on the influencing factors (some model architecture-related factors, like graph filter,  weight norm, number of layers, hyperparameters, etc.) and demonstrates their important roles in improving (or deteriorating) the adversarial generalization.
>
> > 2. The description of Algorithm 1 seems somewhat vague.
>
> **A2**: Thanks for your helpful suggestion. Let $\mathcal{A}$ be a gradient-based attack algorithm (e.g., PGD, BIM, Mettack), the updated version is provided below.
>
> ---
>
> **Input**: Graph $G=(A,X)$, dataset $S$, perturbed dataset $\tilde{S}$, perturbation budget $\theta$, regularization parameter $\lambda$, initialization $W_0$, learning rate $\eta$, number of iterations $T$.
>
> **while** $t<T$ **do**
>
> $\tilde{S}\leftarrow\emptyset$.
>
>  **for** $i = 1, 2, \ldots, n$ **do**
>
> For the input matrix $X_t=[x_{1,t},\dots,x_{n,t}]$, perturb $\tilde{X_t} \leftarrow X_t+\mathcal{A} (X_t,A, \theta)$.
> ​
> For each perturbed node in $\tilde{X_t}=[\tilde{x} _ {1,t},\dots,\tilde{x} _ {n,t}]$, append it to $\tilde{S_t}$ and choose $m$ samples randomly to the training set $\tilde{S} _ {m,t}$.
> ​
> **end for**
> ​
> Define a new objective $L(W _ {i,t})=\frac{1}{m}\sum_{\tilde{X} _ {i,t}\in\tilde{S} _ {m,t}} \ell(f _ {i,t}(A,X, W), y _ {i,t}) + \lambda \Vert W _ {i,t}\Vert _ {\infty}$.
> ​
> For all $i\in [m]$, update $W_t$ using SGD: $W_{i,t+1}\leftarrow W_{i,t}-\eta\nabla {L}(W_{i,t})$.
>
> **end while**
>
> ---
> > 3. Can the derivation process of coverage analysis be simplified? It seems to be difficult to understand and apply.
>
> **A3**: Covering number is a commonly used tool in (adversarial) generalization analysis [1,2]. Please let me briefly introduce our analysis technology first.
>
> **1.  For the maximization over adversarial loss ( $\max_{\tilde{X}}\ell(f_i(A,\tilde{X},W),y_i)$)**(Lemma 4.4). We construct new function classes ( $L$ and $L_{dis}$) and use their covering number to control the covering number of the adversarial loss class $L_{adv}$.
>
> **2. For the interplay between perturbed nodes** (Lemma 4.6). We cover the perturbation set $\mathcal{B}$ and transform the cover of the loss class $L_{disc}$ to that of the perturbed model function class $\hat{F}$ .
>
> **3** (Theorem 4.8). Now we can obtain the relation between the adversarial generalization and the covering number of GNN model class!
>
> **4. Covering number derivation** (Proposition 4.14~4.18). We utilize the relation between the model function $\hat{F}$ and its weight matrix $W$ to derive the covering number of the perturbed GNN class  $\mathcal{N}(\hat{F},\epsilon,\Vert\cdot\Vert)$ by that of the weight matrix set $\mathcal{N}(${$W_j,\Vert W_j\Vert\leq w_j$}$,\epsilon_j,\Vert\cdot\Vert)$.
>
> Actually, the main step that requires complex calculations is step 4, which is due to the propagation rules of GNN.
>
> Therefore, given the Lipschitz continuity assumption about the adjacency matrix, our work can be applied to topology attack scenarios (by step 1-3). Since our covering number-based framework is applicable to general GNNs, given a specific GNN model with the relation between model function and weight matrix, it can be applied to other types of GNN models (by step 4).
>
> ----
>
> [1] Tang, H. and Liu, Y. Towards understanding generalization of graph neural networks. ICML, pp. 33674–33719. PMLR, 2023.
>
> [2] Tu, Z. et al. Theoretical Analysis of Adversarial Learning: A Minimax Approach. Neurips, 32, 2019.

---

### Official Review · Reviewer_1KKU · 2025-03-14

**Overall Recommendation:** 3

**Summary:**

This paper establishes an adversarial generalization bound of various GNNs, such as GCN, APPNP, GCNII, in the context of transductive learning. The authors provide some guidlines for adversarial generalization based on the theoretical results. The guidlines based on theoretical results are all validated in experimental results.

## update after rebuttal

I had no specific concerns about this paper during my initial review. Therefore, I am maintaining my original score after the rebuttal.

**Claims And Evidence:**

Yes, the logical flow of the paper is sound, and all claims may be well supported through both theoretical analysis and empirical evidence.

**Essential References Not Discussed:**

None

**Experimental Designs Or Analyses:**

The theoretical analyses appear sound and comprehensive. Moreover, the empirical results are thoughtfully designed to support and validate the theoretical findings.

**Methods And Evaluation Criteria:**

While the paper does not propose a specific method, it offers valuable hyperparameter guidelines for improving robustness across different GNN backbones. The evaluations are conducted on a variety of well-established datasets, yielding consistent and reasonable results. Additionally, the inclusion of diverse GNN backbones strengthens the validity and generalizability of the theoretical findings.

**Other Comments Or Suggestions:**

- Fig 2(a) caption might be "Experiments of adversarial training for APPNP."

**Other Strengths And Weaknesses:**

- Important topic and solid theoretical analyses.
- Writing quality is very good.
- All theoretical findings are supported by the empirical results.

**Questions For Authors:**

No

**Relation To Broader Scientific Literature:**

Contributing to Learning Theory of GNN adversarial generalization.

**Theoretical Claims:**

I am not confident in assessing the correctness of the theoretical claims or proofs. I did not verify the detailed steps of the proofs, so I cannot confirm their validity. Please take this into consideration.

---

> ### Author Rebuttal · Authors · 2025-03-26
>
> We deeply appreciate your acknowledgment of the solid and comprehensive theoretical analysis and the thoughtfully designed empirical results presented in our paper. Thanks for pointing out the typo, and we will fix it in the future version.

---

### Official Review · Reviewer_jE3b · 2025-03-18

**Overall Recommendation:** 3

**Summary:**

The paper investigates the adversarial robust generalization of GNNs through a theoretical lens. It derives high-probability generalization bounds for general GNNs in adversarial settings using covering number analysis. The key insight is modeling the adversarial loss class’s complexity by constructing a perturbation cover and analyzing GNN architectures (e.g., GCN, APPNP, GCNII). Theoretical results reveal that adversarial generalization depends on factors like perturbation budget, model depth, weight norms, and graph filters. Experiments on benchmark datasets validate these findings, showing that normalized graph filters, shallower architectures, and regularization reduce generalization gaps.

**Claims And Evidence:**

The claims are supported by theoretical proofs.

**Essential References Not Discussed:**

There are no essential related works that are not cited.

**Experimental Designs Or Analyses:**

Experiments vary layers, filters, and attack budget on standard datasets, but there are some weaknesses:

1. The accuracy difference metric varies depending on the dataset split and does not appear to have been tested for a sufficient number of randomized splits.
2. The datasets used are small and large-scale datasets are missing.
3. Experimental validation was performed on only three GNNs.

**Methods And Evaluation Criteria:**

The methods and evaluation make sense.

**Other Comments Or Suggestions:**

I don't have other comments or suggestions.

**Other Strengths And Weaknesses:**

There are some weaknesses:
- The work seems to discuss only the case of counter-attacks against node attributes. However, for graph learning, attacks against structures are more extensively studied.
- Can the proposed theoretical framework be adapted to other commonly used GNNs such as GAT, GraphSAGE, GIN, etc.?
- The experiments conducted were limited, as described earlier.

**Questions For Authors:**

Please see above.

**Relation To Broader Scientific Literature:**

The work extends adversarial generalization theory to GNNs, addressing unique challenges like transductive learning.

**Theoretical Claims:**

The proofs are logically sound.

---

> ### Author Rebuttal · Authors · 2025-03-29
>
> Thank you very much for your valuable comments! Please refer to our response below.
>
> > 1. Lack of testing for randomized splits of datasets.
>
> **A1**:  Thanks for pointing out the lack of considering the impact of dataset splitting. Taking two-layer GCN and two datasets for example, we show the generalization gap under different random split rates of training data (including 0.1, 0.3, and 0.5) in the table below.
>
> |              |                 | Cora            |                 |                 | CoraFull        |                 |
> | ------------ | --------------- | --------------- | --------------- | --------------- | --------------- | --------------- |
> |              | 0.1             | 0.3             | 0.5             | 0.1             | 0.3             | 0.5             |
> | $\theta=0$   | 0.478$\pm$0.013 | 0.296$\pm$0.012 | 0.255$\pm$0.014 | 0.396$\pm$0.003 | 0.284$\pm$0.002 | 0.208$\pm$0.003 |
> | $\theta=0.1$ | 0.516$\pm$0.014 | 0.306$\pm$0.012 | 0.261$\pm$0.012 | 0.418$\pm$0.003 | 0.301$\pm$0.002 | 0.223$\pm$0.002 |
> | $\theta=0.2$ | 0.551$\pm$0.016 | 0.309$\pm$0.014 | 0.269$\pm$0.015 | 0.434$\pm$0.002 | 0.311$\pm$0.001 | 0.234$\pm$0.002 |
>
> The results show that they have **consistent trends** under increasing perturbation budgets. We will include a comprehensive version of the experiments in our future version.
>
> > 2.  Adaptation to other GNNs.
>
> **A2**:  Our results provide a general analytical framework for GNN, and give three classical examples of spectral GNN. Although specific results are not presented in the paper, **spectral GNN** like SGC, AGCN and GPR-GNN are feasible.
>
> Moreover, we are reasonably confident in extending our results to other types of GNN (spatial-based GNN).  Taking a single-head GAT for example, as for the model function  $f_i(X,W)=\sigma_2(\sum_j\alpha_{ij}W_2X_{j*})$, and $\alpha_{ij}=softmax(\sigma_1(w[W_1X_{i*}\Vert W_1X_{j*}]))$, we can get the relation between the covering number of the model function class and that of the weight matrix set (i.e. $\mathcal{N}(\hat{F},\epsilon,\Vert \cdot\Vert)$ and $\mathcal{N}(\{W_j,\Vert W_j\Vert\leq w_j\},\epsilon_j,\Vert\cdot\Vert)$),  which can be applied to our analytical framework.
>
> > 3. Large-scale datasets are missing.
>
> **A3**:  Thanks for your valuable suggestions. Large-scale datasets like Nell and ogbn-arxiv will be included in our future versions.
>
> > 4. Extension to attacks against structures.
>
> **A4**:  Given the similar adversarial settings for topology attacks and node attacks, **we suggest that the methodology (e.g., Lemma 4.4 and Theorem 4.8) developed in this paper could be expanded upon the topology attack.**
>
> To be more specific, let the adversarial graph be generated from $\{\tilde{A}:\Vert\tilde{A}\Vert\leq \gamma \}$, where $\tilde{A}=A - A' $ denotes the perturbation matrix added to the original adjacency matrix. The adversarial loss w.r.t. adversarial graph is defined by $\max_{\Vert\tilde{A}\Vert \leq \gamma} \ell(f_i(\tilde{A},X,W),y_i) $.
>
> Analyzing analogously to the node attacks. For each function $f\in\mathcal{F}$ and a fixed $\tilde{A} _ c\in\mathcal{A}$, we construct a new function $h:\mathcal{Z}\rightarrow(\mathbb{R}^n)^{\mathcal{A}}$ as $h(z_i,\tilde{A} _ c)=\ell(f_i(\tilde{A} _ c,X,W),y_i)$. The adversarial loss is denoted by $\max_{\tilde{A}\in\mathcal{A}} h(z_i,\tilde{A})=\max_{\Vert\tilde{A}\Vert \leq \gamma}\ell(f_i(\tilde{A},X,W),y_i) $ for any $\tilde{A}\in\mathcal{A}$.
>
> From the definition of covering number, we construct the cover of the class $H$ of function $h(z_i,\tilde{A} _ c)$ and obtain the cover of the class $H_{adv}$ of function $\max_{\tilde{A}\in\mathcal{A}} h(z_i,\tilde{A})$. Thus, the following inequality holds
>
> $\mathcal{N}(H_{adv},\epsilon,|\cdot| _ {\infty})\leq\mathcal{N}(H,\epsilon,\Vert\cdot\Vert _ {\infty}).$
>
> Next, we construct a cover to control the infinite class $\mathcal{A}$ by $\mathcal{C} _ {\mathcal{A}}:=${$\hat{A} _ j,j\in[N_A]$}.  Similarly, we can obtain the relation between $\mathcal{N}(H,\epsilon,\Vert\cdot\Vert_{\infty})$ and $\mathcal{N}(H_{dis},\epsilon,\Vert\cdot\Vert_{\infty})$, which needs an assumption that
>
> $|\max h(z,A)-\max h(z,A')|\leq L_A\Vert A-A'\Vert$,
>
>  where the constant $L_A$ can be obtained if given specific GNN models.
>
> This allows us to solve the measurement difficulty caused by the graph structure perturbations and apply it to our main results (Theorem 4.8). The remaining analysis will be left to future work.

---

> > ### Comment · Reviewer_jE3b · 2025-04-08
> >
> > Thank you for your response, it has addressed some of my concerns, but the limited experiments and the utility of the theory are still my concerns. I have raised my score accordingly but am leaning towards a borderline acceptance.

---

> > > ### Author Response · Authors · 2025-04-08
> > >
> > > Thank you so much for increasing the score. We understand the limitations you mentioned and will take them as guidance for further improvement. Your recognition means a great deal to us.

---

### Decision · Program_Chairs · 2025-05-01

**Decision:**

Accept (poster)

**Comment:**

The paper derive adversarial generalisation bounds for GNNs via covering number analysis. The topic is timely and important. Three out of four reviewers positively evaluated the paper with a "weak accept" and found the theoretical analyses to be solid. Issues regarding the comprehensiveness of the experimental validation were raised by several reviewers (small datasets, few models, limited scope). In my opinion this is not a major issues since the main contribution of the paper is theoretical. I disagree with Reviewer 5KXc that "thorough comparison with state-of-the-art adversarial training methods" is necessary, although it would of course strengthen the paper.

In general, I believe that the authors sufficiently addressed all the issues raised by Reviewer 5KXc in the rebuttal, even though the reviewer did not update their score. Specifically, the authors explained that their goal is to theoretically explain (adversarially) robust overfitting.  The authors also articulated how the paper's contributions advance the state-of-the-art, including how they address the analytical challenges introduced by graph data. The authors also clarified the assumptions on the loss functions, which are standard and reasonable.

Nonetheless, I suggest the authors to carefully revise their writing to avoid that impression of "broad claims made by the authors". I also encourage the authors to update the paper to reflect the discussion, e.g. the comments regarding an extension to topology attacks. I recommend the paper to be accepted.